# Initial conditions combine with sensory evidence to induce decision-related dynamics in premotor cortex

Pierre O. Boucher[1], Tian Wang[1], Laura Carceroni[2], Gary Kane[3], Krishna V. Shenoy ®[4,5,6,7,8,9,10,13] & Chandramouli Chandrasekaran ®[1,3,11,12] ✉

We used a dynamical systems perspective to understand decision-related neural activity, a fundamentally unresolved problem. This perspective posits that time-varying neural activity is described by a state equation with an initial condition and evolves in time by combining at each time step, recurrent activity and inputs. We hypothesized various dynamical mechanisms of decisions, simulated them in models to derive predictions, and evaluated these predictions by examining firing rates of neurons in the dorsal premotor cortex (PMd) of monkeys performing a perceptual decision-making task. Prestimulus neural activity (i.e., the initial condition) predicted poststimulus neural trajectories, covaried with RT and the outcome of the previous trial, but not with choice. Poststimulus dynamics depended on both the sensory evidence and initial condition, with easier stimuli and fast initial conditions leading to the fastest choice-related dynamics. Together, these results suggest that initial conditions combine with sensory evidence to induce decision-related dynamics in PMd.

There are 10 minutes to make it to the airport but your phone says you're still 12 minutes away. Seeing a yellow light in the distance you quickly floor it. You get to the intersection only to realize you have run a red light. The sight of the lights result in patterns of neural activity that respectively lead you to respond quickly to your environment (i.e., speed up when you see the yellow) and process feedback (i.e., slow down after running the red). This process of discriminating sensory cues to arrive at a choice is termed perceptual decision-making[1–5].

Research in invertebrates[6,7], rodents[8,9], monkeys[10,11], and humans[12,13] has attempted to understand the neural basis of perceptual decision-making. Barring few exceptions[14–16], these studies have focused on single neurons in decision-related brain regions[10,11,17,18].

However, currently the link between neural population dynamics in these brain areas and decision-making behavior, especially in reaction time (RT) tasks, is largely unclear. Here, we address this gap by using a dynamical systems approach[19–22].

The dynamical systems approach[21–23] posits that neural population activity (e.g., firing rates), $X$, is governed by a state equation of the following form:

$$\frac{dX}{dt} = F(X) + U \tag{1}$$

[1]Department of Biomedical Engineering, Boston University, Boston 02115 MA, USA. [2]Undergraduate Program in Neuroscience, Boston University, Boston 02115 MA, USA. [3]Department of Psychological and Brain Sciences, Boston University, Boston 02115 MA, USA. [4]Department of Electrical Engineering, Stanford University, Stanford 94305 CA, USA. [5]Department of Neurobiology, Stanford University, Stanford 94305 CA, USA. [6]Howard Hughes Medical Institute, HHMI, Chevy Chase 20815-6789 MD, USA. [7]Department of Bioengineering, Stanford University, Stanford 94305 CA, USA. [8]Stanford Neurosciences Institute, Stanford University, Stanford 94305 CA, USA. [9]Bio-X Program, Stanford University, Stanford 94305 CA, USA. [10]Department of Neurosurgery, Stanford University, Stanford 94305 CA, USA. [11]Center for Systems Neuroscience, Boston University, Boston 02115 MA, USA. [12]Department of Anatomy & Neurobiology, Boston University, Boston 02118 MA, USA. [13]Deceased: Krishna V Shenoy. ✉e-mail: cchandr1@bu.edu

Where $F$ represents the recurrent dynamics (i.e., local synaptic input) in the region of interest and usually considered fixed for a given brain area in a task. $U$ is the input from neurons outside the region of interest and depends on various task contingencies (e.g., sensory evidence). $X_0$ is the initial condition for these dynamics. In this framework, dynamics for every trial depend on both the initial condition and input and leads to distinct behavior on every trial.

The dynamical systems approach helps link time-varying, heterogeneous activity of neural populations and behavior[20,24,25]. In a study of motor planning, position and velocity of the neural population dynamics relative to the mean trajectory at the time of the go cue (i.e., initial condition or $X_0$) explained considerable variability in RTs[20] (see Fig. 1a). Similarly, in studies of timing, the initial condition encoded the perceived time interval and predicted the speed of subsequent neural dynamics and the reproduced time interval[26] (see Fig. 1b). In the same study, an input depending on a task contingency (gain) altered the speed of dynamics (Fig. 1b).

Here, we expanded on findings from motor planning and timing studies and investigated which dynamical system best described decision-related neural population activity in dorsal premotor cortex (PMd). To derive hypotheses about dynamics, we leveraged three results from prior studies. First, rate at which choice-selective activity emerges depends on the strength of the sensory evidence (e.g., auditory pulses, random dot motion, static red-green checkerboards, etc.)[8,10,18,27]. Second, in studies of speed-accuracy tradeoff, prestimulus neural activity is different for fast vs. slow blocks[16,28–31] (Fig. 1c). Finally, the prestimulus firing rates are altered by the outcome of the previous trial[32,33] (Fig. 1d). Based on these findings, we hypothesized four different dynamical mechanisms that could describe the data.

- The simplest dynamical system assumes initial conditions do not covary with RT or choice and that neural dynamics and behavior are driven largely by the sensory evidence (Fig. 1e).
- A second dynamical system assumes that the initial conditions do not vary, but that there are either systematic or random delays in sensory evidence processing[34], that alter choice-related dynamics and behavior (Fig. 1f).
- A third system assumes that initial conditions are biased towards one or another choice[35], correlate with RT, and that poststimulus dynamics are influenced by both initial conditions and sensory evidence (Fig. 1g).
- Finally, a fourth system assumes that initial conditions correlate with RT but not choice, and poststimulus dynamics depend on both sensory evidence and initial condition. Additionally, the changes in initial condition are in part due to the outcome of the previous trial (Fig. 1h).

We used these different candidate dynamical mechanisms to build recurrent neural networks with various constraints (Figs. S1a and S2) and simulate synthetic neural populations (Figs. S1b and S3). We analyzed these simulations of neural activity using dimensionality reduction, decoding, and regression analyses. These different dynamical mechanisms make distinct predictions about the principal component trajectories and whether prestimulus activity covaries with choice and RT. We used the predictions to analyze the firing rates of neurons recorded in PMd of monkeys performing a red-green RT perceptual decision-making task[18].

Neural population dynamics in PMd had the following properties. First, state space trajectories were ordered pre- and poststimulus as a function of RT, with such effects observed within a stimulus difficulty. Subsequent KiNeT[26] analysis of these trajectories suggested that faster RTs were associated with faster pre- and poststimulus dynamics as compared to slower RTs. Second, cross-validated single-trial analyses using tensor component analysis, dynamical systems, reduced-rank regression, decoding and regression analyses further corroborated that prestimulus neural state, that is the initial condition, only

predicted RT but not the eventual choice. Third, poststimulus choice-related dynamics depended on both the initial condition and the sensory evidence, with choice-related signals emerging faster for easier compared to harder trials but also modulated by the initial condition. Finally, initial conditions and choice-related dynamics depended on the outcome of the previous trial with pre- and poststimulus dynamics slower on trials following an error as compared to trials following a correct response.

Our results expand on the observations of ref. 20, that the prestimulus position and velocity of the neural trajectories in state space (i.e., initial conditions) are correlated with RT, as we demonstrate that 1) both inputs and initial conditions jointly control dynamics, and 2) that changes in the initial conditions are dependent upon previous outcomes. Together, the results suggest that decision-related activity in PMd is captured by a dynamical system (Fig. 1h) composed of initial conditions, that covary with RT and are dependent upon previous outcome, and inputs (i.e., sensory evidence) which combine to induce choice-related dynamics.

## Results

### Decision-making behavior is dependent on sensory evidence and internal state

We trained two macaque monkeys (O and T) to discriminate the dominant color of a central, static checkerboard composed of red and green squares (Fig. 2a). Fig. 2b depicts the trial timeline. The trial began when the monkey held the center target and fixated on the fixation cross. After a short randomized holding time (300–485 ms), a red and a green target appeared on either side of the central hold (target configurations were randomized). After an additional randomized target viewing time (400-1000 ms), the checkerboard appeared. The monkey's task was to reach to, and touch the target corresponding to the dominant color of the checkerboard. While animals were performing the task, we measured the arm and eye movements of the monkeys. We identified RTs as the first time when hand speed exceeded 10% of maximum speed during a reach. If the monkey correctly performed a trial, he was rewarded with a drop of juice and a short inter-trial interval (ITI, 300 to 600 ms across sessions) whereas if he made an error it led to a longer timeout ITI (ranging from ~1500 ms to ~3500 ms). Using timeouts for errors encouraged animals to prioritize accuracy over speed.

We used 14 levels of sensory evidence referred to as signed color coherence ($SC$, Fig. 2c) as it is dependent on the actual dominant color of the checkerboard. Unsigned coherence ($C$, Fig. 2c), which refers to the strength of stimuli, is independent of the actual dominant color of the checkerboard. Thus, there are 7 levels of $C$.

The behavioral performance of the monkeys depended on the signed coherence. In general, across all sessions, monkeys made more errors when discriminating stimuli with near equal combinations of red and green squares (Fig. 2d). We fit the proportion correct as a function of unsigned coherence using a Weibull distribution function to estimate slopes and psychometric thresholds (average $R^2$; T: 0.99, 75 sessions; O: 0.98, 66 sessions; Threshold ($\alpha$): Mean $\pm$ SD: T: 10.89 $\pm$ 1.37%, O: 16.78 $\pm$ 2.05%; slope ($\beta$): Mean $\pm$ SD over sessions, T: 1.26 $\pm$ 0.18, O: 1.10 $\pm$ 0.14).

As expected, monkeys were generally slower for more ambiguous checkerboards (Fig. 2e). However, per monkey regressions using unsigned coherence ($\log_{10}(C)$) to predict RTs only explained ~12.4% and ~1.5% of RT variability, for monkeys T and O respectively (Fig. 2f). These results suggest that while there is RT variability induced by differences in the sensory evidence, there is also an internal source of RT variability. Indeed, as the box plots in Fig. 2f show, a key feature of the monkeys' behavior is that RTs are quite variable within a coherence, including the easiest ones. In the subsequent sections, we investigated which dynamical mechanism (Fig. 1e–h) was most consistent with this RT variability and choice behavior.

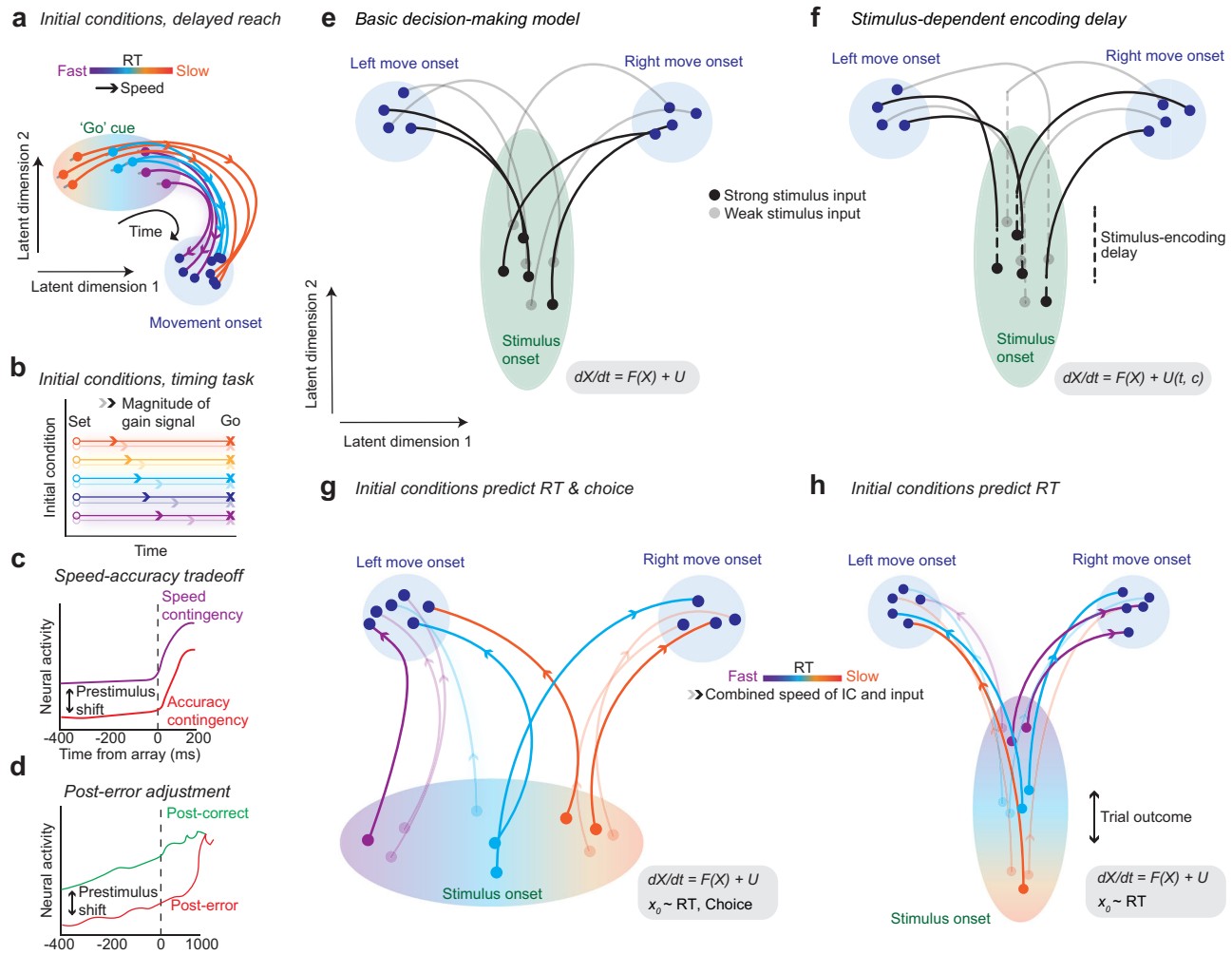

**Fig. 1 | Initial conditions and inputs predict subsequent neural dynamics and behavior. a** The initial condition hypothesis from delayed reach experiments[20] posits that the position and velocity of a neural state at the time of the go cue (initial condition) negatively correlates with RT. **b** The neural population state at the end of a perceived time interval and a gain modifier actuates the initial conditions (Set, circles) determining the speed (arrows) of subsequent dynamics and therefore when an action is produced (Go, X's)[26]. **c, d** Prestimulus neural activity differs for speed and accuracy contingencies for speed-accuracy tradeoff tasks[58] or after correct and error trials[33]. **e** Basic decision-making model with no prestimulus effects, where only the strength of sensory evidence determines RT and choice. **f** Stimulus-dependent encoding delay. Decision-making takes longer as a function of how long it takes to visually process stimuli (dotted line). **g** Biased initial conditions predict both RT and choice ($X_0$ - RT, choice) and combine with sensory evidence to lead to decisions. Initial neural states vary trial-to-trial, and are closer to

the movement onset state for one choice (here left). Trial outcomes have no effect on initial conditions in this model as initial conditions largely reflect a reach bias. **h** Overall dynamics depend on both the initial conditions, which solely predict RT ($X_0$ - RT), and the sensory evidence. The closer the initial condition is to a movement initiation state before checkerboard onset, the faster the velocity of the dynamics will be, leading to faster RTs. Previous outcomes shift these initial conditions such that the dynamics are either faster or slower. Current population state at stimulus onset/go cue (dots within an ellipse; **e, f** color matches stimulus strength; **g, h** color and opacity matches population state and stimulus strength respectively) evolves along trajectories of varying speed (color bars in (**a**) & (**g/h**); apply to (**a**), (**b**), (**g**), (**h**)) as set by initial conditions (**a, g, h**) and/or inputs (**e–g**). In (**g**) and (**h**) light/dark opacity of the arrowhead indicates speed of trajectory as a function of weak/strong stimulus input and initial condition (IC).

## Single unit prestimulus firing rates covary with RT and post-stimulus activity is input dependent

Our database for understanding the neural population dynamics underlying decision-making consists of 996 units (546 units in T and 450 units in O, including both single neurons and multi-units, 801 single neurons) recorded from PMd (Fig. 2g) of the two monkeys over 141 sessions. We included units if they were well separated from noise and if they modulated activity in at least one task epoch. A unit was categorized as a single neuron by a combination of spike sorting and if inter-spike-interval violations were minimal (≤ 1.5% of inter-spike-intervals were ≤ 1.5 ms; median across single neurons: 0.28%).

Fig. 3 shows the smoothed (30 ms Gaussian) trial-averaged firing rates of six example units recorded in PMd aligned to checkerboard

onset and organized either by coherence and choice, plotted until the median RT (Fig. 3a), or organized by RT and choice, plotted until the center of the RT bin (Fig. 3b). Many units showed classical ramp-like firing rates[10,30,35–37] (see Fig. 3, top three rows). However, many neurons demonstrated complex, time-varying patterns of activity that included increases and decreases in firing rate that covaried with coherence, choice and RT[15,18,38,39] (Fig. 3, bottom 3 rows). Additionally, each of the albeit curated neurons in Fig. 3b demonstrated prestimulus firing rate covariation with RT, implying variable initial conditions that ultimately factor into RTs. These firing rate dynamics and those from additional units (Fig. S4a, bottom two rows) were not an artifact of smoothing spike trains with a 30 ms Gaussian kernel, and were near identical even when spike trains were filtered with a 15 ms Gaussian or a causal 50 ms boxcar kernel (Fig. S4b, c).

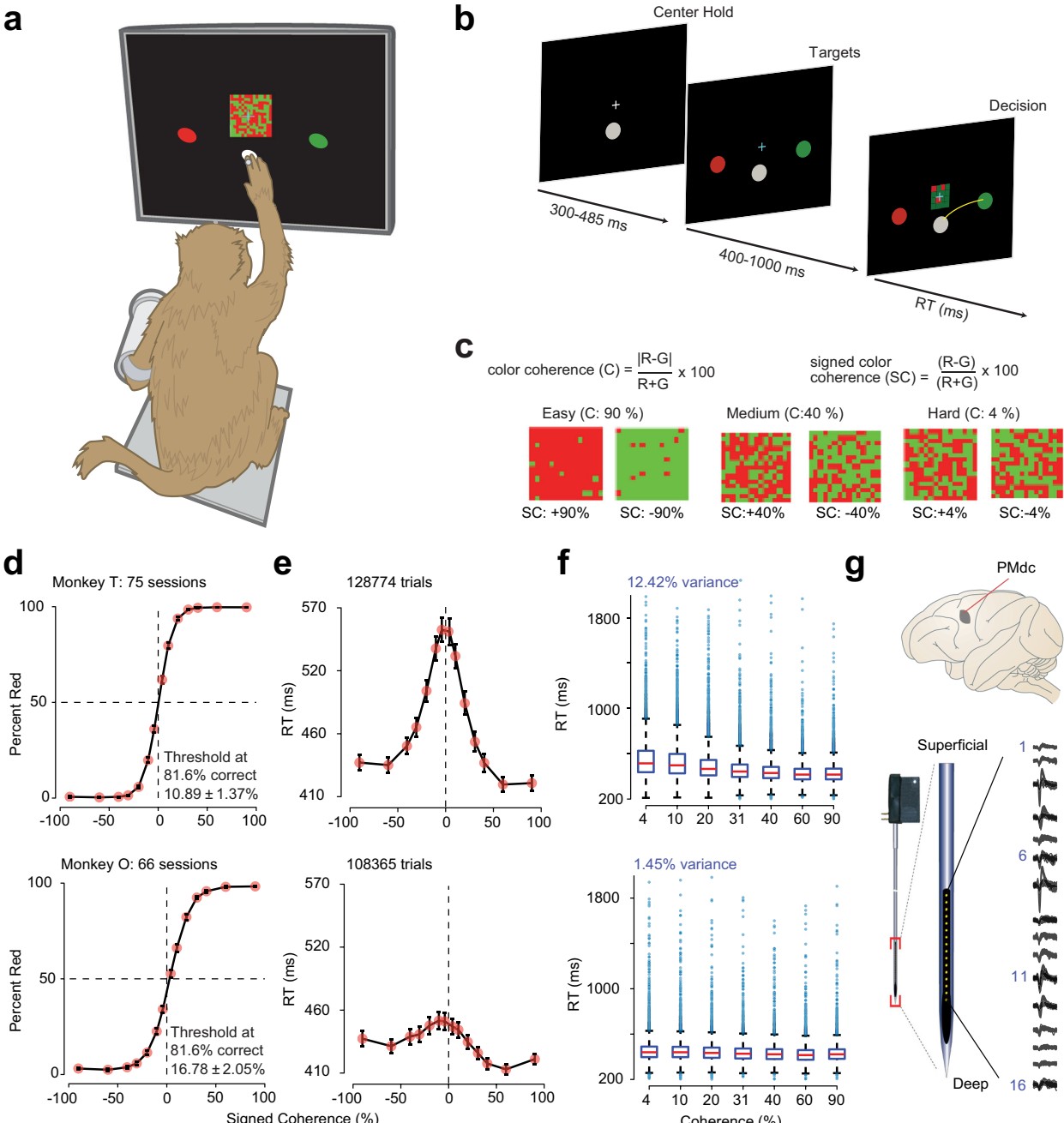

**Fig. 2 | Monkeys can discriminate red-green checkerboards and demonstrate rich variability in RTs between and within stimulus coherences. a** An illustration of the setup for the behavioral task. We loosely restrained the arm the monkey was not using with a plastic tube and cloth sling. A reflective infrared bead was taped on the middle digit of the active hand to measure hand position in 3D space and to mimic a touch screen. Eye position was tracked using an infrared reflective mirror placed in front of the monkey's nose. **b** Timeline of the discrimination task. **c** Examples of different stimuli used in the experiment parameterized by the color coherence of the checkerboard cue. Positive values of signed coherence (*SC*) denote more red (*R*) than green (*G*) squares and vice versa. **d** Psychometric curves, percent responded red, and (**e**) RTs (correct and incorrect trials) as a function of the percent *SC* of the checkerboard cue, over sessions of the two monkeys (T: 75 sessions; O: 66 sessions). Dark orange markers show measured data points along with 2 × SEM estimated over sessions (error bars lie within the marker for many data points). The black line segments are drawn in between these measured data points

to guide the eye. Discrimination thresholds measured as the color coherence level at which the monkey made 81.6% correct choices are also indicated. Thresholds were estimated using a fit based on the cumulative Weibull distribution function. **f** Standard box-and-whisker plots (i.e., center line is median, box limits are upper and lower quartiles, whiskers are 1.5x interquartile range, and outliers are plotted as blue circles) of RT as a function of unsigned checkerboard coherence (RTs from *n* = 128,774 (top) and *n* = 108,365 trials (bottom)). Note large RT variability within and across coherences. **g** The recording location, caudal PMd (PMdc), indicated on a macaque brain, adapted from[89]. Single and multi-units in PMdc were primarily recorded by a 16 electrode (150-*μ*m interelectrode spacing) U-probe (Plexon, Inc., Dallas, TX, United States); example recording depicted. Images in (**a**) and (**g**) are adapted from Chandrasekaran, C., Peixoto, D., Newsome, W.T. et al. Laminar differences in decision-related neural activity in dorsal premotor cortex. Nat Commun 8, 614 (2017). https://doi.org/10.1038/s41467-017-00715-0. Source data are provided as a Source Data file.

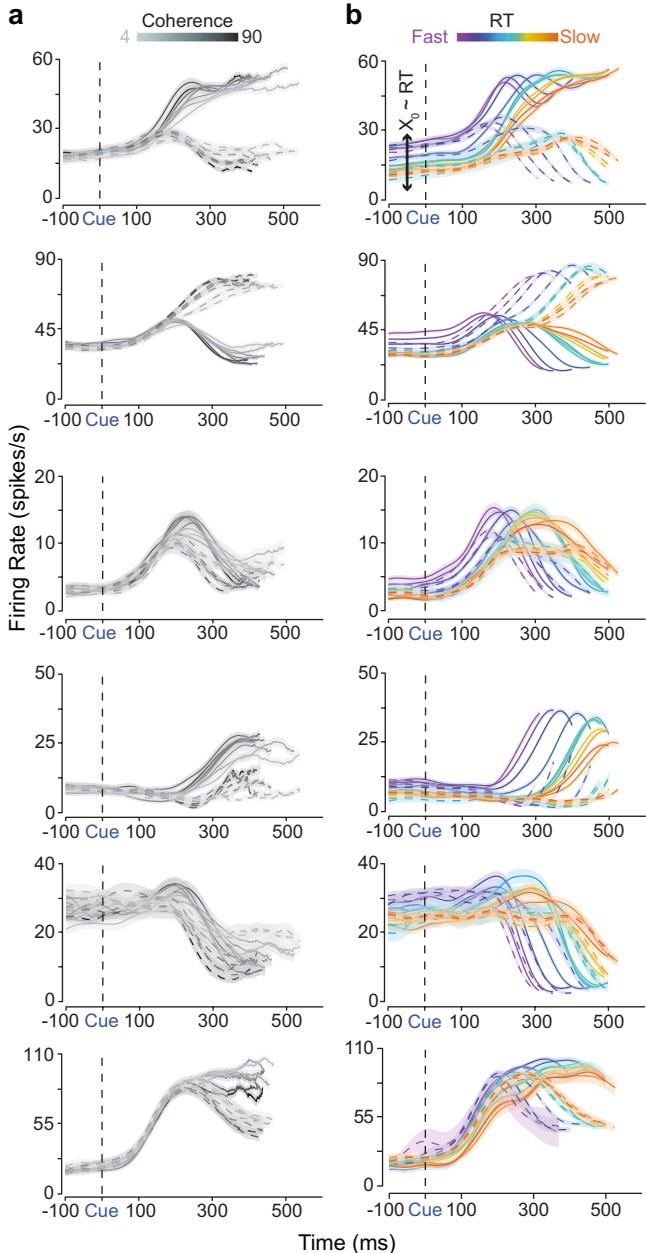

**Fig. 3 | Firing rates of a heterogeneous population of PMd neurons are modulated by the input (i.e., strength of the sensory evidence), and the initial condition (i.e., prestimulus firing rate) covaries with RT.** Firing rate across (**a**) 7 levels of color coherence (grayscale color bar with darker colors indicating easier coherences) and (**b**) 11 RT bins, from fast (violet) to slow (orange), for both action choices (right - dashed, left - solid) of 6 example units in PMd from monkeys T and O aligned to stimulus onset (Cue/vertical dashed black line). Firing rates are plotted until the median RT of each color coherence and until the midpoint of each RT bin (notice slightly different lengths of lines). Color-matched shading is SEM. In (**a**), firing rates tend to separate as a function of coherence, and in (**b**), the same neurons show prestimulus covariation as a function of RT ($X_0$ - RT), and post-stimulus covariation with RT and choice. Source data are provided as a Source Data file.

These examples already suggest that decision-related PMd neural responses are more consistent with some dynamical hypotheses than others. First, when organized by coherence and choice, choice-selective signals are largely absent before checkerboard onset, and latency of choice selective responses after checkerboard onset are only modestly affected by the stimulus coherence. These results are inconsistent with the dynamical hypotheses outlined in Fig. 1e, g. Second, prestimulus correlation with RT is consistent with preliminary

support for the dynamical mechanisms in (Fig. 1g, h), and inconsistent with the hypothesis shown in Fig. 1f.

In the next sections, we use dimensionality reduction, cross-validated single-trial analyses, decoding, and regression methods to understand how RT and choice are represented in the shared, time-varying, and heterogeneous activity of these neurons and reject various dynamical hypotheses. To predict the results of these analyses for various dynamical models shown in Fig. 1e–h we first used two modeling approaches that we describe in the next section.

## Different dynamical mechanisms predict distinct relationships between prestimulus activity with RT and choice

The single unit examples shown in Fig. 3 are consistent with the dynamical mechanisms in Fig. 1g, h in that prestimulus dynamics covary with RT. However, we need to ensure that such effects are also present at the level of the neural population. A common approach to analyze heterogeneous neural populations is to use a dimensionality reduction method such as principal components analysis (PCA) on trial-averaged firing rates organized by various variables of interest and visualize the associated state-space trajectories[15,40]. To derive predictions on how principal components (PCs) from neural data would appear for the various dynamical mechanisms outlined in Fig. 1e–h, we used two complementary approaches.

First, we trained recurrent neural network (RNN) models to perform the same task as our monkeys. RNN models used a ReLu nonlinearity, received noisy evidence for left and right choices, and output two decision-variables for left and right choices (Fig. S1a). Additional details on the RNNs can be found in the methods (section "Recurrent neural network models of various dynamical hypotheses"). Second, we simulated a population of hypothetical neurons, based on our work from[18] that used various metrics to comprehensively characterize the units analyzed in this study. The key observation from that study was that PMd contains a large fraction of neurons that increase their firing rate after stimulus onset, and show strong covariation with RT and choice (i.e., increased neurons). Smaller fractions decreased their firing rate after stimulus onset (i.e., decreased) or were only active around movement onset (i.e., peri-movement). Further details of how these neurons were modeled can be found in the methods (Section "Hypothetical synthetic neural populations") and in Fig. S1b.

After training the RNNs and building the synthetic populations, we performed PCA, decoding, and regression analyses on the firing rates of both types of models (Figs. S2 and S3). Classical decision-making models (Figs. S2a and S3a) and a delayed-input model (Fig. S2b), without any bias for one or another choice as in Fig. 1e, f, show little prestimulus covariation with choice or RT. In contrast, models (Figs. S2c and S3b) with a bias for one of the reaches as in Fig. 1g show a PCA structure where the PCs are biased for one choice over the other. Finally, to simulate the hypothesis shown in Fig. 1h, we used two approaches for the RNN. We biased both left and right choice input before checkerboard onset or altered the gain of the ReLu function. In both RNN cases, we found that prestimulus state covaries with RT. However, in neither of these RNNs (Fig. S2d, e) did we observe a strong and reliable prestimulus covariation with choice. Similarly, in a synthetic neural population where ~20% of neurons (Section "Hypothetical synthetic neural populations") had baseline firing modulation with RT (consistent with the results reported in[18] and the results above), prestimulus population dynamics demonstrated covariation with RT but not choice (Fig. S3c).

These RNN models and synthetic neuron simulations suggest that PCA on trial-averaged neural responses should demonstrate distinct structure consistent with one or another hypothesis. We used these results to evaluate which of these dynamical hypotheses are most consistent with our neural data.

## Principal component analysis reveals prestimulus population state covariation with RT

Informed by our modeling analyses, we next visualized which dynamical hypothesis was most consistent with our PMd data. We initially performed a PCA on trial-averaged firing rate activity (again smoothed with a 30 ms Gaussian) windowed about checkerboard onset, organized by overlapping RT bins (11 levels representing a spectrum from faster to slower RTs; 300–400 ms, 325–425 ms, ..., to 600-1000 ms), and both reach directions (Fig. 4a, b). For this analysis, we pooled all trials, including all stimulus coherences and both correct and incorrect trials, then sorted by and averaged within RT bin and choice. On average, we used 100 to 200 trials per RT bin for these analyses (Fig. S6b).

To identify the number of relevant dimensions for describing this data, we used an approach developed in[41] (see Section "Estimation of number of dimensions to explain the data" for details). Firing rates on every trial in PMd during this task can be thought of as consisting of a combination of signal (i.e., various task related variables) and noise contributions from sources outside the task such as spiking noise for example. Trial averaging reduces this noise but nevertheless when PCA is performed it returns a principal component (PC) space that captures variance in firing rates due to the signal and variance due to residual noise (signal+noise PCA). Ideally, we only want to assess the contributions of the signal to the PCA, but this is not possible for trial-averaged or non-simultaneously recorded data. To circumvent this issue and determine the number of signal associated dimensions, the method developed in[41] estimates the noise contributions by performing a PCA on the difference between single trial estimates of firing rates, to obtain a noise PCA. Components from the signal+noise PCA and the noise PCA were compared component-by-component such that only the signal+noise dimensions that explained significantly more variance than the corresponding noise dimensions were included in further analyses. This analysis yielded six PCs that explained > 90% of the variance in trial-averaged firing rates (Fig. S5a). Similar analyses using different smoothing filters (15 ms Gaussian and 50 ms causal boxcar filters) on spikes to derive firing rates yielded similar numbers of chosen dimensions and similar amounts of variance explained for the first 6 components (87.59% and 85.67%, respectively, Fig. S5b, c).

Fig. 4a plots the first four PCs obtained from this PCA. What is apparent in Fig. 4a is that the prestimulus state strongly covaries with RT but only modestly with choice. In particular, barring component 2, which appears to be most strongly associated with choice, PCs 1, 3, and 4 showed covariation between the prestimulus state and RT (Fig. 4a, highlighted with light blue rectangles)—consistent with the rich covariation between RT and prestimulus firing rates in the single neuron examples shown in Fig. 3b. Visualizing PCs 1, 2, and 4 in a state space plot further supported this observation (Fig. 4b). The axes in Fig. 4b are deliberately not equalized to better highlight prestimulus covariation with RT. A corresponding axis-equalized figure showing the same patterns is shown in Fig. S7.

Such covariation between prestimulus neural state and RT was also not a result of pooling across all the different stimulus difficulties and was even observed within a level of stimulus coherence (note similarities between state space trajectories in Fig. 4b & its inset). We discuss this further in Section "Inputs and initial conditions both contribute to the speed of poststimulus decision-related dynamics" where we analyze the joint effects of inputs and initial conditions. These analyses are also robust to whether they are performed with multi-units and single units[42] (996 units, Fig. 4) as compared to solely well-isolated single neurons (801 single neurons, Fig. S10a), and not dependent on the smoothing used to produce the firing rates (Fig. S11a).

In summary, the PCA trajectories demonstrate a lawful organization with respect to RT and modestly with choice prior to stimulus onset (Fig. 4b). Note such structure was not an artifact of using overlapping RT bins. We observed very much the same structure even when we used non-overlapping RT bins (Fig. 5a). These results are strongly consistent with the dynamical hypotheses proposed in Fig. 1h, weakly consistent with Fig. 1g and inconsistent with hypotheses in Fig. 1e, f. Additionally, direct comparison of these plots to the PCAs of RNNs (Fig. S2d, e) and synthetic neural populations (Fig. S3c) also suggests that perhaps Fig. 1h is overall more consistent with the data than Fig. 1g.

## Position and velocity of initial condition correlate with post-stimulus dynamics and RT

Hypotheses shown in Fig. 1g, h predict that poststimulus dynamics and behavior should demonstrate dependence upon the position and velocity of prestimulus neural trajectories in state space (i.e., initial conditions)[21,22]. Position is the instantaneous location in a high-dimensional state space of neural activity (i.e., firing rate of neurons) and velocity a directional measure of how fast these positions are changing over time (i.e., directional rate of change from one neural state to the next). In Fig. 4b, both the position and the velocity of the prestimulus state appear to covary with RT. For instance, prestimulus trajectories for the fastest RTs are 1) spatially separated, and 2) appear to have covered more distance along the paths for movement initiation by the time of checkerboard onset than the prestimulus trajectories for the slowest RTs (small squares which denote 20 ms time steps are more spread out for faster versus slower trajectories, Fig. 4b). In contrast, only a modest separation by choice occurs before stimulus onset.

We used the Kinematic analysis of Neural Trajectories (KiNeT[26]) approach to more quantitatively test these predictions. KiNeT measures the spatial ordering of trajectories and how each trajectory evolves in time, all with respect to a reference trajectory. For KiNeT analyses we used the first six PCs ( > 90% of variance) as these PCs were significantly different from noise principal components[41]. KiNeT analyses are first performed within a choice and then averaged across choices. Fig. S8 shows a visualization of the KiNeT analyses and Section "Kinematic analysis of neural trajectories (KiNeT)" provides a detailed description of KiNeT calculations.

First, we used KiNeT to assess if position of the initial conditions was related to RT to assess if our data was consistent with hypotheses shown in Fig. 1g, h. If the position of the initial condition covaries with RT then we expect a lawful ordering of neural trajectories organized by RT bin, otherwise they would lie on top of each other indicating a lack of spatial organization. Thus, we examined the spatial ordering of six-dimensional neural trajectories grouped by RT bins for each reach direction. We estimated the signed minimum Euclidean distance at each point for the trajectory relative to a reference trajectory (the middle RT bin, cyan, for that reach direction, Fig. 4c). Trajectories were 1) organized by RT with trajectories for faster and slower RT bins on opposite sides of the the reference trajectory, and 2) the relative ordering of the Euclidean distance with respect to the reference trajectory was also lawfully related to RT (Fig. 4c) as measured by a correlation between the center of each of the 11 RT bins (e.g., $RT_{bin1}$: 325 - 400, $\overline{RT_{bin1}} = 362.5$) and the average signed Euclidean distance from 50 bootstraps, 90 ms before checkerboard onset ($r_9 = -0.85$, $p = 9.33 \times 10^{-4}$). These data are consistent with the dynamical system in Fig. 1g and h that the position of the initial condition correlates with RT.

Second, if the data are consistent with a dynamical system then the relative ordering of trajectories by RT in the prestimulus period should predict the ordering of poststimulus trajectories. We tested if this was the case by measuring the subspace similarity angle, and average alignment (Section "Kinematic analysis of neural trajectories (KiNeT)"). To calculate subspace similarity, representative of the geometry of a subspace, we first estimate vectors between adjacent trajectories at all

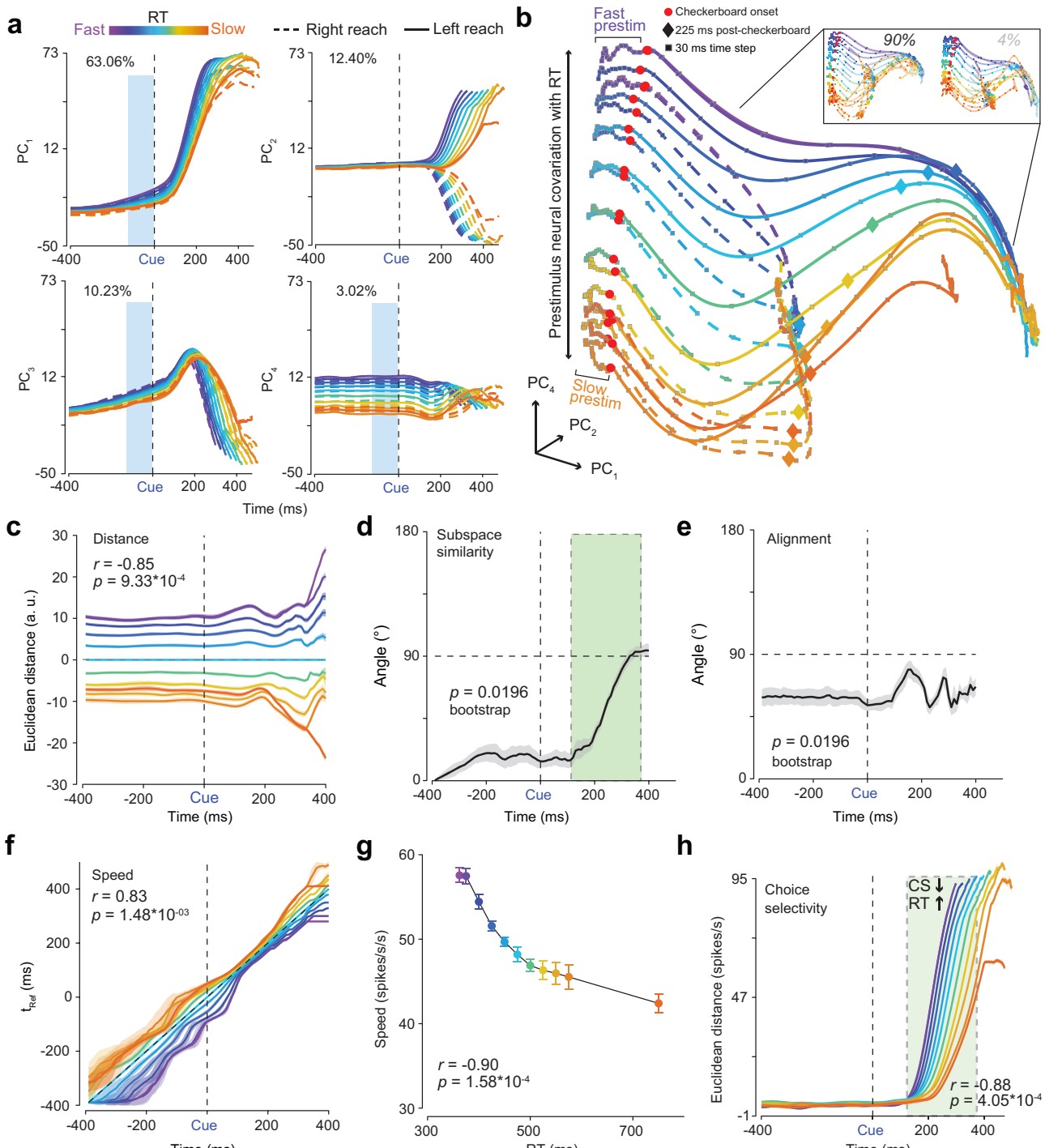

**Fig. 4 | Prestimulus population firing rates covary with RT. a** PCs$_{1-4}$ of trial-averaged firing rates organized across 11 RT bins, both reach directions, and aligned to checkerboard onset. Percent variance explained is provided. Blue boxes highlight prestimulus covariation with RT. **b** State space trajectories of PCs$_{1,2,4}$ aligned to checkerboard onset. Faster RT trajectories appear to move faster in the prestimulus period than slower RTs (fast/slow prestim, also see **g**). *Inset:* State space trajectories within a single stimulus coherence (i.e., 90% and 4%). **c** KiNeT distance analysis showing consistent spatial organization of trajectories peristimulus and correlated with RT. **d** Angle between subspace vector at the first timepoint (-400 ms) and subspace vector at each timepoint is largely consistent but increases as choice signals emerge (green box). **e** Average relative angle between adjacent trajectories at each timepoint was largely less than 90° for the prestimulus period but approach orthogonality as choice signals emerge poststimulus. **f** KiNeT Time to

reference (t$_{Ref}$) analysis shows that trajectories for faster RTs reach similar points on the reference trajectory (cyan, middle trajectory) earlier than trajectories for slower RTs. **g** Average scalar speed for the prestimulus period (-400 to 0 ms epoch) as a function of RT bin. **h** Choice-selectivity signal measured as the Euclidean distance in the first six dimensions between the two reach directions for each RT bin aligned to checkerboard onset. Rate for choice selectivity (CS) is faster for faster RTs compared to slower RTs (green box). In (**c**) and (**f**) the x-axis is time on the reference trajectory. Black dashed lines track the reference trajectory. In **d** & **e** black dashed horizontal line indicates 90°. Error bars are color-matched SEM (n = 50 bootstraps) in (**c**)–(**h**). Correlations in (**c**)–(**h**) were tested with two-sided t tests. p values in (**d**) and (**e**) were derived from one-sided bootstrap tests (n = 50, comparison to 90°). Cue, Checkerboard onset, a. u., Arbitrary units. Source data are provided as a Source Data file.

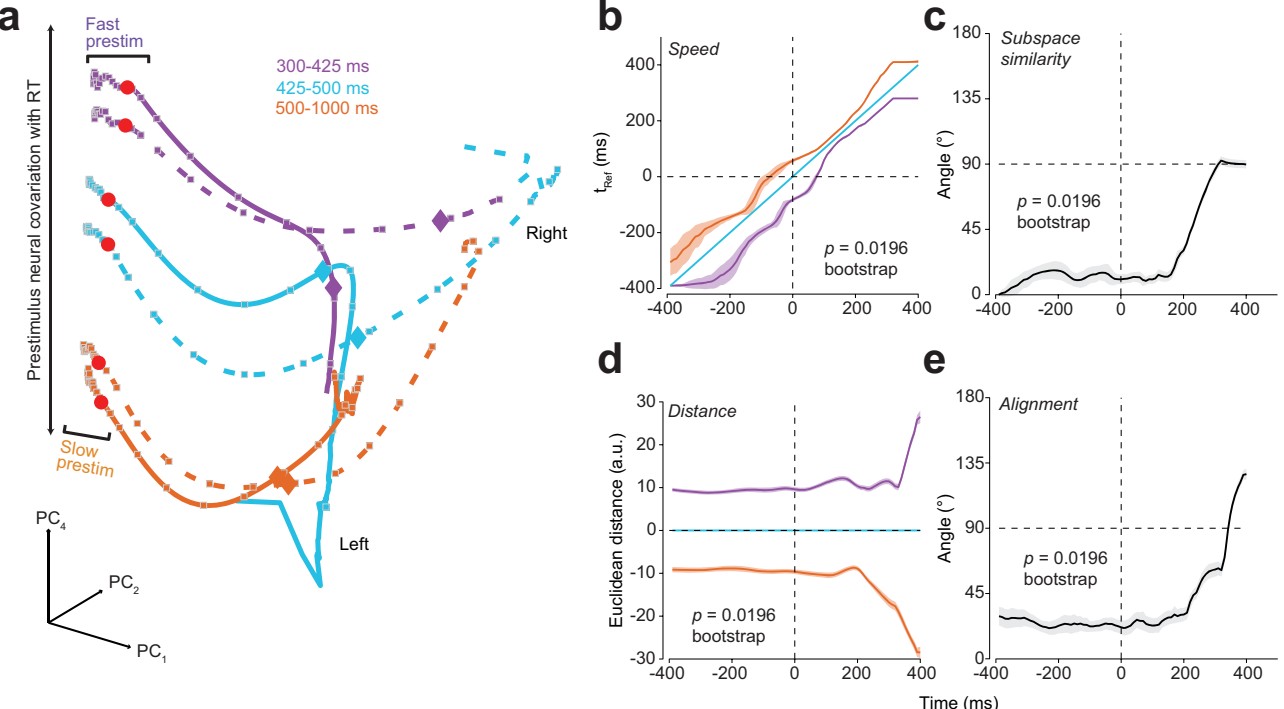

**Fig. 5 | Replication of PCA and KiNeT findings using non-overlapping RT bins.** **a** State space trajectories of the 1st, 2nd and 4th PCs ($PC_{1,2,4}$) aligned to checkerboard onset (red dots). Prestimulus neural activity robustly separates as a function of RT bin. Diamonds and squares, color matched to their respective trajectories, indicate 225 ms post-checkerboard onset and 30 ms time steps respectively. Faster RT trajectories appear to move faster in the prestimulus period than slower RTs (fast/slow prestim). Note axes are deliberately not equalized to better visualize the fluctuations in the initial condition before checkerboard onset. **b** KiNeT Time to reference ($t_{Ref}$, relative time at which a trajectory reaches the closest point in Euclidean space to the reference trajectory) analysis shows that trajectories for faster RTs reach similar points on the reference trajectory (cyan, middle trajectory) earlier than trajectories for slower RTs. This result suggests that the dynamics for faster RTs are closer to a movement initiation state than slower RTs. **c** Angle between subspace vector at each timepoint and subspace vector at the first timepoint (-400 ms). The angle between subspace vectors is largely consistent but the space rotates as choice signals emerge ( ~ 200 ms). **d** KiNeT distance analysis showing that trajectories are consistently spatially organized before and after stimulus onset. **e** Average relative angle between adjacent trajectories for each timepoint. The angles between adjacent trajectories were largely less than 90° for the prestimulus period but are orthogonal post-choice signals ( ~ 350 ms). Error bars are color-matched SEM ($n$ = 50 bootstraps) in (**b**)–(**e**). $p$ values in (**b**)–(**e**) were derived from one-sided 50-repetition bootstrap tests ((**b**) and (**d**) difference between fast and slow RT trajectories different from 0; (**c**) and (**e**) subspace and average relative angle different from 90°). Source data are provided as a Source Data file.

timepoints. These vectors are averaged to derive an average inter-trajectory vector (subspace vector, Fig. 4d) at each time point. We then measure how this average vector rotates over time relative to the subspace vector for the first time point by estimating the angle (subspace angle) between these vectors. Alignment measures the degree to which neural trajectories diverge from one another in state space by estimating the average angle of the normalized vectors between pairwise adjacent trajectories at each timepoint. The null hypothesis is that the ordering of trajectories before stimulus onset is in no way predictive of the ordering of trajectories after stimulus onset. Under this null hypothesis, the subspace angle and alignment would be randomly distributed around 90° poststimulus. Alternatively, if prestimulus dynamics predict ordering of poststimulus dynamics, the average subspace angle and alignment will be largely constant from prestimulus to the poststimulus period until choice and movement initiation signals begin to emerge for the fastest RTs ( ~ 300 ms).

Consistent with the alternative hypothesis, the subspace angle (Fig. 4d) between the first point in the prestimulus period and subsequent timepoints was < 90° before and after checkerboard onset and only increased when movement initiation began to happen for the fastest RTs (50-iteration bootstrap test vs 90°, 90 ms before stimulus onset: $p$ = 0.0196). Similarly, the alignment measured as the angle between adjacent trajectories (Fig. 4e) was largely similar throughout the trial for each reach direction and only begun to change after choice and movement initiation signals began to emerge, suggesting that the

ordering of trajectories by RT was preserved well into the poststimulus period. These results imply that the initial condition strongly predicted poststimulus state and eventual RT (50-iteration bootstrap test vs 90°, 90 ms before stimulus onset: $p$ = 0.0196), again consistent with the predictions of the dynamical systems approach.

Third, we examined if the velocity of the peristimulus dynamics was faster for faster RTs compared to slower RTs. For this purpose, we used KiNeT to find the timepoint at which the position of a trajectory is closest (minimum Euclidean distance) to the reference trajectory, which we call Time to reference ($t_{Ref}$, Fig. 4f). Trajectories slower than the reference trajectory will reach the minimum Euclidean distance relative to the reference trajectory later in time (i.e., longer $t_{Ref}$), whereas trajectories faster than the reference trajectory will reach these positions earlier (i.e., shorter $t_{Ref}$). Given that trajectories are compared relative to a reference trajectory, $t_{Ref}$ can thus be considered an indirect estimate of the velocity of the trajectory at each timepoint. Note, $t_{Ref}$ was referred to as speed in[26]. Although a trajectory could reach the closest point to the reference trajectory later due to a slower speed, it could also be due to unrelated factors such as starting in a position in state space further from movement onset or by taking a more meandering path through state space. All of these effects are consistent with a longer $t_{Ref}$ and a slower velocity, but not necessarily a slower speed.

KiNeT revealed that faster RTs involved faster pre- and post-stimulus dynamics whereas slower RTs involved slower dynamics as

compared to the reference trajectory (trajectory associated with the middle RT bin, cyan) (Fig. 4f). There was also a positive correlation between RT bin center and average $t_{Ref}$ as measured by KiNeT 90 ms before checkerboard onset ($r_9 = 0.83$, $p = 1.48 \times 10^{-3}$, one sided test to 0). Additionally, the overall scalar speed of trajectories in the prestimulus state for the first six dimensions (measured as a change in Euclidean distance over time and averaged over the 400 ms prestimulus period) covaried lawfully with RT ($r_9 = -0.90$, $p = 1.58 \times 10^{-4}$; Fig. 4g). Thus, the velocity of the initial condition, relative to the reference trajectory, is faster for faster RTs compared to slower RTs, coherent with the prediction of the initial condition hypothesis[20].

One concern is that perhaps these correlations are difficult to interpret because they use overlapping RT bins. We also repeated the KiNeT analysis for nonoverlapping bins and found exactly the same pattern of results (Fig. 5b–e). In addition, all KiNeT results were replicated even if we only 1) used single units for our analyses (Fig. S10b–e), or used different smoothing kernels (15 ms Gaussian or a 50 ms boxcar, Fig. S11B–E). Collectively, these results firmly establish that the initial condition in PMd correlates with RT and that the geometry and trial-averaged dynamics of these decision-related trajectories strongly depend on the position and velocity of the initial condition consistent with the hypotheses shown in Fig. 1g, h and inconsistent with the hypotheses shown in Fig. 1e, f.

### Cross-validated single-trial analyses corroborate PCA results that prestimulus neural activity predicts future neural activity and RT

Our PCA and KiNeT analyses on trial-averaged data (Fig. 4) strongly support the dynamical mechanisms in Fig. 1g, h that prestimulus state correlates with RT and suggest that prestimulus neural activity predicts future neural activity (Fig. 4d, e). However, it is unclear whether this is simply an effect of trial averaging, or whether such effects would also be seen at the single-trial level. In this section, we use Tensor component analysis (TCA), fits to a linear dynamical system (LDS), fits of a nonlinear dynamical system (LFADS), and reduced-rank regression to confirm that prestimulus state is predictive of future neural activity and RT at the single-trial level.

We applied TCA[43], a matrix factorization technique akin to PCA, to binned (50 ms) spiking activity from 600 ms before to 600 ms after checkerboard onset for all trials. TCA was performed for each of the 44 sessions (23 from monkey T, and 21 from monkey O) containing V-probe data (~2–32 units) and returns three connected low-dimensional descriptions of neural activity, or tensors: neuron factors (N) × temporal factors (T) × trial factors (K) (Fig. S12a, b; Section "Tensor component analysis"). Regardless of cross-validation method, either speckled holdout[43], or with a more conservative neuron hold-out, the variance of neural activity explained in both the training and the test sets increased as the rank of the low-dimensional model increased (Fig. S12d). Thus, TCA provides a reasonable description of neural activity especially considering that this was performed with small numbers of units.

We next visualized the low-dimensional activity profiles for fast and slow RT trials from a single session (Fig. S12c) by multiplying the temporal factors and the trial-specific factors (Fig. S12a, b). Consistent with our results from PCA, we found that prestimulus neural activity indeed separated by RT (Fig. S12c). A regression analysis suggested that >25% of the variance in RT was explained by the rank 4 low-dimensional descriptions in the prestimulus period (Fig. S12e). Therefore, this TCA is the first line of single-trial evidence that prestimulus neural activity in PMd correlates with RT and is consistent with hypotheses shown in Fig. 1g, h.

The core thesis of this study is that neural activity in PMd is well described by a dynamical system and that the initial condition (prestimulus neural activity) is strongly predictive of future neural activity and RT. The KiNeT subspace similarity and alignment analyses

(Fig. 4d, e) provide indirect evidence that PMd activity is consistent with a dynamical system and that initial conditions are predictive of future neural activity and RT.

As a more direct test, we fit a simple low-dimensional autonomous dynamical system to binned single-trial firing rates (50 ms bins). We fit separate dynamical systems to the pre- and post-stimulus period of sessions with at least 10 units (31 sessions), fit left and right choices separately, and used leave-one-out cross validation to assess the model as a function of the dimensionality of the dynamical system (Fig. S13). We found that firing rates at the current time point closely predicted the firing rates 100 ms later (Fig. S13a). Furthermore, the LDSes were excellent models of the pre- and post-stimulus neural dynamics and could describe neural data on held-out trials, with increasing model size improving the fit of the model to the data (Fig. S13b). We then estimated the firing rates predicted by the LDS at each time point and investigated if pre- and poststimulus firing rates predicted RT. Again, prestimulus firing rates strongly predicted RT, with prediction accuracy improving for dynamical systems of larger dimensionality (Fig. S13c).

We also replicated these results using a nonlinear dynamical system fit, the Latent Factor Analysis of Dynamical Systems (LFADS[44], described further in Section "Latent Factors Analysis of Dynamical Systems (LFADS)"). Again, we could predict firing rates of neurons on trials held out from the fitting process reliably (Fig. S14a, b), and pre- and poststimulus activity was correlated with RT (Fig. S14c). Thus, these two analyses strongly demonstrate that neural activity in PMd in this task is well modeled by a dynamical system where prestimulus activity can be used to predict future neural activity and RT.

Finally, we used an alternative cross-validated reduced-rank regression[45,46] to test if prestimulus neural activity predicted future neural activity (Fig. S15a). Consistent with our KiNeT analysis, the prestimulus state better predicted neural activity than a shuffle control in both the pre- and post-stimulus epochs in a single session and averaged across multiple sessions (Fig. S15b, c). Additionally, angles between reduced-rank regression beta values were measured from the first timepoint to all other timepoints and found to be stable from the pre- to post-stimulus period until the point where choice signals emerge, especially as compared to beta values from shuffled activity (Fig. S15d). This single-trial analysis therefore corroborates the subspace similarity and alignment analyses performed with the trial-averaged data.

All together, these four cross-validated single-trial analyses again show that neural activity in PMd is consistent with a dynamical system where prestimulus activity predicts future neural activity and RT (Fig. 1g, h).

### Initial conditions do not predict eventual choice

The previous analyses demonstrated that initial conditions strongly covaried with RT consistent with the hypotheses shown in Fig. 1g, h. However, does the initial condition also predict choice? If it does not, then the data rules out the dynamical hypothesis in Fig. 1g at least for PMd in this task. To investigate this issue, we first examined the covariation between prestimulus and poststimulus state with choice by measuring a choice-selectivity signal identified as the Euclidean distance between the left and right choices in the first six dimensions at each timepoint. The choice-selectivity signal was largely flat during the prestimulus period and increased only after stimulus onset (Fig. 4h). We also found that slower RT trials had delayed and slower increases in the choice-selectivity signal compared to the faster RTs, a result consistent with the slower overall dynamics for slower compared to faster RTs (Fig. 4h). Consistent with this observation, we found a negative correlation between the average choice-selectivity signal in the 125 to 375 ms period after checkerboard onset and the center of the RT bin (mean and 95% CI, $r_9 = -0.88$, $p = 4.05 \times 10^{-4}$).

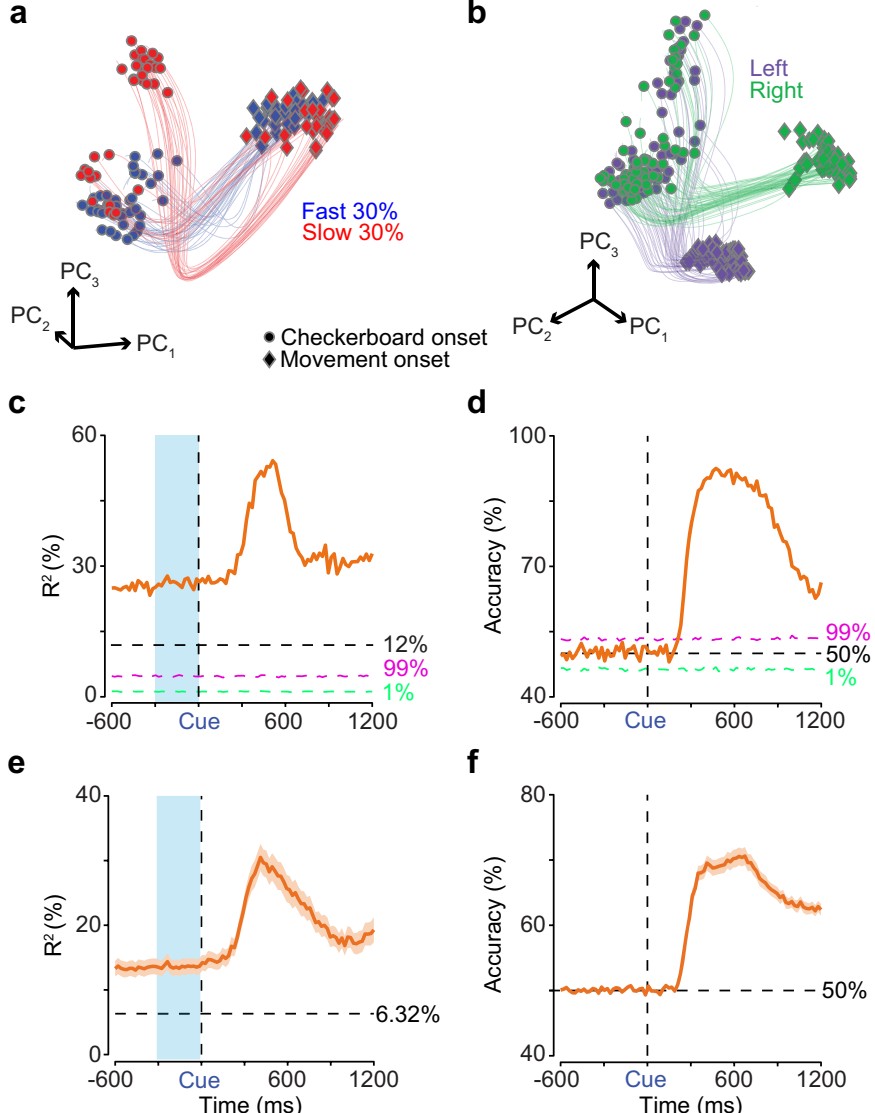

**Fig. 6 | Single-trial analysis, linear regression, and decoders reveal that initial conditions predict RT but not choice.** LFADS[44] trajectories in the space of the first three orthogonalized factors (PC$_{1,2,3}$), obtained via PCA on LFADS latents, plotted for (**a**) the fastest 30% of trials (blue) and the slowest 30% of trials (red) for left reaches and (**b**) for left (purple) and right (green) reaches, all for the easiest coherence from a single session (23 units). Each trajectory is plotted from 200 ms before checkerboard onset (dots) to movement onset (diamonds). **c**, **d** Variance explained ($R^2$)/decoding accuracy from linear/logistic regressions of binned spiking activity (20 ms) and coherence to predict trial-matched RTs/eventual choice from all 23 units in the LFADS session shown in (**a** and **b**). Horizontal black dotted lines

are the variance explained by a regression using stimulus coherence to predict RTs (12%) and 50% accuracy respectively. The magenta and light green dotted lines are the 99th and 1st percentiles of $R^2$/accuracy values calculated from an analysis of trial-shuffled spiking activity (500 repetitions) and RTs/choice. **e/f** $R^2$/accuracy values, calculated as in (**c**) and (**d**), averaged across 51 sessions. Orange shaded area is SEM. **e** 6.32% is the average percentage of variance explained across the 51 sessions for both monkeys by regressions using stimulus coherence to predict RTs. The horizontal black dotted line in (**f**) denotes 50% accuracy. Blue highlight boxes in (**c**) and (**e**) denote prestimulus neural covariability with RT. Cue - Checkerboard onset. Source data are provided as a Source Data file.

To further discriminate between the hypotheses shown in Fig. 1g, h, we further explored the initial condition and subsequent poststimulus dynamics using a combination of single-trial analysis, decoding, and regression. We first used the cross-validated LFADS[44] approach to estimate single-trial dynamics in an orthogonalized latent space for left reaches and the easiest coherence in a single session (23 units). This analysis revealed that: 1) initial state for a majority of the slow RT trials are separated from the fast RT trials, 2) initial conditions associated with a minority of the slow trials are mixed in with fast initial conditions, and 3) slower RT trajectories also appear to have more curved trajectories (Fig. 6a). All of these are consistent with the results of the trial-averaged PCA reported in Fig. 4 and mirror single-trial results from TCA (Fig. S12c). Most importantly, initial neural states related to left and right reach directions are mixed prior to stimulus onset (Fig. 6b)–again

consistent with the results of the trial-averaged PCA. These single-trial dynamics suggest that prestimulus spiking activity covaries with RTs but not choice, even on single trials.

Regression and decoding analyses of raw firing rates supported insights from the LFADS visualization (Fig. 6a) that prestimulus spiking activity would be predictive of RT. A linear regression with binned prestimulus spiking activity (20 ms causal nonoverlapping bins) and coherence as predictors explained ~25% of the variance in RT from the same session used for LFADS (Fig. 6c), significantly higher than the 99th percentile of variance explained by a similar regression using trial-shuffled spiking activity instead. Identical linear regressions were performed for each of 51 sessions and $R^2$ values were averaged across sessions. Across these sessions (Fig. 6e), prestimulus spiking activity and

coherence again explained significantly more RT variance than a shuffle control of spiking activity for 47 out of 51 sessions (Mean ± SD: 13.50 ± 8.57%, 4.70 ± 3.61%, one-tailed binomial test, $p = 1.11 \times 10^{-10}$, Fig. S16a).

Note, prediction of RT by spiking activity was not an artifact of RT covarying with the coherence. For instance, neural activity and coherence combined explain ~25% of the variance in RTs for the example session shown in Fig. 6a, b, but coherence alone only explains only 12% of the variance in RT (Fig. 6c). Similarly, on average across sessions, linear regression with binned spiking activity and coherence as predictors explained significantly more variance in RTs in all prestimulus bins than a linear regression of RTs with solely coherence as the predictor (only the last prestimulus bin is reported here: Mean ± SD: 13.66 ± 8.9%, 6.32 ± 5.97%; Wilcoxon rank sum comparing median $R^2$, $p = 2.97 \times 10^{-9}$, Fig. 6e). Therefore, nearly equal amounts of RT variance are explained by prestimulus neural spiking activity (~7%) and the coherence of the eventual stimulus (6.32%, Fig. 6e). These decoding results are essentially a replication of the results shown in Figs. S12e, S13c, and S14c.

In contrast, a logistic regression using binned spiking activity failed to predict choice at greater than chance levels in the prestimulus period. The choice-decoding accuracy was not significantly greater than the 99th percentile of accuracy from a logistic regression using trial-shuffled spiking activity, until after stimulus presentation (Fig. 6d). Similar logistic regressions were built for each session and accuracy was averaged across bins and sessions. The average prestimulus accuracy for predicting choice (Fig. 6f) was no better than chance or than the 99th percentile of averaged prestimulus accuracy from similar logistic regressions built on trial-shuffled spiking activity (Mean ± SD: 50.08 ± 0.51%, 50.00 ± 0.03%, only one session was larger than the shuffled data out of 51 comparisons, one-tailed binomial test, $p = 0.999$, Fig. S16b).

We further explored whether there was a prestimulus bias for faster RT bins (apparent larger prestimulus separation by choice for faster RT bins, Fig. 4b) or harder coherences as prestimulus activity has been found to be predictive of choice for harder coherences in previous experiments[35]. For one, prestimulus spiking activity was no better than chance at predicting eventual choice even when trials were grouped by RT bins (Fig. S16c). Next, we further refined this analysis by performing a decoding analysis where we restricted the analysis to just the hardest coherence and the fastest and slowest RT bins (Fig. S17b). Again we found no relationship between prestimulus neural activity and choice for any of the RT bins. Results were similar even when we restricted the trials to just the easiest coherence (Fig. S17a). Second, we also performed a simple regression analysis (50 ms causal bins stepped by 1 ms) where we examined if neural activity covaried with choice on a neuron-by-neuron basis for just the fastest and slowest RTs for the hardest coherences, and compared it to the percent of neurons that covaried with RT (Fig. S17c). We found that percent of neurons that covaried with choice before stimulus onset was largely at chance levels, whereas a modest (~5%) but significant portion of neurons correlated with RT even before stimulus onset. Including all coherences in this regression also did not change the results—again ~20% of neurons covary with RT before stimulus onset but only ~1% of units covary with choice (Fig. S17d). Thus, we found further evidence of the neural population covarying with RT before stimulus onset but did not observe any prestimulus choice bias.

These results are a key line of evidence in support of the dynamical hypothesis outlined in Fig. 1h that initial conditions covary with RT but not choice and thus help reject Fig. 1g as a candidate model for our neural data. They also provide independent validation of the results from the analysis of the PC trajectories.

## Inputs and initial conditions both contribute to the speed of poststimulus decision-related dynamics

Thus far we have shown that the initial conditions predict RT but not choice. Our monkeys clearly demonstrate choice behavior that depends on the sensory evidence, and also are generally slower for harder compared to easier checkerboards. These behavioral results and the dynamical systems approach make two key predictions: 1) sensory evidence (i.e. the input), should modulate the properties of the choice-selectivity signal after stimulus onset and 2) the overall dynamics of the choice-selectivity signal should depend on both sensory evidence and initial conditions (Fig. S9).

To test the first prediction, we performed two analyses. First, we performed a PCA on firing rates of PMd neurons organized by stimulus coherence and choice. Figure 7a shows the state space trajectories for the first three components. In this space, activity separates faster for easier compared to harder coherences. Consistent with this visualization, choice selectivity increases faster for easier compared to harder coherences (Fig. 7b). However, there is little to no change in the latency of choice selectivity as a function of the stimulus coherence and thus firing rates in PMd are inconsistent with the hypothesis that there are stimulus-dependent encoding delays (Fig. 1f). These results suggest that poststimulus dynamics are at least in part controlled by the sensory input, consistent with the predictions of the dynamical systems hypothesis.

To test the second prediction of how sensory evidence and initial conditions jointly impact the speed of poststimulus dynamics, we performed a PCA of PMd firing rates conditioned on RT and choice within a coherence. To obtain these trajectories, we first calculated trial-averaged firing rates for the 11 RT bins within each coherence. We then projected these firing rates into the first six dimensions of the PC space organized by choice and RTs (Fig. 4a, b). This projection preserved more than 90% of the variance captured by the first six dimensions of the data organized by RT bins and choice within a coherence. Typically the first six dimensions explained 75% of the total variance of the data for a given coherence. Consistent with the results in Fig. 4b, the prestimulus state again covaries with RT even within a stimulus difficulty (Fig. 7c).

To assess how inputs and initial conditions jointly influenced decision-related dynamics, we again computed the time-varying choice-selectivity signal (CS(t)) by computing the high-dimensional distance between left and right trajectories at each timepoint for each of the RT bins and coherences. Fig. 7d shows this choice-selectivity signal as a function of RT bin for the three different coherences shown in Fig. 7c. For the easiest coherence, the choice-selectivity signal starts ~100 ms after checkerboard onset and it increases faster (i.e., steeper slope) for faster RTs compared to slower RTs (Fig. 7d, left panel, blue highlight box). In contrast, for the hardest coherence, the choice-selectivity signal is more delayed for the slower RTs compared to the faster RTs, while the slope effect is much weaker (i.e., slope for fast RTs is similar to that of slow RTs, Fig. 7d, right panel). These plots suggest that inputs and initial conditions combine and alter the rate and latency of choice-related dynamics.

We quantified these patterns by first measuring the average choice-selectivity signal in the 250 ms period from 125 to 375 ms after checkerboard onset as a function of the initial condition and for each of the 7 coherences. We obtained an estimate of the initial condition by using a PCA to project the average six-dimensional location in state space in the -400 ms to -100 ms period before checkerboard onset for each of these conditions on to a one-dimensional axis (see "Initial condition as a function of RT and coherence"). As Fig. 7e shows, the average choice-selectivity signal is larger for easier coherences across the board but also weaker or stronger depending on the initial condition. Furthermore, when coherence is fixed, the average choice-selectivity signals depends on the initial condition. A partial correlation analysis found

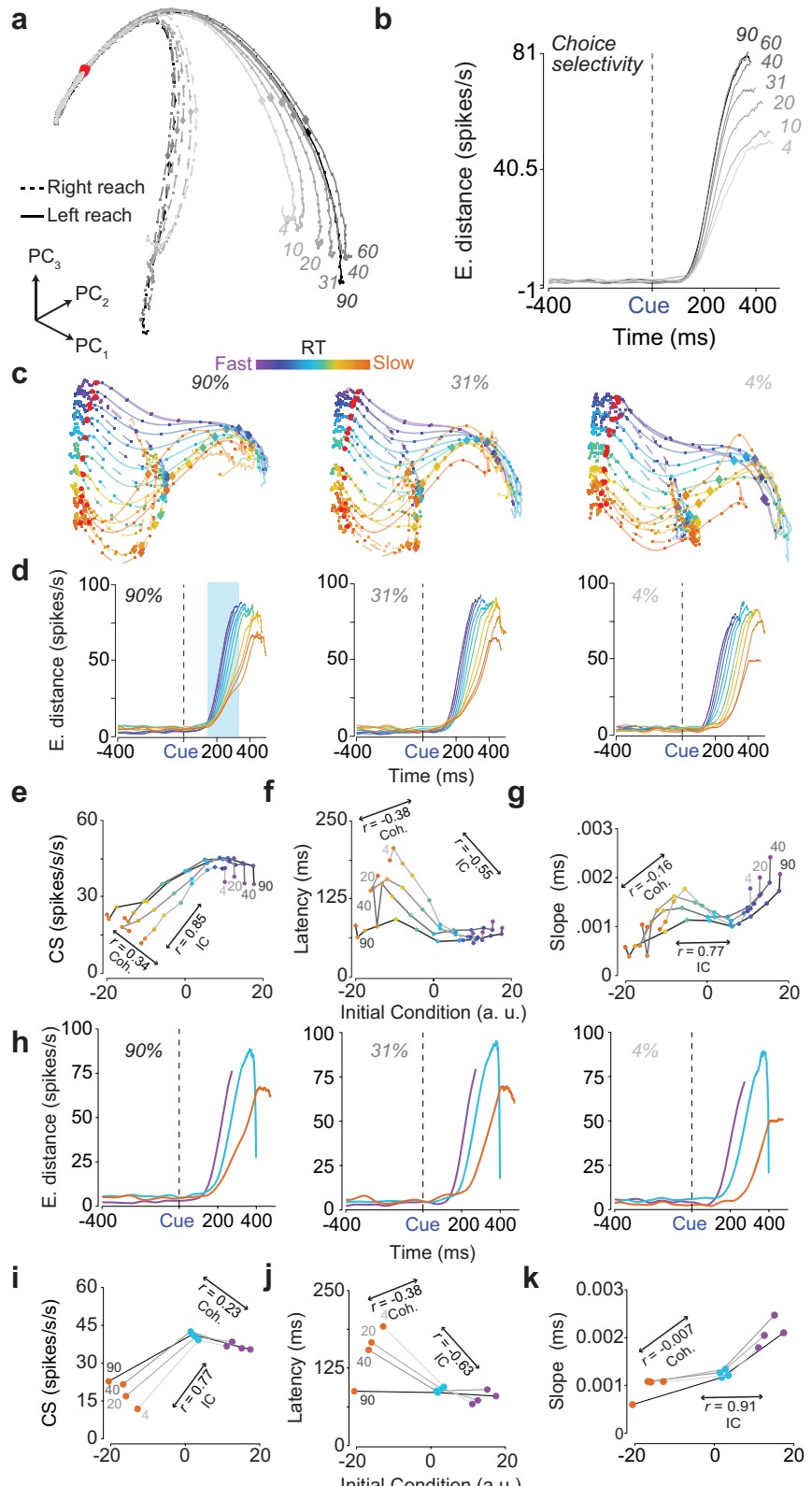

that the average choice selectivity in this time epoch depends on both the initial condition (50 bootstraps, mean and 99% confidence interval, $r_{74} = 0.85$ (0.846 - 0.854), $p = 0.0196$) and the sensory evidence ($r_{74} = 0.341$ (0.331 - 0.350), $p = 0.0196$). These results are key evidence that choice-selective, decision-related dynamics are controlled both by the initial condition and the sensory evidence.

Do effects observed in Fig. 7 emerge from slope changes, latency changes or both? To address this question, we fit the choice-selectivity signal (CS(t)) using a piecewise function (eq. 21) with a latency and slope parameter. Fig. 7f plots the latency of the choice-selectivity signal ($t_{Latency}$) as a function of the sensory input and the initial condition. Latencies depend on both the initial condition and sensory evidence.

**Fig. 7 | Initial conditions and inputs determine speed of dynamics and ultimately choice and RT behavior. a** First three PCs (PC$_{1,2,3}$) of firing rates aligned to checkerboard onset (red dots) and conditioned on stimulus coherence and choice. We observed strong poststimulus separation as a function of choice and coherence, but no observable prestimulus (-400 ms to 0 ms) separation. Color-matched diamonds and squares, indicate 225 ms post-checkerboard onset and 20 ms time steps respectively. **b** Choice-selectivity signal measured as the 6-dimensional Euclidean distance (E. distance) between left and right reaches as a function of stimulus coherence. **c** State space trajectories of the 1st, 2nd and 4th PCs of PCAs conditioned on RT bins and action choice within three stimulus coherences (90%, 31%, & 4%). **d** Choice-selectivity signal for each of the three coherences shown in **c** as a function of RT bin. **e** Average choice-selectivity (CS) signal in the 125 to 375 ms after checkerboard onset as a function of the initial condition (IC) within each coherence (Coh.). Easier coherences lead to higher choice-selectivity signals regardless of initial condition, but the magnitude of this signal depends on the initial condition as well as sensory evidence. **f** Latency of the choice-selectivity signal as a function of the initial condition (IC) and for each stimulus coherence (Coh.). As expected from (**d**), the latency is largely flat for the easier coherences and faster RT bins (regardless of coherence), but slower for the harder coherences. The bend down in **e** and the bend up in (**f/g**) for the fastest RT bin (300-400 ms, violet) is likely an artifact of fewer trials in this bin. **g** Slope of the choice-selectivity signal (*m* term in eq. 21) as a function of the initial condition (IC) and coherence (Coh.). Slopes strongly depend on initial condition but only weakly on coherence. **h** Same as (**d**), but for non-overlapping bins. Notice similarity between (**h**) and (**d**). Bend down in (**h**) emerges due to truncation of firing rates at movement onset when averaging. **i–k** Same as (**e**)–(**g**) but for non-overlapping bins. Again the choice selectivity depends on both initial condition (IC) and coherence (Coh.). Cue - Checkerboard onset. Source data are provided as a Source Data file.

Latencies are slowest when the initial condition is in the slow RT state and for weak inputs but faster for strong inputs or when the initial condition is in a fast RT state. Consistent with this joint dependence, a partial correlation analysis found that the latency of choice selectivity depends on both the initial condition ($r_{74} = -0.55$ (-0.59, -0.51), $p = 0.0196$) and stimulus coherence ($r_{74} = -0.38$ (−0.4, −0.36), $p = 0.0196$).

Figure 7g plots the slope of the choice-selectivity signal (*m*) as a function of the sensory input and the initial condition. In contrast to latency, slope of the choice-selectivity signal was strongly dependent on the initial condition but only weakly modulated by coherence. A partial correlation analysis confirmed these observations. Slope was strongly correlated with initial condition ($r_{74} = 0.77$ (0.75, 0.79), $p = 0.0196$) but had a very weak relationship to sensory evidence ($r_{74} = -0.16$ (−0.18, −0.13), $p = 0.0196$).

Again, these effects were reaffirmed when using non-overlapping bins for the analyses. Figure 7h shows the choice-selectivity signals for the three non-overlapping bins considered in Fig. 5a. For instance, for the easiest coherence, activity increases faster for faster RTs compared to slower RTs but does not appear to do so for the harder coherences, consistent with the patterns observed in Fig. 7d-g. We computed the average choice-selectivity for each of these three RT bins in the 125 to 375 ms period and again found both initial condition and coherence had an impact on the average choice-selectivity in the 125 to 375 ms period (mean and 99% confidence intervals, coherence: $r = 0.225$ (0.204, 0.241), initial conditions: $r = 0.77$ (0.76, 0.78), $p = 0.0196$ for both cases, 50 bootstraps, Fig. 7i). Subsequent analysis of the latency and slope of these choice-selectivity signals (Fig. 7j, k) were also consistent with the conclusions from overlapping bins. Latency was strongly impacted by initial condition ($r_{18} = -0.63$, (−0.72, −0.53)) and modestly by coherence ($r_{18} = -0.38$ (−0.43, −0.33)). Slope was again strongly influenced by the initial condition ($r_{18} = 0.91$ (0.88, 0.93), $p = 0.0196$) but had almost no relationship to coherence ($r_{18}$=-0.007, (-0.099, 0.085), $p = 0.46$).

In summary, choice selectivity depends on both the initial conditions and the inputs. The initial condition exerts a strong independent effect on both slope and latency of this signal, whereas sensory evidence interacts with the initial condition in altering the latency of the signal. Collectively, these results strongly support a dynamical system for decision-making where both initial conditions and inputs together shape decision-related dynamics and behavior.

**The outcome of the previous trial influences the initial condition**
So far we have demonstrated that the initial condition, as estimated by prestimulus population spiking activity, explains RT variability and poststimulus dynamics in a decision-making task. However, why initial conditions fluctuate remains unclear. One potential source of prestimulus neural variation could be post-outcome adjustment, where RTs for trials following an error are typically slower or occasionally faster than RTs in trials following a correct response[32,47,48].

We examined if post-outcome adjustment was present in the behavior of our monkeys. We identified all error, correct (EC) sequences and compared them to correct, correct (CC) sequences. The majority of the data are from sequences of the form CCEC (78%). We compared any remaining EC sequence to the nearest CC sequence, either before or after the EC sequence (22%). We did not observe any error streaks (Fig. S18a). Associated RTs were aggregated across both monkeys and sessions.

We found that correct trials following an error were significantly slower than correct trials following a correct trial (Mean ± SD: 487 ± 129 ms, 446 ± 96 ms; Wilcoxon rank sum comparing median RTs, $p = 2.23 \times 10^{-308}$, Fig. S18b). Additionally, we found that correct trials following a correct trial were modestly faster than the correct trial that preceded it (M ± SD: 446 ± 96 ms, 451 ± 105 ms; Wilcoxon rank sum comparing median RTs, $p = 1.81 \times 10^{-4}$, Fig. S18b). Thus, trials where the previous outcome was a correct response led to a trial with a faster RT, whereas trials where the previous outcome was an error led to a trial with a slower RT.

Such changes in RT after a previous trial were mirrored by corresponding shifts in initial conditions. A PCA of trial-averaged firing rates organized by trial outcome and choice revealed that prestimulus population firing rate covaried with the previous trial's outcome. Post-error correct trials, hereafter post-error trials, showed the largest prestimulus difference in firing rates as compared to other trial outcomes (Fig. 8a, b).

A KiNeT[26] analysis using the first 6 PCs (again 6 dimensions captured ~90% of the variance; Fig. S18c) further corroborated prestimulus firing rate covariation with the previous trial's outcome. Peristimulus trajectories for post-error trials occupied the reflected side of state space, relative to the reference trajectory (Correct trials), as compared to all other trial types (Fig. 8c). The averaged, windowed (i.e., −400:−200 ms, −200:0 ms, 0:200 ms, 200:400 ms) post-error trajectory was significantly different from the distance reference trajectory (0) for all windows (50-repetition bootstrap test, $p < 0.0196$; Fig. 8c). In a similar finding, a decoder revealed that the current trial's spiking activity can predict, at greater than chance levels, the previous trial's outcome from before stimulus onset until about the overall mean RT, ~450 ms (equal numbers of correct and error trials, were used in training the decoder, Fig. 8d), suggesting that the previous trial's outcome has an effect on the current trial's pre- and post-stimulus population firing rates.

KiNeT analyses reveal that post-error trials also had significantly slower prestimulus trajectories as compared to the speed reference trajectory for both prestimulus windows (i.e., −400:−200 ms & −200:0 ms; 50-repetition bootstrap test vs reference trajectory: $p = 0.0196$; Fig. 8e), suggesting that error trials or, similarly, infrequent outcomes[47] result in slower population dynamics in the following trial. Additionally, trials that follow correct trials (errors generally followed correct trials) have slightly faster prestimulus dynamics as compared to the

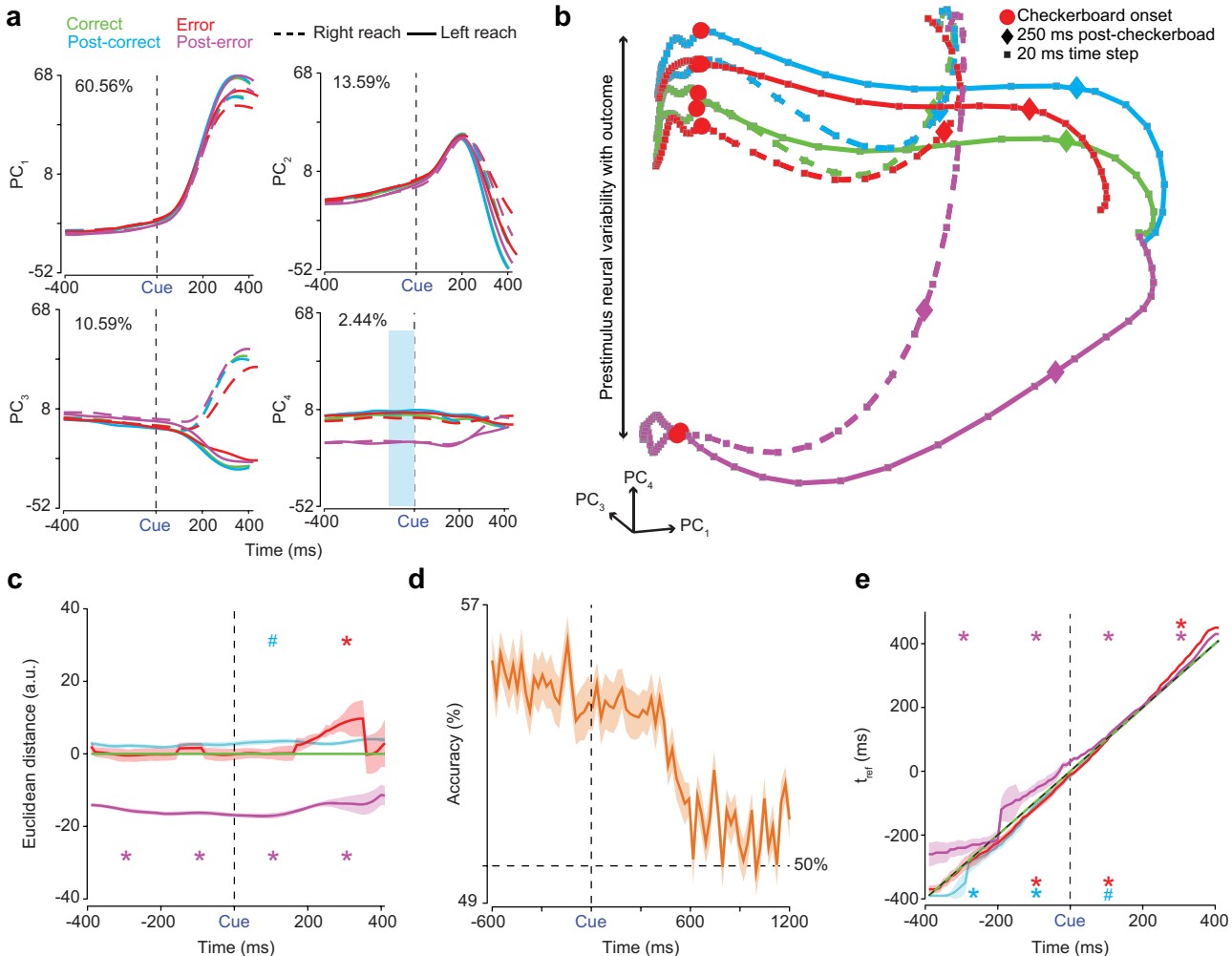

**Fig. 8 | Prestimulus neural activity covaries with the previous trial's outcome.**
**a** The first four PCs (PC$_{1,2,3,4}$) of trial averaged firing rates aligned to checkerboard onset (Cue & black dashed line) of all 996 neurons from monkeys T & O and all sessions organized by choice (right - dashed lines, left - solid lines) and trial outcome (green - correct trial, cyan - correct trial following a correct trial, red - error trial, and magenta - correct trial following an error trial). Percentage variance explained by each PC presented at the top of each plot. **b** 1st, 3rd and 4th PC (PC$_{1,3,4}$) state space aligned to checkerboard onset (red dots). Plotting of PCs extends 400 ms before checkerboard onset and 400 ms after. Observe how neural activity separates as a function of outcome, but not by choice, up to 400 ms before stimulus onset. Squares and diamonds, color matched to their respective trajectories, indicate 20 ms time steps and 250 ms post-checkerboard onset respectively. **c** KiNeT distance analysis demonstrating that trajectories are spatially organized with post-error trials furthest from other trial types peri-stimulus as compared to a reference trajectory (green, middle trajectory). **d** Logistic regressions were built per session (51 total) from current trial spiking activity to predict the outcome of the previous trial. Plot shows the average percent accuracy of these 51 logistic regressions peristimulus. Orange outline is SEM. **e** KiNeT Time to reference (t$_{Ref}$) analysis reveals that prestimulus velocity is slower peri-stimulus for post-error trials as compared to the reference trajectory (green, middle trajectory). In (**c**) and (**e**) shaded regions, color matched to their respective trajectories, are bootstrap SEM. The x-axis is labeled Time (ms), this should be understood as time on the reference trajectory. * - $p = 0.0196$ and # - $p = 0.05$. $p$ values derived from one-sided 50-repetition bootstrap tests of differences between each outcome trajectory and the reference trajectory. a.u. - arbitrary units. Source data are provided as a Source Data file.

speed reference trajectory. Post-correct trial dynamics were significantly faster in both prestimulus windows (50-repetition bootstrap test: $p = 0.0196$), whereas error-trial dynamics were significantly faster in the last prestimulus window (−200:0 ms; 50-repetition bootstrap test: $p = 0.0196$; Fig. 8e). Finally, error and post-error trials have significantly slower poststimulus trajectories in the last poststimulus window (200:400 ms) as compared to the speed reference trajectory, (50-repetition bootstrap test: $p = 0.0196$; Fig. 8e) consistent with their longer RTs. Altogether, these results complement behavioral results in that the initial condition shifts as a function of previous trial outcome and not just due to errors. These results suggest that slower or faster RTs after an error or correct trial are at least partially due to slower or faster prestimulus dynamics respectively (see Fig. S18d for complementary findings in single trials). Results were near identical when we used CCE and ECC sequences (Fig. S19).

The single-trial dynamics organized by trial outcome also suggest that neural state for post-error trials is separated from the post-correct trials at the time of movement onset (Fig. S18d). We reasoned that such differences would lead to differences in choice selectivity between the different trial outcomes before movement onset. Consistent with this reasoning, the six-dimensional Euclidean distance between left and right choice trajectories was largely flat until ~250 ms before movement-onset at which point it increased for all trial types (Fig. S18e). Post-error trials demonstrated the strongest choice selectivity as compared to all other trial types at least 250 ms before movement onset, with the difference between trial types peaking ~90 ms before movement onset (Fig. S18e).

These findings are consistent with the dynamical systems approach and the hypothesis in Fig. 1h as they demonstrate that initial condition before stimulus onset is dependent upon trial history and

that pre- and post-stimulus dynamics slow down after errors as compared to after correct trials.

### Changes in initial condition due to outcome of the previous trial likely leads to RT-related changes in the initial condition

Our results in the previous section suggest that changes in RT are at least in part due to alterations in the initial condition due to the previous trial's outcome. To further test this hypothesis, we performed two analyses.

First, we projected the firing rates organized by outcome and choice, onto the RT subspace (defined using the first 6 principal components of the PCA in Fig. 4). If the space defined by trial outcome, and space defined by RT show a strong degree of overlap (High Overlap in Fig. S20a), then the cross projection would reveal meaningful structure. In contrast, if the subspaces were independent or non-overlapping, then cross projection would be largely unstructured[25] (Independent in Fig. S20a). Consistent with our hypothesis, when we projected the firing rates organized by outcome and choice onto the RT subspace, we found near identical structure to what we observed when performing PCA on the firing rates organized by trial outcome and choice (Fig. S20b). We also performed the converse of this analysis where we projected firing rates organized by RT and choice into the space defined by trial outcome and choice (Fig. S20c). Consistent with our hypothesis that changes in trial outcome leads to changes in RT, we found near identical structure to what we observed when performing PCA on the firing rates organized by choice and RT (Fig. 4). These cross projection analyses show that the subspaces identified by trial outcome and choice, and RT and choice are highly overlapping with one another.

To quantify the strength of this overlap we first used a previously developed alignment index[25]. Briefly, the index calculates the trace of the matrix that results from the projection of the RT space onto the first six principal components of the outcome subspace (i.e., sum of eigenvalues) and divides this by the sum of the eigenvalues from the PCA on firing rates organized by RT and choice (see "Subspace overlap analysis"). Thus the index, as used here, quantifies the amount of variance in the RT space (Fig. 4a, b) that could be accounted for by the outcome subspace (Fig. 8a, b). This analysis revealed that ~ 77% of the total variance for the RT space was explained by the outcome subspace, suggesting that the previous trial's outcome has a large impact in explaining prestimulus firing rate covariation with RT.

Second, we performed a dPCA[49] on the population firing rates in the 600 ms before checkerboard onset organized by previous trial's outcome and choice, and another organized by RT and choice. The respective axes that maximally separated as a function of previous trial's outcome and that maximally separated as a function of RT demonstrated significant overlap with an angle of 47.8° between them. These results suggest that the previous trial's outcome leads to a shift in prestimulus dynamics consistent with determining the speed of the dynamics and therefore eventual RTs.

Collectively, the past trial's outcome leads to different initial conditions, slower pre- and poststimulus dynamics and ultimately leads to RT variability, all in line with the hypothesis in Fig. 1h.

## Discussion

Our goal in this study was to identify the dynamical system that best described decision-related neural population activity in PMd. Inspired by studies of neural population dynamics related to motor planning and timing[20–22,26], we investigated the neural population dynamics in PMd of monkeys performing a red-green RT decision-making task[18,27]. The prestimulus neural state in PMd, a proxy for the initial condition of the dynamical system, strongly predicted RT, but not choice. We observed these effects across and within stimulus difficulties and also on single trials. Furthermore, faster RT trials had faster neural dynamics and separate initial conditions from slower RT trials.

Additionally, poststimulus, choice-related dynamics were altered by the inputs with easier checkerboards leading to faster dynamics than harder ones. Finally, these initial conditions and the behavior for a trial depended on the previous trial's outcome, where RTs and prestimulus trajectories were slower for post-error compared to post-correct trials. Together, our results suggest that decision-related neural population activity in PMd is well described by a dynamical system where the choice-related dynamics (the output of the system) depend on initial conditions (influenced by trial outcome) and the sensory evidence (which solely determines the eventual choice).

At the highest level, these observations highlight how a dynamical systems approach (alternatively, computation through dynamics) can help understand the link between the time-varying activity of neural populations and behavior[6,15,26,50–54]. Regardless of species or brain region, an increasingly common finding is that neurons associated with cognition and motor control are often heterogeneous and demonstrate complex time-varying patterns of firing rates and mixed selectivity[8,15,41,52,55]. Simple models or indices, although attractive to define, are often insufficient to summarize the activity of these neural populations[15,52,56], and even if one performs explicit model selection on single neurons using specialized models[37], the results can be brittle because of the heterogeneity inherent in these brain regions[56]. The dynamical systems approach addresses this problem by using dimensionality reduction methods such as PCA and TCA, reduced-rank regression, decoding, RNN models, and optimization techniques to understand collective neuronal activity of different brain regions and tasks, generally summarizing the trial-averaged dynamics of large population datasets in lower dimensional subspaces[14,15,41]. Here, we demonstrated that >85% of the variance from the trial-averaged firing rate activity of nearly 1000 neurons in PMd during decisions could be explained in just a few (six) dimensions.

This trial-averaged, low-dimensionality should be appreciated in the context that it allows us to understand the task-related, latent neural dynamics underlying decision-making. However, that six dimensions suffice to explain neural activity in PMd is first due to trial-averaging that suppresses variability in both firing rates and spiking variability (Fig. S21). Other factors in our study including the use of highly-trained animals performing a task with small stimulus sets, one-to two-dimensional behavioral reports, and well controlled behavioral conditions further minimize possible variability. The single-trial dimensionality of PMd during decision-making is likely to be much higher under more naturalistic conditions when factors such as arousal, unintended arm, body, and eye movements, and variability in spiking are present. Consistent with this notion, experiments with large population datasets ( > 10,000 units) in the primary visual cortex in mice running on a ball and passively viewing visual stimuli drawn from large stimulus sets often require 100s of dimensions to capture a majority of the variance[57].

Besides compactly describing trial-averaged population activity, there are three other clear advances afforded by using a dynamical systems approach to study decisions. First, we find lawful relationships between the low-dimensional activity of neural populations and task variables such as choice, RT, stimulus difficulty and past outcomes[14,15]. Second, analysis of this low-dimensional activity allows one to arbitrate between different dynamical hypotheses. The dynamical system that was most consistent with the PMd data was one where prestimulus activity correlated with RT, but only minimally with choice (Fig. 1h). Finally, this dynamical system naturally bridges previously disparate findings from studies of speed-accuracy tradeoff[28–31,58], post-outcome adjustment[32,59], motor planning[20] and timing[26] and provides a common framework for deriving models for the neural computations underlying decision-making.

We augmented our trial-averaged analyses with cross-validated single-trial approaches that used TCA, reduced-rank regression, decoding, and regression to suggest that the dynamical system most

consistent with our data is likely one where the initial conditions covary with RT and are modified by trial outcome (Fig. 1h). We also showed that a simple unconstrained linear dynamical system without external input can explain single-trial variance in neural and behavioral data before and after checkerboard onset (Fig. S13), and a fully expressive nonlinear dynamical system through LFADS[44] could predict single-trial firing rates of the population of neurons (Fig. S14). Both LFADS firing rates and factors (Fig. S14c), and firing rates from the LDS (Fig. S13c) predicted RT suggesting that these dynamical systems are capturing behaviorally relevant neural variance.

Larger single-trial datasets could provide two advantages to better understand dynamics underlying decision-making. First, these datasets would allow us to fully characterize the relative contributions of the position of the initial condition and the velocity of the initial condition to decision-related dynamics and behavior. In particular, further analyses of the curvature, velocity relative to the mean trajectory, path length, and speed of the trajectories will lead to an even better description of the single-trial dynamics underlying decisions as has been done for motor planning[20,60,61]. Second, we could directly recover the dynamical system from the data either through reverse engineering the generator RNN from our LFADS fits or using methods that can fit switching linear dynamical systems with switch times either constrained to various task events (e.g., target onset, checkerboard onset, and movement onset)[62], or use unsupervised switching linear dynamical systems[63,64].

Our results, mainly that decision-related neural activity and behavior are well described by a dynamical system dependent upon both initial conditions and inputs, are inconsistent with simple one-dimensional drift diffusion models (DDM) where decision-making behavior is solely driven to a bound by accumulation of sensory evidence[65–67]. Including variable drift rates and starting points biased towards a particular choice in a one-dimensional DDM would be insufficient towards recapitulating prestimulus decision-related signals that covary with RT but not choice. Variable non-decision times could potentially explain the RT behavior reported here. However, the neural effect of a change in non-decision time is thought to relate to changes in the initial latency of decision-related responses and does not predict changes in the prestimulus neural state. Thus, while simple one-dimensional DDMs with a variable non-decision time may explain the behavior observed herein they would fail to recreate the observed variability in the initial condition.

Thus, these results suggest that additional dimensions might be needed for the DDM to faithfully replicate our neural data. For example, consider a two-dimensional DDM, where the x and y axes are the bounds. If the initial state is close to the origin along the 45° diagonal line, then RTs would be faster for both choices. Conversely, if the initial state is farther from the origin along the 45° diagonal line for both choices then the RTs would be longer. Such a model is consistent with our data and could potentially explain the behavior and neural responses described here.

Cognitive process models with an additive or multiplicative stimulus-independent gain signal, previously described as urgency and successfully used to describe monkey behavior and neural activity[28,68–71], could also faithfully model the behavior and the neural dynamics. A variable additive gain signal, which adds inputs to accumulators for left and right choices in a race model for decisions, would lead to different initial conditions and thus faster dynamics for faster RTs and slower dynamics for slower RTs[30]. Similarly, a multiplicative gain signal would also lead to differences in both the initial firing rates and control the speed of decision-making behavior[28,68]. Both types of gain signals generate similar predictions about RT and choice behavior and are often difficult to distinguish using trial-averaged firing rates as done here. Even in our RNN models, both inputs and gain changes lead to shifts in prestimulus activity and these shifts are correlated with RT (Fig. S2d, e). One way to resolve this impasse would be to employ single-trial analysis[57,72] of neural responses in multiple brain areas using a task paradigm that dispenses sensory evidence over the course of a trial such as in the tokens[17] or pulses task[8].

Typically, researchers have focused on the slowing down of responses after an error, a phenomenon termed post-error slowing[32,48]. However, our findings suggest that both correct and error outcomes can influence the pre- and poststimulus decision-making neural dynamics on subsequent trials suggesting that post-error slowing could be better understood under the umbrella of post-outcome adjustments[47]. It is currently unclear how these post-outcome adjustments in PMd emerge. One possibility is that these adjustments emerge from the internal dynamics of PMd itself. We examined the loadings on the various principal components to test for the possibility of "error" cells driving the effects of trial outcome (Fig. S22). However, our preliminary analysis did not provide clear evidence for the existence of such error cells and instead suggest that error signals are better characterized as distributed throughout the population. Errors vs. correct trials could lead to a shift in the initial condition due to recurrent dynamics that occur in PMd due to the presence or absence of reward. Such error-related signals have been observed in premotor and motor cortex and have even been used to augment brain computer interfaces[73]. Alternatively, the changes observed in PMd could emerge from inputs from other brain areas such as the anterior cingulate cortex (ACC) which is known to monitor trial outcome[74], or the supplementary motor area (SMA), which has been implicated in timing of motor actions and evaluative signals related to outcome[29,75]. Simultaneous recordings in PMd and these brain areas are necessary to tease apart the contribution, if any, of these areas to the initial condition changes observed in PMd.

We have shown that the outcome of the previous trial alters the initial conditions for subsequent trials. There are certainly other factors that can alter initial conditions. In particular, recent studies have shown that both neural activity and behavior as indexed by RT, performance, and pupil size drifts over slow time scales and that these slowly drifting signals are likely a process independent of deliberation on sensory evidence[70,76,77]. Such effects emerge over several hours with maximal alterations in behavior happening at the end of sessions. However, our monkeys do not demonstrate a large drop off or change in performance over the course of a session (Fig. S18a). Nevertheless, we believe that such effects could also contribute to changing initial conditions in other studies[70,77].

We found that prestimulus neural activity in PMd and in this task did not covary with or predict eventual choice. However prestimulus neural activity in lateral intraparietal (LIP) cortex was found to be predictive of choice for low coherence or harder random dot stimuli[35]. Our lack of an observed covariation between the initial condition and choice may be due to the randomization of target configurations. Thus, the monkeys in our experiment were disincentivized from pre-planning a reach direction. To be clear, our lack of a finding does not preclude prestimulus activity in other brain areas or even in PMd with different tasks from covarying with choice[78].

We suspect that the observed effects where initial conditions predict the RT in a cognitive task are likely to be observed in many brain areas. For example, previous results recorded in monkey dorsomedial prefrontal cortex during timing tasks[26] and in motor cortex (M1)/PMd from motor planning tasks[20,61,79,80] bear out the contention that our observation of prestimulus PMd neural population activity covarying with and predicting RTs in a decision-making task is likely not solely localized to PMd or constrained to occur only in this task. In fact, differences in baseline neural activity between speed and accuracy conditions of speed-accuracy tradeoff tasks is found in frontal eye field[58], pre-supplementary motor area[29], M1 & PMd[16,31], and LIP[30]. We also showed that prestimulus beta band activity in this same task and in PMd was correlated with RT[81]. Additionally, in a study of post-error

slowing the level of prestimulus phase synchrony in fronto-central electrodes, was found to positively correlate with the speed of RTs[59]. These findings of neural activity changing as a result of different conditions of a speed-accuracy tradeoff task or being predictive of RTs, strongly suggest that initial conditions in multiple brain regions, and potentially some putative fronto-central motor network, effect the speed of a response. In other words, changes in the initial conditions in various brain regions before stimulus onset is likely not a localized effect and suggests either broad signalling[82] from some source or even feed-forward/feedback mechanisms between brain regions.

In summary, our results are a significant advance over a previous study of dynamics in PMd during reach planning[20] that showed that in a delayed reach task, the position and velocity of the initial conditions correlated with RT. It was unclear from the study, what the role of inputs was and how changes in initial conditions emerge across trials. Our study suggests that initial conditions, sensitive to previous outcomes, jointly with sensory evidence (i.e., input) determine the choice-related dynamics in PMd.

## Methods

Several method sections are adapted from ref. 18 as the same data set is reanalyzed in this study. For completeness and readability, some aspects are replicated here, but much of the methods focuses on key details about the various dimensionality reduction techniques such as PCA, TCA, fits of linear dynamical systems, reduced-rank regression, decoding, and LFADS analyses. Table 1 summarizes the major analyses performed in the manuscript and corresponding figures associated with the analyses. The majority of the analyses were performed using MATLAB including the statistics and machine learning, parallel computing and curve fitting toolboxes.

### Subjects

Experiments were performed using two adult male macaque monkeys (*Macaca Mulatta*; monkey T, 7 years, 14 kg & monkey O, 11 years, 15.5 kg) trained to touch visual targets for a juice reward. Monkeys were housed in a social vivarium with a normal day/night cycle. Protocols for the experiment were approved by the Stanford University Institutional Animal Care and Use Committee (Protocol 8856). Animals were initially trained to come out of their housing and to sit comfortably in a chair. After initial training (as described in[18]), monkeys underwent sterile surgery where cylindrical head restraint holders (Crist Instrument Co., Inc., Hagerstown, MD, United States) and standard circular recording cylinders (19 mm diameter, Crist Instrument Co., Inc.) were implanted. Cylinders were placed surface normal to the cortex and were centered over caudal dorsal premotor cortex (PMdc; +16, 15 stereotaxic coordinates, see Fig. 2g). The skull within the cylinder was covered with a thin layer of dental acrylic or palacos.

### Apparatus

Monkeys sat in a customized chair (Synder Chair System, Crist Instrument Co., Inc.) with their head restrained. The arm that was not used to respond in the task was gently restrained with a tube and cloth sling. Experiments were controlled and data collected using a custom computer control system (Mathworks' xPC target and Psychophysics Toolbox, The Mathworks, Inc., Natick, MA, United States). Stimuli were displayed on an Acer HN2741 monitor approximately 30 cm from the monkey. A photodetector (Thorlabs PD360A, Thorlabs, Inc., Newton, NJ, United States) was used to record the onset of the visual stimulus at a 1 ms resolution. A small reflective spherical bead (11.5 mm, NDI passive spheres, Northern Digital, Inc., Waterloo, ON, Canada) was taped to the middle finger, 1 cm from the tip, of the active arm of each monkey; right for T and left for O. The bead was tracked optically in the infrared range (60 Hz, 0.35 mm root mean square accuracy; Polaris system, NDI). Eye

position was tracked using an overhead infrared camera with an estimated accuracy of 1° (ISCAN ETL-200 Primate Eye Tracking Laboratory, ISCAN, Inc., Woburn, MA, United States). To get a stable image for the eye tracking camera, an infrared mirror (Thorlabs, Inc.) transparent to visible light was positioned at a 45° angle (facing upward) immediately in front of the nose. This reflected the image of the eye in the infrared range while allowing visible light to pass through. A visor placed around the chair prevented the monkey from touching the juice reward tube, infrared mirror, or bringing the bead to its mouth.

### Task

Experiments were made up of a sequence of trials that each lasted a few seconds. Successful trials resulted in a juice reward whereas failed trials led to a time-out of 2–4 s. A trial started when a monkey held its free hand on a central circular cue (radius = 12 mm) and fixated on a small white cross (diameter = 6 mm) for ~ 300–485 ms. Then two iso-luminant targets, one red and one green, appeared 100 mm to the left and right of the central hold cue. Targets were randomly placed such that the red target was either on the right or the left trial-to-trial, with the green target opposite the red one. In this way color was not tied to reach direction. Following an additional center hold period (400–1000 ms) a static checkerboard stimulus (15 × 15 grid of squares; 225 in total, each square: 2.5 mm × 2.5 mm) composed of isoluminant red and green squares appeared superimposed upon the fixation cross. The monkey's task was to move their hand from the center hold and touch the target that matched the dominant color of the checkerboard stimulus for a minimum of 200 ms (for full trial sequence see Fig. 2b). For example, if the checkerboard stimulus was composed of more red squares than green squares the monkey had to touch the red target in order to have a successful trial. Monkeys were free to respond to the stimulus as quickly or slowly, within an ample ~ 2s time frame, as they chose. There was no delayed feedback therefore a juice reward was provided immediately following a successful trial[10]. An error trial or miss led to a timeout until the onset of the next trial.

The checkerboard stimulus was parameterized at 14 levels of red ($R$) and complementing green ($G$) squares ranging from nearly all red (214 $R$, 11 $G$) to all green squares (11 $R$, 214 $G$) (for example stimuli see Fig. 2c). These 14 levels are referred to as signed coherence ($SC$), defined as $SC = 100 \times \frac{(R-G)}{(R+G)}$ (R: 4%:90%, G: –4%:–90%). Correspondingly there are seven levels of color coherence, agnostic to the dominant color, defined as $C = 100 \times \frac{|R-G|}{(R+G)}$ (4-90%).

The hold duration between the onset of the color targets and onset of the checkerboard stimulus was randomly chosen from a uniform distribution from 400 to 1000 ms for monkey T and from an exponential distribution for monkey O from 400-900 ms. Monkey O attempted to anticipate the checkerboard stimulus therefore an exponential distribution was chosen to minimize predictability.

### Effects of coherence on accuracy and reaction time (RT)

Behavior was analyzed by fitting psychometric and RT curves on a per-session basis and averaging the results across sessions. Behavioral data was analyzed in the same sessions as the electrophysiological data. In total there were 75 sessions for monkey T (128,774 trials) and 66 sessions for monkey O (108,365 trials). On average there were ~ 1500 trials/session. Both incorrect and correct trials for each $SC$ were included for estimating RT/session.

Data were fit to a psychometric curve to characterize how discrimination accuracy changed as a function of stimulus coherence. For each session a monkey's sensitivity to the checkerboard stimulus was estimated by estimating the probability (p) of a correct choice as a function of the color coherence of the checkerboard stimulus ($C$). The accuracy function was fit using a Weibull cumulative distribution function.

**Table 1 | Summary of analyses, models, and simulations used in the manuscript**

| Analysis | Sessions/Units | Data/Parameters | Figures |
|---|---|---|---|
| Principal Components Analysis (PCA) with overlapping RT bins and choice | • 141 sessions<br>• 996 units<br>• 30 ms Gaussian<br>• 801 units (*)<br>• 15 ms Gaussian and 50 ms Boxcar (') | Smoothed PSTHs of trial-averaged units combined across sessions and organized by overlapping RT bins and choice; 6D trajectories for associated KiNeT analysis | • Fig. 4<br>• Fig. S5*<br>• Fig. S7<br>• Fig. S10*<br>• Fig. S11'<br>• Fig. S20 |
| Principal Components Analysis (PCA) with coherence and choice | • 141 sessions<br>• 996 units<br>• 30 ms Gaussian | Smoothed PSTHs of trial-averaged units combined across sessions and organized by coherence and choice; 6D trajectories for associated Kinematic analysis of Neural Trajectories (KiNeT) | • Fig. 7 |
| Principal Components Analysis (PCA) with trial outcome and choice | • 141 sessions<br>• 996 units<br>• 30 ms Gaussian | Smoothed PSTHs of trial-averaged units combined across sessions and organized by outcome of trials and choice; 6D trajectories for associated KiNeT | • Fig. 8<br>• Fig. S18<br>• Fig. S19<br>• Fig. S20 |
| Principal Components Analysis (PCA) with nonoverlapping RT bins and choice | • 141 sessions<br>• 996 units<br>• 30 ms Gaussian | Smoothed PSTHs of trial-averaged units combined across sessions and organized by nonoverlapping RT bin and choice; 6D trajectories for associated KiNeT | • Fig. 5 |
| Choice/outcome decoding analysis & RT regression analysis<br>  • Curve Fitting Toolbox<br>  • Statistics & Machine Learning Toolbox | • 51 sessions<br>• 2 to 32 units<br>• 157 to 2349 trials | Binned spike counts (20ms bin size) | • Fig. 6<br>• Fig. 8<br>• Fig. S16<br>• Fig. S17 |
| Single neuron regression analyses | • 141 sessions<br>• 996 units | Smoothed firing rates (50ms causal boxcar) | • Fig. S10 |
| Latent Factor Analysis of Dynamical Systems (LFADS)<br>  • https://github.com/lfads | • 1 session; 23 units (*)<br>• 16 sessions; 11 to 32 units (')<br>• 631 to 1403 trials | Binned spike counts (10ms bin size); 8 factors | • Fig. 6*<br>• Fig. S14'<br>• Fig. S18* |
| Tensor Component Analysis (TCA),<br>  • https://www.tensortoolbox.org/<br>  • https://github.com/ahwillia/tensor-demo | • 44 sessions<br>• 2 to 32 units<br>• 370 to 2349 trials | Causally smoothed firing rates (50 ms boxcar), ranks from 1 to 4. Sessions with fewer than 4 units we only ran TCA up to the maximum number of units for that session. | • Fig. S12 |
| Linear Dynamical Systems (LDS),<br>  • https://github.com/gamaleldin/CFR | • 31 sessions<br>• 11 to 32 units<br>• 97 to 960 trials | Causally smoothed firing rates (50 ms boxcar); step of 100 ms. Only left trials were analyzed. | • Fig. S13 |
| Reduced-rank regression<br>  • https://github.com/cmccomb/RedRank | • 41 sessions<br>• 2 to 32 units<br>• 370 to 2349 trials | Causally smoothed firing rates (50 ms boxcar); step of 100 ms. | • Fig. S15 |
| Recurrent Neural Network models (RNNs)<br>  • https://github.com/murraylab/PsychRNN | • 100 units | ReLu nonlinearity | • Fig. S2 |
| Simulated population of PMd neurons | • 350 units | 200 Increased, 100 Decreased, and 50 perimovement units | • Fig. S3 |

Where applicable, links to the toolboxes used are provided. Most analyses were performed using MATLAB.

### Weibull cumulative distribution function:

$$p(C) = 1 - 0.5 e^{-\left(\frac{C}{\alpha}\right)^{\gamma}} \qquad (2)$$

The discrimination threshold $\alpha$ is the color coherence level at which the monkey would make 81.6% correct choices. The parameter $\gamma$ describes the slope of the psychometric function. Threshold and slope parameters were fit per session and averaged across sessions. We report the mean and standard deviation of threshold and $R^2$ values from the fit in the text.

Mean RT was calculated per *SC* on a session-by-session basis and averaged across sessions. Results are displayed in Fig. 2e with error bars denoting $2 \times$ SEM and lines between the averages to guide the eyes. RT was also regressed with $\log_{10}(C)$ per session. The fit coherence-RT model was used to predict RTs and calculate $R^2$ on a per session basis. $R^2$ values were averaged across sessions per monkey and are reported in Fig. 2f as percentage of variance explained. The general framework and equations for linear regression and $R^2$ calculations are provided in "Linear regression to relate RT and firing rate, and logistic regression to decode choice".

### Electrophysiological recordings

Electrophysiological recordings were guided by stereotaxic coordinates, known response properties of PMd, and neural responses to muscle palpation. Recordings were made anterior to the central sulcus, lateral to the precentral dimple and medial to the spur of the arcuate sulcus. Electrodes were placed in the PMd contralateral to the dominant hand of the monkey (T: right arm, O: left arm). Recording chambers were placed surface normal to the cortex to align with the skull of the monkey and recordings were performed orthogonal to the surface of the brain. Estimates of upper and lower arm representation was confirmed with repeated palpation at a large number of sites to identify muscle groups associated with the sites.

Single electrode recording techniques were used for a subset of the electrophysiological recordings. Small burr holes in the skull were made using handheld drills (DePuy Synthes 2.7 to 3.2 mm diameter). A Narishige drive (MO-972A, Narishige International USA, Inc., Amityville, NY, United States) with a blunt guide tube was placed in contact with the dura. Sharp FHC electrodes ($> 6$ MΩ, UEWLGCSEEN1E, FHC, Inc., Bowdoin, ME, United States) penetrated the dura and every effort was made to isolate, track, and stably record from single neurons.

180 μm thick 16-electrode linear multi-contact electrode (U-probe, see Fig. 2g; Plexon, Inc., Dallas, TX, United States); interelectrode spacing: 150 μm, contact impedance: ~100 kΩ) recordings were performed similarly to single electrode recordings with some modifications. Scraping away any overlying tissue on the dura, under anesthesia, and a slightly sharpened guide tube aided in slow U-probe penetration ( ~2–5 μm/s). U-probe penetration was stopped once a reasonable sample of neurons was acquired, potentially spanning multiple cortical layers. Neural responses were allowed to stabilize for 45-60 minutes before normal experimentation began. Monkey T had better recording yields on average ( ~16 units/session) than monkey O

( ~ 9 units/session). Additionally, lowering the electrode necessitated careful observation to ensure the electrode did not bend or break at the tip, or excessively dimple the dura. Therefore, it was not possible to precisely localize the U-probes with a grid system between sessions. We used the Blackrock system and the Cerebus Software for acquiring all electrophysiological data.

## Unit selection and classification

The electrophysiological recordings consist of 996 units (546 units in T and 450 units in O, including both single neurons and multi-units) recorded from PMd of the two monkeys as they performed the task over 141 sessions. Chosen units were included as they were well isolated from other units/separated from noise and modulated activity in at least one task epoch.

U-probes were useful for recording from isolated single neurons as U-probes are low impedance ( ~ 100 kΩ) with a small contact area. A conservative threshold was used to maximize the number of well defined waveforms and to minimize contamination from spurious non-neural events. Single neurons were delineated online by the hoops tool of the Cerebus system software client (Blackrock Microsystems, Salt Lake City, UT, United States) after the electrodes had been in place for 30 - 45 minutes. When a spike was detected via thresholding, a 1.6 ms snippet was stored and used for subsequent evaluation of the clusters as well as modifications needed for spike sorting.

Some electrodes in U-probe recordings captured mixtures of 2 or more neurons, well separated from each other and noise. In the majority of cases the waveforms were separable and labeled as single units. These separations were verified by viewing the waveforms in principal component (PC) space using custom code in MATLAB (The MathWorks, Inc., Natick, MA, United States). MatClust, a MATLAB based clustering toolbox (https://www.mathworks.com/matlabcentral/fileexchange/39663-matclust), or Plexon Offline Sorter (Plexon, Inc.) were used to adjust the clusters that were isolated online. Recording activity labeled as multi-units were mixtures of 2 or more neurons whose waveforms were reasonably demarcated from the noise but not easily separable using a PCs method.

The number of interspike interval (ISI) violations after clustering and sorting was used to mitigate subjectivity in the classification of units. A unit was labeled as a single neuron if the percentage of ISI violations (refractory period of ≤ 1.5 ms) was ≤ 1.5%, otherwise it was labeled as a multi-unit. 801/996 PMd units were labeled as single neurons (T: 417, O: 384, median ISI violation = 0.28%, mean ISI violation = 0.43%, ~ 0.13 additional spikes/trial). Therefore 195/996 units were labeled as multi-unit (T: 129, O: 66, mean ISI violation = 3.36%, ~ 1.4 additional spikes/trial). We included multi-units as well because they gave us additional power for our decoding analyses and prior work has shown that the inclusion of multi-units does not distort recovery of low-dimensional dynamics from neural activity[42].

Units from both monkeys were pooled together as the electrophysiological characteristics were similar. Change-of-mind trials ( ~ 2–3%) were excluded from averaging as the change in reach direction mid-movement execution made the assignment of choice ambiguous. Incorrect and correct trials arranged by choice were averaged together.

## Peri-event firing rates

We calculated peri-event time histograms and other analyses (e.g., PCAs) aligned to either checkerboard (e.g., in Fig. 3) or movement onset (i.e., Fig. S18e) using the following procedure. 1) We first binned spike times for each trial at 1 ms resolution for a condition of interest (e.g., fast RT bin and left reaches) aligned to checkerboard or movement onset. When trials were aligned to checkerboard onset we removed (i.e., replaced with NaNs) all spikes 50 ms before movement onset (estimated from RT) until the end of the trial to ensure that movement related spiking activity did not spuriously lead to ramping

in the checkerboard period. 2) We then convolved the spike train with a Gaussian kernel ($\sigma = 30$ ms) to estimate the instantaneous firing rate (e.g., $r_i(t, RT, left)$) for a trial. 3) Finally, firing rates were trial-averaged within a condition (e.g., $\bar{r}(t, RT, left)$). We used a 30 ms kernel based on previous studies in our and other lab(s). However to ensure our results were not a trivial artifact of smoothing, peri-stimulus time histograms (Fig. S4) and other key analyses (Fig. S11) were performed with a smaller Gaussian kernel (15 ms) and a causal boxcar kernel (50 ms). As expected, results were very similar if only slightly noisier. None of the conclusions change when using these alternative smoothing approaches.

## Principal component analysis (PCA) of trial-averaged PMd firing rates

PCA was used to examine firing rate variance in the recorded PMd neural population. PCA reveals dimensions that explain a large percentage of the data while making few assumptions about the underlying structure of the data. The dimensions extracted by PCA may not always be meaningful[49]. However, they often align well with behavioral variables.

The general procedure for performing a PCA involved creating a 4D matrix containing all 996 units (or 801 single units as in Fig. S10) and their condition-averaged firing rate activity (i.e., peristimulus time histograms; Section "Peri-event firing rates") windowed about checkerboard onset ( ~ -600 ms: ~ 1200 ms) and organized by level of condition (e.g., coherence, RT, or past outcome) within a reach direction. Typical matrix organization was windowed firing rate x units x reach x condition (C) ( ~ 1800 × 996 × 2 × C). Condition could be coherence (7, i.e., Fig. 7a), RT (11, i.e., Fig. 4b), or past outcome (4, i.e., Fig. 8b). For the RT PCA we used overlapping RT bins ( ~ 100 - 200 trials/bin, Fig. S6b) as it afforded a degree of smoothing to understand how neural population dynamics covary with RT and choice. However, this could be statistically problematic as trials then contribute to multiple conditions. Thus we also split our data into nonoverlapping RT bins ( ~ 120 - 220 trials/bin, Fig. S6a) and found that our results are not a trivial artifact of using overlapping bins (Fig. 5).

The 1800 × 996 × 2 × C matrices are reorganized into 2D matrices appropriate for PCA. First, we further constrain the time window such that individual conditions are now windowed 400 ms before checkerboard onset to the median RT of the condition (i.e., for coherence and outcome analyses) or lower bound of the RT bin (i.e. for RT analyses) minus 25 ms (e.g., 300 ms - 25 ms = 275 ms). The 25 ms is subtracted to remove any artifact from data replaced with NaNs (as explained above, "Peri-event firing rates"). Thus each condition is now windowed − 400: ~ 400 ms around stimulus onset for all units and vertically concatenated by order of condition. Thus the RT, coherence, and outcome condition matrices respectively become ~ 17000 × 996, ~ 11,000 × 996 and ~ 6,000 × 996 matrices (11/7/4 conditions each with slightly varying time windows vertically concatenated × all 996 units, with all left reach conditions vertically concatenated above all right reach conditions).

These trial-averaged 2D firing rate matrices are centered by subtracting the mean of each column (i.e. units) and then normalized by dividing by the square root of the 99th percentile of that column (i.e., soft normalization). Soft normalization reduces the bias of units with high firing rates and ensures that each unit has roughly the same overall variability across conditions. Eigenvectors and eigenvalues were calculated using the pca function in MATLAB.

To perform the within-coherence RT PCAs (3 shown, Fig. 7c) we first generated seven (1 for each coherence) trial-averaged firing rate matrices organized by RT and choice (each 11 RT bins × 2 choices). Then we projected each of these 7 matrices into the PC space obtained by PCA on the RT and choice data (i.e., Fig. 4)

## Bootstrapped trial-averaged PMd firing rate distributions for statistical analysis

We also generated 50 bootstrapped firing rate distributions by sampling with replacement for each unit's trials in a condition. We performed PCA on these bootstrapped firing rates and used them for generating the KiNeT analysis errorbars (e.g., Fig. 4, Fig. 5), compute confidence intervals for correlation analyses (e.g., Fig 7), as well as for comparing between conditions in Fig. S18.

## Estimation of number of dimensions to explain the data

We used the approach developed by[41] to estimate the number of dimensions that best described our data. The assumption of this method is that the firing rates of the $k^{th}$ neuron for the $i^{th}$ trial given a RT bin and choice ($r_k^i(t,RT,choice)$) are assumed to be composed of a mean signal rate ($q_k(t|RT,choice)$) and a noise rate that fluctuates across trials ($\eta_k^i(t,RT,choice)$).

$$r_k^i(t,RT,choice) = q_k(t,RT,choice) + \eta_k^i(t) \tag{3}$$

Noise here encompasses both contributions from the random nature of spike trains as well as systematic but unknown sources of variability. Averaging over trials:

$$\bar{r}_k(t|RT,choice) = q_k(t|RT,choice) + \bar{\eta}_k(t|RT,choice) \tag{4}$$

Where $\bar{\eta}_k(t,RT,choice)$ is the average noise over N instantiations (i.e., trials) of the noise term $\eta_k^i(t,RT,choice)$.

The overall mean firing rate over time and conditions ($\bar{r}$) is given as:

$$\bar{r}_k = <q_k(t,RT,choice)> + <\bar{\eta}_k(t,RT,choice)> \tag{5}$$

$$= q_k + \bar{\eta}_k \tag{6}$$

Note, none of these assumptions are strictly true. Noise may not be additive and it may depend on RT bin and may increase or decrease during various phases of the trial. However, these assumptions illustrate the problem encountered in identifying the number of dimensions to best describe the data.

Under these assumptions PCA attempts to identify a covariance matrix as

$$C_{ij} = <\bar{r}_i(t,RT,choice) - \bar{r}_i><\bar{r}_j(t,RT,choice) - \bar{r}_j> \tag{7}$$

Which can be simplified (see[41] for more details) to:

$$C_{ij} = Q_{ij} + H_{ij} \tag{8}$$

Where $Q_{ij}$ is a signal covariance and $H_{ij}$ is the noise covariance.

Our goal is to perform PCA on $Q_{ij}$. However, because our data were not collected simultaneously, we cannot calculate $Q_{ij}$ as we do not have a good estimate of $H_{ij}$.

Nevertheless, even with trial-averaged data, one can provide an estimate of $H_{ij}$ by constructing putative noise matrices based on the simplifying assumption that the noise is largely independent in neurons with perhaps modest noise correlations. To generate representative noise traces for our firing rates, consider firing rates of two trials $r_i^k(t)$, and $r_i^l(t)$ for the $i^{th}$ neuron (the superscripts $k$ and $l$ refer to any two different trials). Notice that if one subtracts the firing rates for these trials, then the signal components, which by assumption are identical

for the two trials are removed and we are only left with the noise:

$$r_i^k(t|RT,choice) - r_i^l(t|RT,choice) = \eta_i^k(t,RT,choice) - \eta_i^l(t,RT,choice) \tag{9}$$

Which is just subtraction of two random instantiations of the same process, which can be written as:

$$\eta_i^k(t,RT,choice) - \eta_i^l(t,RT,choice) = \sqrt{2}\eta_i^m(t,RT,choice)$$
$$= \sqrt{2M}\bar{\eta}_i(t,RT,choice) \tag{10}$$

Where the final equality emerges from the equations for standard error of the mean. For example, $var(\bar{X}) = var(\sum_{i=1}^M \frac{X_i}{M}) = \sum_{i=1}^M \frac{var(X_i)}{M}$.

Thus, we can generate estimates of the average noise $\bar{\eta}_i(t,RT,choice)$

$$\bar{\eta}_i(t,RT,choice) = \frac{1}{\sqrt{2M}}(r_i^k(t,RT,choice) - r_i^l(t,RT,choice)) \tag{11}$$

Using this equation, we can estimate $H_{ij}$.

We denote $C_{ij}$ as the signal+noise covariance matrix and $H_{ij}$ as the noise covariance matrix. We estimate the eigenvalues and eigenvectors of both covariance matrices and compare them to identify the number of dimensions needed to explain the data. We used bootstrapping to derive error estimates on the signal+noise PCA and identified the number of dimensions needed to explain the data as the first dimension where signal+noise variance was significantly below the noise variance (by at least $3 \times$ SEM).

## Kinematic analysis of neural trajectories (KiNeT)

We used the recently developed KiNeT analysis (Ref. 26, https://github.com/jazlab/KiNeT/tree/master) to characterize how state space trajectories evolve over time in terms of relative speed and position as compared to a reference trajectory. We used the first six PCs (~90% of variance) of the PCAs organized by choice and RT/outcome as these PCs were significantly different from noise in both PCAs[41].

As such we have a collection of six-dimensional trajectories ($\Omega_1, \Omega_2...\Omega_n$) differing in RT bins and choice in one analysis (Fig. 4c-f) and trial outcome and choice in another (Fig. 8c, e). The trajectory associated with the middle RT bin (cyan, Fig. 4c, f) and the trajectory associated with the Correct trial outcome (Fig. 8c, e) were chosen as reference trajectories ($\Omega_{ref}$) to calculate various parameters (e.g., Time to reference) of the other non-reference trajectories (i.e., trajectories associated with the ten other RT bins and the three other trial outcomes). All of the following calculations in this section were first performed within a particular choice and then averaged across choices. Please refer to Fig. S8 for a visualization of KiNeT analyses and glossary of terms used in the following equations.

**Time to reference**: KiNeT finds the Euclidean distances between the six-dimensional position of the reference trajectory at timepoint j ($s_{ref}[j]$) and the six-dimensional position of a non-reference trajectory ($\Omega_i(\tau)$) at all of its timepoints ($\Omega_i(\tau)$). We identified the timepoint ($t_i[j]$) at which the six-dimensional position of a non-reference trajectory ($s_i[j]$) is closest to $s_{ref}[j]$ (minimum Euclidean distance).

$$s_i[j] = \Omega_i(t_i[j]) \tag{12}$$

$$t_i[j] = \text{argmin}_\tau ||\Omega_i(\tau) - s_{ref}[j]|| \tag{13}$$

If the non-reference trajectory reaches a similar position to the reference trajectory at an earlier timepoint then it's a faster trajectory ($t_i[j] < t_{ref}[j]$) whereas if it reaches the same point at a later timepoint then it is a slower trajectory ($t_i[j] > t_{ref}[j]$) (Fig. 4f).

**Distance**: The distance between reference and non-reference trajectories at timepoint j ($D_i[j]$) is taken as the minimum Euclidean distance between the position of the reference trajectory at timepoint j ($s_{ref}[j]$) and the position of the non-reference trajectory at all its time-points ($s_i[j]$). Additionally, the size of the angles between a normalized non-reference trajectory and normalized trajectories for the 1st and last conditions (e.g., 1st and last RT bins) determines whether the current non-reference trajectory is closer to either the 1st or last condition. As defined here, if a trajectory is closer (i.e. smaller angle) to the trajectory for the 1st condition then ($D_i[j]$) is positive, otherwise it is negative (Fig. 4c).

$$D_i[j] = \pm ||s_{ref}[j] - s_i[j]|| \qquad (14)$$

**Angle**: KiNeT computes the vector between adjacent trajectories by subtracting the positions of two non-reference trajectories when they are respectively closest to the reference trajectory at timepoint j. These vectors are then normalized and all the angles between all adjacent normalized vectors is found at all timepoints. Finally, the average angle is found at each timepoint between all adjacent trajectories (Fig. 4e).

$$\Delta_i^\Omega[j] = s_{i+1}[j] - s_i[j] \qquad (15)$$

$$\theta_i[j] = \angle(\Delta_i^\Omega[j], \Delta_{i+1}^\Omega[j]) \qquad (16)$$

**Subspace similarity**: We first identified normalized vectors between adjacent trajectories for all timepoints. We then averaged these normalized vectors, so that we have the mean between trajectories (i.e. conditions) vector for each timepoint. This mean vector is again normalized as averaging normalized vectors doesn't maintain unit length ($\bar{\Delta}$). We calculate the angle between the average vector at timepoint t ($\bar{\Delta}[t]$ and the average vector at the first timepoint $\bar{\Delta}[1]$, for all timepoints $t \in \tau$. In other words we are measuring how this vector, representative of the state space, rotates relative to the first timepoint across a trial (Fig. 4d). This data is calculated separately for each choice and again is bootstrapped and averaged across the separate reaches and then across the bootstraps.

$$\bar{\theta}[t] = \angle(\bar{\Delta}[t], \bar{\Delta}[1]) \quad \forall \quad t \in \tau \qquad (17)$$

## Bootstrap calculation

Assume the test statistic for the $i^{th}$ bootstrap is $S_i$ and $S_{test}$ is what you are comparing to (e.g., 90° for the alignment angles). If you perform n bootstraps, then the unbiased $p$ value for a one-tailed $t$ test is $(1 + \sum_{i=1}^n (S_i > S_{test}))/(1 + n_{bootstraps})$. We typically bootstrapped firing rates 50 times to obtain our $p$ values (e.g., Fig. 4d, e). Thus the minimum possible $p$ value possible is 1/51 (0.0196) for a one-tailed $t$ test.

## Scalar speed

We computed scalar speed in firing rate state space for the prestimulus period within a RT bin (Fig. 4g) as the $\ell^2$ norm between the six-dimensional coordinates, of the PC data at adjacent 10 ms time steps, per RT bin and for each choice separately.

$$\ell_i^2(t) = ||\Omega_i(t + \delta t) - \Omega_i(t)||_2 \qquad (18)$$

Where $\Omega_i(t + \delta t)$ and $\Omega_i(t)$ are six-dimensional trajectories within condition i at time t + $\delta t$ and time t in the prestimulus period, respectively. $\ell_i^2(t)$ is the $\ell^2$ norm between six-dimensional trajectories within a condition at time t and t + 1. We then averaged speeds across choices and over the entire prestimulus period (−400 ms to 0).

The plotted prestimulus firing rate speed was averaged across 50 bootstraps in which trials were sampled with replacement 50 times (Fig. 4g). Separate PCA and speed calculations were performed per bootstrap.

## Choice-selectivity signal

We estimated the choice-selectivity signal by calculating the Euclidean distance between left and right reaches at all timepoints for the first six PCs within each level of the RT (Fig. 4h & Fig. 7d) and coherence (Fig. 7b) conditions aligned to checkerboard onset. We performed the same analysis for each level of the outcome condition (Fig. S18e) but aligned to movement onset.

$$CS(t) = ||\Omega_L(t) - \Omega_R(t)||_2 \qquad (19)$$

$\Omega_L(t)$ and $\Omega_R(t)$ are the six-dimensional location in state space for a left and right choice at time t.

To calculate the latency of this choice-selectivity signal, we fit the time varying choice-selectivity signal with a piecewise function of the form

$$CS(t) = b \quad \forall \quad t \le t_{Latency} \qquad (20)$$

$$CS(t) = m(t - t_{Latency})^2 \quad \forall \quad t > t_{Latency} \qquad (21)$$

$b$ is the baseline choice-selectivity and $t_{Latency}$ is the time after which the choice-selectivity signal begins to increase. The piecewise linear function assumes that before $t_{Latency}$ the value of the choice-selectivity signal is constant and roughly equal to $b$, and after $t_{Latency}$ increases as a quadratic function with a slope ($m$).

## Initial condition as a function of RT and coherence

To estimate the initial conditions shown in Fig. 7e–k, we performed the following procedure. For each coherence and RT bin, we concatenated the average location in the six-dimensional state space in the −400 ms to −100 ms epoch before checkerboard onset for both reach directions and obtained a 77x12 matrix (7 coherences, 11 RT bins, and 2 choices). We then performed a PCA on this 77x12 matrix and used the top PC as a measure of the initial condition that we used for plotting and subsequent partial correlation analyses.

## Latent factors analysis of dynamical systems (LFADS)

LFADS is a generative model which assumes that neuronal spiking activity is generated from an underlying dynamical system[44]. This dynamical system is assumed to be relatively low-dimensional (i.e. considerably smaller than the number of neurons involved) and latent factors can be extracted and exploited to recreate spiking activity on single trials. This method uses a trained autoencoder to generate initial conditions based on a trial's neurons' spike counts. This latent code serves as the initial condition to the generator RNN. From the latent code the generator infers the latent factors of all the neurons in that trial. Here LFADS was used to reconstruct single-trial trajectories within a single session recorded from 23 neurons (Figs. 6a, b and S18d). Our model consisted of eight latent factors to recreate spiking activity of single trials. Since these factors are not orthogonal to each other, PCA was performed on these eight factors and the first three PCs were visualized in Figs. 6a, b and S18d. LFADS was also used to estimate single-trial firing rates from held-out units for 16 sessions (Fig. S14). Please refer to ref. 44 for fuller descriptions of the LFADS method.

## Tensor component analysis

Our PCA on trial-averaged data revealed that prestimulus neural state covaried with RT. To test whether such a result was observable at the

single-trial level, we employed tensor component analysis (TCA)[43]. TCA decomposes a tensor X of the form $N \times T \times K$ according to the following equation:

$$X_{NTK} \sim \sum_{r=1}^{R} w_n^r b_t^r a_k^r \qquad (22)$$

Like PCA, $w^r$ is a prototypical firing rate pattern across neurons and $b^r$ is a temporal basis function within trials. Neuron factors and temporal factors are common across all trials. $a^r$ are trial factors (green vector in Fig. S12a) that can be thought of as trial-specific amplitudes for the within-trial activity patterns identified by the neuron and temporal factors.

Thus, in TCA, the trial-to-trial fluctuations in neural activity are also embedded in an R-dimensional space. TCA reduces the dimensionality of the original data tensor, reducing $NTK$ data points to $R(N + T + K)$ values, while still capturing trial-to-trial variability.

We used the Tensor Toolbox for MATLAB[83] as well as the implementation provided by[43]. We used two forms of cross-validation to assess how well our data were described by this approach. Like[43] we first used a speckled holdout approach and randomly removed 25% of the data as a test set. We trained on 75% of the data and assessed the fit of the model to the training and test datasets to measure how much neural data is explained by the TCA decomposition. One issue with this speckled hold out approach is that the $R^2$ obtained from it can be inflated due to the autocorrelation in the neural data (Fig. S12d, top panel). So we also performed a second type of cross-validation where we held out a random neuron on 25% of trials (Fig. S12d, bottom panel). In both cases, as the rank of the decomposition increased the proportion of variance explained increased for both training and test sets (Fig. S12d).

We finally used the trial-specific factors and the temporal factors and estimated if the activity in the prestimulus period predicted RT on single trials in a single session (Fig. S12c) and also across sessions (Fig. S12e).

### Fits of linear dynamical systems to single-trial activity

We examined if binned single-trial PMd firing ($X \in R^{T \times N \times M}$) rates (50 ms bins) during the prestimulus and poststimulus epochs were well described by a dynamical system. For X, $N$ is the number of neurons, $T$ is the time points of interest and $M$ is the number of trials analyzed. We first projected X onto the top principal components (PCs) of the data to produce a reduced tensor ($\underline{X} \in R^{T \times \underline{N} \times M}$). The linear dynamical system models the temporal evolution of these low-dimensional neural trajectories as fixed across trials, namely for any given trial m:

$$\underline{\dot{X}}(:,:,m) \approx \underline{X}(:,:,m)J, \forall m \in (1 \ldots M) \qquad (23)$$

This analysis thus asks if the change in neural activity at time $t + 100ms$ is predicted by using the activity at time $t$ (note 50 ms bins were used to estimate firing rates).

Where $J \in R^{\underline{N} \times \underline{N}}$ is the dynamics matrix underlying the flow field that governs evolution of the activity. $\underline{N}$ therefore determines the dimensionality of the model. To find $J$, we optimized the following objective function:

$$\hat{J} = \text{argmin} \frac{\sum_{m=1}^{M} ||\underline{\dot{X}}(:,:,m) - \underline{X}(:,:,m)J||_F^2}{\sum_{m=1}^{M} ||\underline{\dot{X}}(:,:,m)||_F^2} \qquad (24)$$

The solution of the objective function can be obtained via standard least squares optimization, and the quality of the fit is easily assessed by a cross-validated coefficient of determination ($R^2$) that equals 1 minus the minimum normalized error. To perform cross-validation, we excluded the firing rate of one test trial $m_{test}$, and fit the $J$ from data that did not include this trial. We then

obtained a measure of the cross-validated performance of the model by performing leave-one-condition-out cross-validation on the reconstruction of $X(:,:,m_{test})$ from $\hat{J}$. Other cross-validation approaches yielded similar results. We repeated this procedure for $m_{test} \in [1, ..., M]$, yielding an average leave-one-condition-out cross-validation $R^2$ statistic for each session. Data was then averaged over sessions and monkeys. We used the LDS fitting code provided with this study[84].

We fit separate dynamical systems for the pre- and post-stimulus period based on our KiNeT results that showed that the subspace angles begin to diverge after stimulus onset and with the general notion that a single LDS is unlikely to capture the rich dynamics of neural activity in these brain areas[62]. We also only show the fits of the data from left trials as results were largely similar for right trials. Including the stimulus at each time point did not materially alter the results of the dynamics analysis. We believe that PMd is much removed from the processing of the sensory stimuli and in our monkeys the sensory evidence has weaker impact on RT compared to the ongoing internal neural state of the animal (Fig. 2f).

We varied $\underline{N}$ from 2 to 10 for our fits. Including additional dimensions improved our ability to predict future neural activity. More importantly, we assessed whether this modeled neural activity was useful for describing behavior and we found that although our dynamical system only explained 40% of the single-trial neural variance, it was as good as the full activity for a session to explain the RT variability in both pre- and post-stimulus epochs (Fig. S13).

### Reduced-rank regression

We used reduced-rank regression to understand if prestimulus neural activity predicted poststimulus neural activity. In reduced-rank regression, we have a $Y$, which in our case is neural data at time $t$ of size $n \times p$, and we ask if we can predict it from neural activity at time $t_0$, and other covariates such as choice, and the unsigned coherence of the checkerboard, which are then combined together in a matrix $X$ of size $n \times q$. Where $\beta$ is a matrix of size $q \times p$.

Typically in a standard multivariate regression we minimize the following loss function:

$$L = ||Y - X\beta||^2 \qquad (25)$$

And $\beta$ can be estimated by the following equation for the standard ordinary least squares solution as:

$$\hat{\beta}_{OLS} = (X^T X)^{-1} X^T Y \qquad (26)$$

However, for reduced-rank regression, we assume an additional constraint that we need a low rank $\beta_{rrr}$ such that

$$\text{rank}(\beta_{rrr}) \leq r \qquad (27)$$

To achieve such a solution, we use the following loss function:

$$L = ||Y - \hat{Y}||^2 + ||\hat{Y} - X\beta||^2 \qquad (28)$$

where,

$$\hat{Y} = X\hat{\beta}_{OLS} \qquad (29)$$

Only the second term depends on $\beta$, we can choose $\beta$ to satisfy $X\beta = \hat{Y}_s$, where $\hat{Y}_s$, is sum of the first $s$ terms of the singular value decomposition of Y. Such a solution is possible because of the Eckart-Young theorem, which states that $\beta$ chosen in such a way will minimize the second term in the equation (28) and thereby the total loss term. We split our data into a training and validation set, and chose the rank that minimizes the mean square error in the validation set and also report this

minimum cross-validated $R^2$ (Fig. S15b, c) in the results and figures. To perform this analysis, we used an implementation available on MATLAB Central[45].

We performed reduced-rank regression on 41 sessions from both monkeys.

## Linear regression to relate RT and firing rate, and logistic regression to decode choice

We used linear and logistic regressions (decoders) to determine the variance in RT explained by spiking activity and whether spiking activity predicted choice or past outcomes, respectively. For these analyses, we leveraged the U-probe sessions where multiple neurons were recorded from at once. For monkey T, we used 24 sessions (36,690 trials) where there was a minimum of 9 neurons (one session only has 2 neurons; otherwise all other sessions had at least 9) and a maximum of 32 neurons. For monkey O, we used 27 sessions (30,831 trials) where there was a minimum of 5 neurons in a single session and a maximum of 18. Some sessions had distinct portions (e.g., the electrode was moved). In the later portion of three sessions, 2 neurons were recorded from and in another, 3 neurons were recorded from. Otherwise in all other sessions at least 5 neurons were recorded from. Variance explained and decoding accuracy shown in Fig. 6c–f is pooled across both monkeys.

For regression and decoding analyses, we used 1800 ms of spiking activity from each trial (600 ms prestimulus and 1200 ms poststimulus). For the choice and outcome decoders respectively, we used 20 ms nonoverlapping bins and 50 ms overlapping (10 ms time step) bins to bin the spike times. This provided us with 90 timepoints for the choice decoder, and 72 timepoints for the outcome decoder across all units within a session.

**Linear Regression:**
For analysis of the relationship between activity in PMd and RT, we regressed spike counts for each bin for all trials across all units for that session to RT according to the following equation:

$$RT_i = \beta_0 + \sum_{j=1}^{N} \beta_j X_{ij}(t) + \beta_c c_i \qquad (30)$$

Where $RT_i(t)$ is the RT on the $i^{th}$ trial, $X_{ij}(t)$ is the spike count in a 20 ms bin for the $i^{th}$ trial and the $j^{th}$ unit, $c_i$ is the coherence for the $i^{th}$ trial, and the $\beta_{j/c}$ are coefficients for the model. After regression, we calculated variance explained by spiking activity and coherence together for each bin by using the standard equation for variance explained.

$$R^2 = 1 - \frac{\sum_{k=1}^{M} (RT_k - \widehat{RT}_k)^2}{\sum_{i=k}^{M} (RT_k - \overline{RT})^2} \qquad (31)$$

Where $\overline{RT}$ is the mean RT, $RT_k$ is the RT for the $k^{th}$ trial, and $\widehat{RT}_k$ is the RT predicted for the $k^{th}$ trial.

For assessing if the $R^2$ values were significant, we computed a shuffled distribution (500 shuffles) where we shuffled the trials to remove the relationship between the RTs and spiking activity. We then assessed if the per bin $R^2$ values were significantly different from the 99th percentile of the shuffled distribution $R^2$ values.

**Logistic Regression to decode choice:** For decoding choice and previous outcome on a bin-by-bin basis, we used a regularized logistic regression approach. Decoders were trained with an equal number of trials for the opposing outcomes (i.e., left vs. right reaches; previous correct vs. previous error trials). The logistic regression approach assumes that the log odds in favor of one event (e.g., left) vs. right reach is given by the following equations:

$$\log\left(\frac{p(\text{Left}|X)}{1 - p(\text{Left}|X)}\right) = \beta_0 + \sum_{j=1}^{N} \beta_j X_j \qquad (32)$$

$\beta_0$ is the intercept of the model, $\beta_j$ is the model coefficient for the $j_{th}$ neuron in the current bin, $X_j$ is the spiking activity of the $j_{th}$ neuron of the current bin. The following equation is used to produce the outputs of the system: if $p(\text{Left}|X) < 0.5$ then −1 and if $p(\text{Left}|X) > 0.5$ then 1.

We used the implementation provided in MATLAB via the fitclinear function and the Broyden-Fletcher-Goldfarb-Shanno quasi-Newton algorithm to find the optimal fit for the parameters[85]. We typically attempted to predict choice or previous outcome using tens of units. To simplify the model, we decreased the collinearity of the coefficients and to avoid overfitting, we used L2 regularization (ridge regression):

$$J = \frac{\lambda}{2} \Sigma \beta^2 \qquad (33)$$

Where $J$ is the cost associated with coefficients, $\lambda$ is the penalty term (1/ number of in-fold observations), and $\beta$ are the coefficients of the model. We used 5-fold cross validation and calculated loss for each model. Accuracy is reported as, accuracy = 1 - mean(loss). We performed several variants of this choice-decoding analysis. The typical variant is shown in Fig. 6d, f and involves decoding choice from firing rates of all trial types (i.e., any coherence/RT bin). We also restricted this decoding analysis to trials of any coherence but each model is built from firing rates from within RT bins (Fig. S16c), or just the hardest or easiest coherences and also separated the data by fast, medium, and slow RT bins (Fig. S17a, b).

## Choice selectivity and covariation with RT per neuron

To assess whether individual neurons are selective for choice (or covaried with RT) we performed a regression analysis where we assessed whether the firing rate in each 50 ms bin could be explained by choice (or RT). For example, to assess if a neuron was selective for choice, we assumed that for the $j^{th}$ neuron, the firing rate on the $i^{th}$ trial at time $t$ is explained by the following linear equation:

$$FR_j^i(t) \sim \beta_0 + \beta_{\text{Choice}} \text{Choice}^i + \epsilon \qquad (34)$$

We used a standard least squares regression and estimated the 99% confidence intervals for the $\beta_{\text{Choice}}$ (or $\beta_{\text{RT}}$) and assessed if the confidence intervals overlap with 0. We then report the percent of neurons with significant $\beta_{\text{Choice}}$ (or $\beta_{\text{RT}}$) as a function of time. With a 99% confidence interval, we assume 1% of neurons to have a significant relationship with choice (or RT) just by chance.

We performed these regressions by restricting the regression to trials of the hardest coherence (Fig. S17c). We also performed another regression where we included all trials in the regression along with the coherence of the checkerboard and predicted both choice and RT (Fig. S17d). Even in these expanded regressions, prestimulus activity covaried with RT but not choice.

## Subspace overlap analysis

We wanted to determine how much variance is shared between firing rates organized by RT and choice (Fig. 4a, b) versus firing rates organized by outcome and choice (Fig. 8a, b). To do this we used a modified version of a subspace overlap index[25]. Essentially we took the covariance matrix of the firing rates organized by RT and choice (time/RT bin/choice × units; 17000 × 996; RT space) and projected them onto the first 6 PCs of the PC space organized by outcome and choice (outcome subspace; visualization of method: Fig. S20a). This makes up the numerator of the index which is then divided by the sum of all eigenvalues of the RT space (equation below).

$$A = \frac{\text{tr}(D_{\text{outcome}}^T C_{\text{RT}} D_{\text{outcome}})}{\Sigma_{i=1}^{996} \sigma_{\text{RT}}(i)} \qquad (35)$$

The alignment index, A, provides an estimate of the fraction of variance that is explained by projecting one subspace into another. tr() is the trace of a matrix, which can be proved to be the sum of its eigenvalues. $D_{outcome}$ is the first six eigenvectors of all 996 units from the PCA organized by outcome and choice. $C_{RT}$ is the covariance matrix of the firing rates of all 996 units organized by RT and choice. $\sigma_{RT}(i)$ are the eigenvalues (i) obtained from PCA on firing rates organized by RT and choice. For our purposes, we used the total variance in the denominator instead of the same number of dimensions as the numerator. Thus, the alignment index calculates the ratio of how much of the total variance from firing data organized by RT and choice is explained by the outcome subspace.

## Demixed principal component analysis (dPCA)

We used dPCA[49], a semi-supervised dimensionality reduction technique to further understand if prestimulus activity which covaried with RTs shared variance with firing rate activity that covaried with the previous trial's outcome. We performed two dPCAs. The first identified axes that maximally accounted for firing rate variability from trial outcome and the second identified axes that maximally accounted for firing rate variability that covaried with RTs. We then calculated the dot product between these axes and estimated the angle using the inverse cosine of the dot product. An angle of zero would indicate that these axes completely overlap and that their sources of variance are the same, whereas orthogonal angles would mean that the axes do not overlap and therefore share no variance. We used the freely available dPCA toolbox for this purpose (https://github.com/machenslab/dPCA).

## Models of dynamical hypotheses

**Recurrent neural network models of various dynamical hypotheses.** We created single layer recurrent neural network models corresponding to hypotheses in Fig. 1e–h using the PsychRNN tool box[86]. Each model contains 100 recurrent units, with the state activity $x$ of each unit specified by the following equation:

$$x_t = (1-\alpha)x_{t-1} + \alpha(W_{rec}r_{t-1} + W_{in}u_t + b_{rec}) + \sqrt{2\alpha N_{rec}^2}\mathcal{N}(0,1) \quad (36)$$

The firing rate of each recurrent unit is obtained by passing state activity through a relu nonlinearity:

$$r_t = \text{relu}(x_t) \quad (37)$$

Finally, the output of the network $y_t$ is given by:

$$y_t = \text{sigmoid}(W_{out}r_t + b_{out}) \quad (38)$$

The outputs of the model $y^{T\times2}$ are decision-variables for left and right choices.

For each model, we used a training set of 50,000 trials and a testing set of 5000 trials. For each trial, we simulate 500 time points $T$ with 10 ms intervals giving us a total time of 5000 ms.

For each model, the input $u^{T\times2}$ is a two-dimensional vector representing left evidence and right evidence specified by left coherence $Coh \in [0,1]$. Coh = 0 signifying pure right evidence and Coh = 1 signifying pure left evidence. For training trials, the input was given as:

$$u_{train}^{left} = \begin{cases} 0, & T <= T_c. \\ (Coh + noise), & \text{if } T > T_c. \end{cases} \quad (39)$$

$$u_{train}^{right} = \begin{cases} 0, & T <= T_c. \\ ((1 - Coh) + noise), & \text{if } T > T_c. \end{cases} \quad (40)$$

For this task, $T_c$, a random number between 750 ms and 1500 ms, represents the checkerboard onset, after which evidences for left and right choices are available to the network.

For the desired outputs $y_{train}$, a left trial has the equation,

$$y_{train}^{left} = \begin{cases} y_0, & T <= T_c. \\ 1, & \text{if } T > T_c. \end{cases} \quad (41)$$

For a right trial,

$$y_{train}^{right} = \begin{cases} y_0, & T <= T_c. \\ 1, & \text{if } T > T_c. \end{cases} \quad (42)$$

$y_0$ is the starting value, which we set to 0.2.

We used back propagation through time with a mean squared error loss function to train the networks. We assumed a decision was made once the value of either the left or the right decision variable reached the threshold of 0.7. After the decision is chosen, both inputs and outputs were reduced to their starting value.

We simulated five models with different conditions to quantitatively model the dynamical hypotheses that we proposed in Fig. 1e–h.

1. The first model (Fig. S2a), corresponding to Fig. 1e, applied the basic RNN without further modifications.
2. For the second model (Fig. S2b), corresponding to Fig. 1f, we applied a delay between 0 and 300 ms on inputs with a linear mapping to coherence.
3. For the third model (Fig. S2c), corresponding to Fig. 1g, the RNN model was trained in the same way as the basic RNN. In testing sets, a random value between 0.3 and 0.6 was added on every left input over the trial to mimic a bias in the animal towards one or the other choice.
4. For the fourth model (Fig. S2d), corresponding to Fig. 1h, the RNN model was trained in the same way as the basic RNN. In testing sets, a random value between 0 and 0.3 was added on both left and right inputs over the trial to mimic an overall impulse in the animal to make the choice.
5. For the fifth model (Fig. S2e), corresponding to Fig. 1h, a multiplicative gain was applied to state $x$:

$$r_t = \text{relu}(g_0 {}^* x_t) \quad (43)$$

After the choice was made, the gain variable $g_0$ was set to 0.

Results of RNNs are shown in Fig. S2. For each time step, RT PCA trajectories were generated using the same method as Fig. 4.

We used the same approach as used for the PMd data to regress firing rates to RT (Fig. S2, middle). We report the regression between model firing rates and RTs for left trials only. Finally, we performed choice decoding analyses by median splitting the RTs into fast and slow bins (Fig. S2, right).

**Hypothetical Synthetic Neural Populations.** We created hypothetical neurons based on the population of neurons that we observed in our PMd dataset. In a previous study, we used various metrics to categorize this continuum of neurons into increased, decreased, and perimovement neurons[18]. Increased and decreased neurons, as their name suggests had sustained increases and decreases in their firing rates after checkerboard onset. In contrast, the perimovement neurons just responded at or around the time of movement onset with a variable lead and lag relative to movement onset. We also know from a regression analysis (Fig. S17d), that roughly 20% of our population of neurons shows prestimulus covariation with RT. We used these pieces of information to build simulated neural populations. We then used dimensionality reduction, decoding, and regression analyses to make predictions for our neural data. These targeted simulations of

hypothetical synthetic populations are meant to guide interpretation of our analyses of the neural data, and complement the RNN simulations.

We created three simulations of hypothetical neural activity in PMd. In all of these simulations, post-cue firing rates covaried with choice and RT.

Simulation 1 assumed that the firing rates of the neurons had no covariation with RT or choice before cue onset (consistent with Fig. 1e). Simulation 2 assumed that the firing rates of the neurons covaried with choice before cue onset (consistent with Fig. 1g). Finally, simulation 3 assumed that prestimulus firing rates of the neurons covaried with RT but not choice (consistent with Fig. 1h).

For all three cases, we simulated the firing rate patterns of 200 increased neurons, 100 decreased neurons, and 50 neurons (transient response around movement RT) for 600 trials in total (300 left trials and 300 right trials). For each trial, RT was generated randomly from a gamma distribution:

$$RT = 200 + 100\Gamma(5, 0.5) \tag{44}$$

We assumed the checkerboard onset was at time 0, and that post-cue firing rates changed a short time period later defined by a variable ($t_{latency}$) defined as the sum of non decision-related time including stimulus encoding and motor execution, etc. $t_{latency}$ was set to 100 ms.

For all three simulations, the firing rate for each increased neuron at a given time point $t$, was a combination of a baseline firing rate ($r_{base_1}$ = 5 spikes/s), a choice-selective component, and a condition-independent signal that increased with time after cue onset and depended on RT.

The choice-selective increase in firing rates after stimulus onset was as follows: For choice 1 (chosen direction), $r_{choice_1} = 5 + 7*\mathcal{U}(0,1)$. For choice 2, $r_{choice_2} = 2 + \mathcal{U}(0,1)$. These choice-selective components were activated after checkerboard onset as a function of time by multiplying by $\max(\frac{t - t_{lag}}{RT}, 0)$ with $t_{lag}$ for trial $i$ defined as $t_{latency} + 0.2*RT_i$. For the chosen direction, $r_{choice1}$ was added to the firing rate at each time point, whereas for non-chosen direction $r_{choice2}$ was subtracted from the neuron's firing rate at each time point.

The condition-independent signal for each trial and neuron was modeled as a general stimulus-triggered increase in firing rate with $r_{time} = \max(15*(t - t_{lag}), 0)$. This condition-independent signal depends on RT based on prior work[87].

- In Simulation 1, these were the only components that altered the firing rate.
- For Simulation 2, where we hypothesized a prestimulus bias towards one or the other choice, we added a second firing rate term that was uniformly drawn from 0 to 4 spikes/s for one choice and from 0 to 2 spikes/s for the other choice.
- For Simulation 3, where we hypothesized that prestimulus firing rate covaries with RT but not choice, we added a second firing rate term derived from a uniform distribution and dependent on RT ($r_{base_2} = \frac{2 + base_{neuron}}{RT}$) for both left and right choice trials. $base_{neuron}$ is generated for each neuron from a uniform distribution between 0 and 3 spikes/s. Only 20% of neurons had this baseline covariation with RT and these neurons also have 0 baseline firing rate. This simulation is schematized in Fig. S1b.

We created 100 decreased neurons by subtracting 10 spikes/s from the firing rate of the first 100 increased neurons.

To generate the PSTHs of these simulated neurons we used the following procedure: Once we obtained a simulated firing rate $r$ for a trial, we used the time-rescaling theorem[88] to generate spike trains from a poisson process. Then we smoothed the spike train with a Gaussian kernel to generate hypothetical neuron PSTHs for each trial. We then averaged these PSTHs and

calculated PCA trajectories with the same method as in Fig. 4. Finally, we randomly selected 50 neurons from this simulated neural ensemble and performed regression analyses to understand the link between these firing rates, and RT and choice to closely match the analyses performed for the real PMd data (Fig. 6).

### Reporting summary
Further information on research design is available in the Nature Portfolio Reporting Summary linked to this article.

## Data availability
Source data are provided with this paper and can be used to recreate figures. Data that are analyzed and used for generating figures have been deposited in Dryad [https://doi.org/10.5061/dryad.9cnp5hqn0]. Raw session data that lead to the summarized data analyzed in the paper are completely available upon request. Source data are provided with this paper.

## Code availability
MATLAB and Python code and functions for generating all the main and supplementary figures are publicly available [https://github.com/chand-lab/Dynamics2023].

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

## Acknowledgements

We thank Dr. Michael Economo, Dr. Kamal Sen, Dr. Tatiana Engel, and Dr. Matt Golub for comments on previous versions of the results. C.C. was supported by a NIH/NINDS R00 award R00NS092972 (CC), NIH/NINDS R01 NS121409 (CC), NIH/NINDS NS122969 (CC), the Moorman-Simon Interdisciplinary Career Development Professorship from Boston University (CC), the Whitehall foundation (CC), and the Young Investigator Award from the Brain and Behavior Research Foundation (CC). The late Dr. KVS was supported by the following awards: NIH Director's Pioneer Award 8DP1HD075623 (KVS), NIDCD R01-DC014034 (KVS), NIDCD U01-DC017844 (KVS), NINDS UH2-NS095548 (KVS), NINDS UO1-NS098968 (KVS), DARPA-BTO 'REPAIR' Award N66001-10-C-2010 (KVS), DARPA-BTO 'NeuroFAST' award W911NF-14-2-0013 (KVS), Simons Foundation Collaboration on the Global Brain awards 325380 and 543045 (KVS), Office of Naval Research award N000141812158 (KVS), Larry and Pamela Garlick (KVS), Wu Tsai Neurosciences Institute at Stanford (KVS), the Hong Seh and Vivian W. M. Lim endowed professorship and the Howard Hughes Medical Institute (KVS). The funders had no role in study design, data collection and interpretation, or the decision to submit the work for publication.

## Author contributions

C.C. trained both monkeys and recorded in PMd using multi-contact electrodes under the mentorship of KVS. C.C. and K.V.S. developed initial hypotheses of how initial conditions and sensory evidence could be combined to drive decisions. P.B., T.W., and C.C. jointly collaborated on the various analyses. G.K. and L.C. provided helpful insights for analysis and relevant literature for the manuscript. P.B. and C.C. wrote initial drafts of the paper. All authors refined further drafts contributing analyses, insights, and writing.

## Competing interests

The late K.V.S. consulted for Neuralink Corp. and CTRL-Labs Inc. (part of Facebook Reality Labs) and was on the scientific advisory boards of MIND-X Inc., Inscopix Inc., and Heal Inc. All other authors have no competing interests. These companies provided no funding and had no role in study design, data collection, and interpretation or the decision to submit the work for publication.
