## [Peer Review File · Nature Communications]

Initial conditions combine with sensory evidence to induce decision-related dynamics in premotor cortexREVIEWER COMMENTS

Reviewer #1 (Remarks to the Author):

This paper extends previous work showing consistent relationships between reaction times and neural states in dorsal premotor cortex (PMd), prior to cue onset, during a perceptual discrimination task. The authors recorded in PMd of two monkeys performing a “reaction time” version of a 2-alternative forced-choice color discrimination task. While many results from this data set have already been published, here the authors specifically examined how a “neural space” approach may reveal features of the dynamical system that includes PMd. In particular, they examined whether the initial state of neural activity, prior to the discriminatory color checkerboard cue, predicts the choice and/or the RT of the animal, as well as the neural dynamics (speed, magnitude) of the state after the cue. They also examined how activity depends on the previous trial outcome, and whether that can explain post-error slowing, as previously reported in PMd during reaching (Thura et al. 2017) and LIP during oculomotor tasks (Purcell & Kiani 2016).

The authors report two main results as well as several auxiliary findings. First and foremost, they show that the neural state prior to the checkerboard cue robustly predicts reaction times in the task, in agreement with previous studies (Afshar et al 2011). Second, they show that the neural state does not predict choice, which is surprising given current models. In addition to these main results, they also show how post-cue dynamics depend on pre-cue state, and how the pre-cue state depends on previous trial outcome.

The paper is well written and the research is of very high-quality. The results are not very novel because the individual phenomena have previously been shown, but it is compelling to see the relationships among these phenomena analyzed together with congruent methods. I have a few major suggestions on how the manuscript could be further improved, as well as a number of smaller comments.

MAJOR COMMENTS:

1. I fully agree with the dynamical system approach, but its generality presents both advantages and challenges. Almost anything could be described as a dynamical system. In fact, equation 1 is so general that I am not sure what alternative hypothesis could not be expressed in this way just by using different settings of $F(X)$ and gains. What model are the authors really arguing against? The two hypotheses they present in Figure 1E,F are both dynamical systems models, and other types of models (e.g. drift-diffusion, etc.) could all also be expressed as a dynamical system. In the introduction, the authors state two key predictions: 1) that initial conditions predict dynamics and behavior; and 2) that they should combine with inputs. But what model does not make those predictions? The predictions are spelled out

a bit more precisely later (line 280) but again it is hard to see what model would not predict these phenomena. One way to respond to this comment would be to implement alternative hypotheses using synthetic neurons and then perform similar PCA analyses on the resulting activity patterns. But it would be best if those alternative hypotheses were not mere straw men.

2. The result that I found most surprising and interesting was the finding that the initial state of neural activity in PMd did not predict the animal's choice. Sure, for high coherence stimuli this is not surprising because any initial bias should be overcome by the strong evidence signal, but for the low, 4% coherence, one might expect some choice bias due to pre-cue activity. In almost all of the analyses, the authors combine across coherences, so such a phenomenon would be lost. One exception is the analysis on Figure 6D, where the choice selectivity signal is shown separately for different coherences, including the hard 4% trials at the bottom. I don't see any evidence of a pre-cue bias there, and I assume this is the main reason the authors have made this conclusion. However, I wonder if a more targeted examination could see something else. For example, if you limit the analysis to the fastest RT bin of the 4% coherence trials, in which performance is presumably close to chance (including of course both correct and error trials), does a bias appear? When I zoom into the plot I do see a trend for the blue lines to be above the orange, except for one, but I doubt this would pass significance. Nevertheless, a more careful and rigorous examination of this would be important, because it bears upon whether PMd is causally involved in determining the animal's choice in this task. For example, I would be curious to see a standard "population histogram" analysis of this (i.e. plot activity separately for when the monkey chose a cell's "preferred direction" versus when it chose the opposite target). It is possible that some pre-cue bias might be seen with such an approach (again, limited to the fastest RTs in the 4% trials, to exaggerate "selection bias"), but not with an analysis using PCA, which specifically focuses on components that explain the highest variance. My guess is that a population histogram will not show a significant pre-cue bias either, but it's worth doing just to be sure.

3. The high degree of overlap between the RT bins that were analyzed (with adjacent bins sharing approximately 75% of the trials) is problematic because it violates the assumptions of most standard statistical tests, which rely on independent samples. This casts doubt on many of the statistics reported here (e.g. Figure 4C,G). Furthermore, it could mask some interesting phenomena such as differing slopes in choice selectivity as a function of RT (e.g. Figure 6D). I recommend doing the analyses with fewer bins that are not overlapping, and I'm certain you have enough trials for that. Incidentally, somewhere the number of trials per RT bin should be reported, especially if they're very non-uniform.

4. Most of the analyses combine data across trials with different coherences, raising the concern that all of these RT relationships are simply due to different strengths of evidence (especially given the high degree of overlap between the RT bins). The authors lay this concern to rest by subsequent analyses that document the effects within trials of a given coherence level, but why wait? I think the within-coherence analyses should come first, making the case more persuasively, and then combining across different coherences becomes largely superfluous. But this brings me to the next point.

5. There are many different ways of doing the PCA, and it's not entirely clear to me what the authors did. Line 710 states that the trials were grouped by "coherence/RT/past outcome (... x 7/11/2)". What does that mean? Does it mean you grouped data into 154 categories? That seems unlikely as most neurons would not have data in each category. So does it mean that for some analyses (e.g. Figure 4) you just had 11 RT categories each of which mixes coherences and outcomes, while for other analyses (e.g. Figure 7) you just had 2 categories for outcomes each mixing coherences and RTs? I assume the latter because the resulting PCs are clearly different (at least swapped). If that is correct, then what was done to produce the analysis in Figure 6C and D. Did you redo the entire analyses separately using RT categories but only one level of coherence? Or did you just use the same weight matrix produced by the aggregate PCA to project data into the same PCs but separately from trials of each coherence? In any case, what was done should be spelled out explicitly.

But I wonder if a different approach could be taken (optionally): to group trials into much more general categories before performing the PCA and then project more specific trial types into that space. For example, if you defined just three RT bins and three coherence levels then that's only 9 categories for which all cells should have data. I suspect the top 6 PCs will be quite similar to what you have, and then you can still project specific trial types into their space to do comparisons of finer RT bins, each coherence level separately, etc. Perhaps you may even see the effect of previous trials because that variance might be captured by same component as the variance in RTs. Just a suggestion...

6. The analysis of the effects of previous trial outcomes is very interesting, but it needs to be done more carefully. If the animals have performance "streaks", whereby sometimes errors are common and sometimes they're rare, then most of your post-error trials will come from the bad streaks as opposed to the good streaks. This means that the post-error effects (both RT and neural) could be caused not by previous trials, but by sampling unequally from times where the animal was performing differently. One way to avoid this problem is to limit the analysis to comparing ECC (a correct trial preceded by an error and followed by a correct trial) to CCE trials (a correct trial preceded by a correct trial and followed by an error). This will reduce the number of post-correct trials in your data but those are numerous anyway, and now you'll be sure that both groups come from a similar mixture.

If I've understood correctly, the effect of previous trial outcome was shown in the space of PCs constructed using data that separated post-correct and post-error trials. Would such an effect be visible if you just plotted post-correct and post-error trials in the space of your original PCs (Figure 4)? It would be interesting to see if the state varied along PC3 or PC4. If not, then it is possible that there exist "error specific" cells in your data, which you can pull out by doing PCA on different previous-outcome groups but are lost if you combine across previous outcomes.

MINOR COMMENTS:

Line 7: This is not a correct definition of how “perceptual decision-making” is used in the field. Despite how it’s often tested in the lab, perceptual decision-making is really intended to be about perception, not action selection – e.g. about discriminating the direction of visual motion, not selecting a saccade for reporting that percept. Perhaps the tasks used didn’t ultimately address that question successfully, and instead revealed mechanisms of action selection, but it is not correct to say that the “process of choosing actions ... is termed perceptual decision-making”.

Line 11: The neural basis for what kind of decision-making? Perceptual? Economic? Emotional?

Line 40, 431, 534: In several places the authors cite work on the neural correlates of speed-accuracy tradeoffs. In these places it would be appropriate to add citations to Hanks, Kiani & Shadlen (2014) who showed these effects in LIP, and Thura & Cisek (2016) who showed them in PMd.

Line 107: Why include 195 “multi-units” in your data (including ones that are only “weakly separable from noise”, line 681) when you already have 801 well-isolated single neurons? I think a slightly smaller but cleaner data set is better, and I’m sure the results will be very similar and easier to be confident about.

Line 123: I know it’s trendy, but using the word “interrogate” in this context seems a bit silly. Is the neural activity seated under a bright lamp at a police station? ;-) How about just “test”?

Figure 3: Here and in all of the plots, there should be more tick-marks on the axes so that the reader has an easier time comparing the various magnitudes, even when the units are arbitrary. Sometimes readers might want to look at something you haven’t discussed and giving them just the minimum and maximum values is not helpful.

Figure 4: The analysis on Figure 4E shows that “adjacent” trajectories were similar, but wouldn’t that be expected given the high degree of overlap between adjacent RT bins? Again, I think using fewer bins that are non-overlapping would be more revealing of real trends and less subject to averaging artifacts.

The excursions along the different PCs are quite different in magnitude, and this is not always clear. For example, Figure 4A shows the top four PCs using the same y-scale, but Figure 4B shows the trajectories

with vastly different scales (I estimate about 20 for PC1 and 5 for PC4). This should be made clear in the plot. See my comment above about tick marks.

Line 210: I don't understand the rationale for comparing the speed of neural state change across different trajectories using the KiNeT approach. Why not just plot the magnitude of the derivative? What is the downside of this more straightforward measure?

Line 267: This sentence needs to be fixed grammatically.

Line 304: It is said that the fast RTs have a steeper slope than slow RTs. I don't see that. To me the lines all look pretty parallel, particularly for 4% trials. The change in slope is not so compelling in the 90% and 31% coherence groups either, at least for the faster RT bins.

Figure 6: Are the PCs here the same as in Figure 4?

Figure 7C: Why is there no response to the cue? I would expect to see one given PC3, above.

Line 376: I did not understand this paragraph.

Line 447: This paragraph is perhaps a bit too quick to dismiss the DDM. Indeed, a one-dimensional variable cannot explain the results shown here, but what about a 2-dimensional drift model in which the state can shift around on a plane? If the "thresholds" for initiation converge together near one end of that plane, then reaction times could be shortened if the system starts closer to that end.

Line 664: It is said that recordings were "lateral to the spur of the arcuate sulcus". Is that correct? Most people would consider that to be PMv, not PMd.

Equation 9: To what do the superscripts k and l refer? Are these just randomly chosen trials?

Line 837: Perhaps replace the "-" with "is the", so it's clearer.

Figure S3: It is stated that “Post-error choice selectivity may be larger than other trial outcomes”. What are you referring to? The slightly higher purple trace at the end? Is that significant?

Reviewer #2 (Remarks to the Author):

In this manuscript, the authors use a dynamical systems framework to interpret neural responses measured from PMd during a perceptual decision-making task. They find that their framework provides a parsimonious way to interpret results that, by themselves, might appear to be arbitrary. Specifically, a dynamical systems approach interprets the firing rates before cue onset as the “initial state”, which, along with the stimulus input, sets how the dynamics unfold on each trial. The authors show that initial state covaries strongly with reaction time and not so much with choice. The paper is well written, the data and results are clear, and the concepts well illustrated.

That said, I have three concerns:

- 1) I’m worried the results depend on the choice of smoothing.
- 2) I’m worried the analyses in this manuscript are overstating the variance explained.
- 3) I would like to see some demonstration of how good the dynamical systems framework is as a model of neural activity so that other approaches have something they can actually compare to.

Full disclaimer: I don’t use the dynamical systems framework to interpret neural activity, but I was left with a general feeling reading this: dimensionality reduction is a useful visualization tool, but is dynamical systems a good model of neural activity? If the fundamental statement here is that the activity of neural populations depends both on new inputs and the previous state of the system, what is the alternative? In other words, how could the authors build more trust in these extremely smooth trajectories they plot? How much variance in the single trial responses is explained by this approach? Or, is this just a convenient way to visualize and interpret heterogeneous PSTHs? If it is the latter, then could we not test underlying models of neural activity more directly with regression? How should people using other approaches compare to your approach?

Essentially, I would like to see some of the dynamics analyses explicitly cross-validated and I would like to see the results run with smaller amounts of smoothing (and only causal smoothing). For example, in the PCA analysis, 6 PCs explain 90% of the variance of the PSTHs... after smoothing with 30ms gaussians. That is a lot of smoothing! As a quick check, I took white noise of the similar dimensions to what was reported here (1800 x 996) and smoothed it columnwise with a gaussian (with standard deviation = 30) and did PCA. With that level of smoothing, 90% of the variance of my simulated 996 white noise neurons

were explained with 20 dimensions. 6 is a lot smaller than 996, but it's not a lot smaller than 20. So, I'm left wondering how good a model are these smooth trajectories?

The regression analysis in figure 5 is cross-validated, which is the only place in the manuscript that phrase is mentioned, and one of the appealing reasons to use regression to analyze and neural data. Why do I care? Well, the regression analyses in figure 5 tells us the main data result: firing rates before cue onset are predictive of RT, but not choice. Which isn't even true of all perceptual decision-making datasets (as the authors point out in the discussion). Something like reduced-rank regression can provide low-dimensional descriptions of neural activity which can be modeled as dependent on both inputs and previous states of the neural activity. With reduced-rank regression, it would be easy to evaluate the variance explained on withheld data. There are a lot of strengths a regression framework would have for interpreting these results that are missing from the dynamical systems framework.

So, I think, as an outsider to this approach, any good model of neural activity has to first show that it can predict neural activity. As it stands, that is missing, and this manuscript would benefit from stating that. At a minimum, it would reveal whether this framework is just an (admittedly nice) visualization tool or whether it is a useful model of neural activity.

So, why not withhold single trials from the PCA analyses and ask how much of their variance is explained? Or, even better, use the pre-stimulus firing rates plus some measure of the dynamics during the stimulus to predict responses to withheld data. Ideally, you would do matrix factorization on pseudo populations at the single trial level with something like the speckled holdout approach in Williams et al., 2018 (which has some author overlap, so I expect you could come up with something). Although, in that paper, the hold out should have blocked the speckling to the level of trials (i.e., withhold neurons for entire trials not timepoints). There is simply too much autocorrelation to hold out individual time points, which leads to bleed-through from the training set to the test set and an overestimation of the variance explained.

Basically, what I am saying is that this seems like a potential weakness of the dynamical systems framework and since this paper is mainly an argument that dynamical systems is useful as a framework here, I think it's necessary to address.

Minor:

Throughout the manuscript, I kept finding myself worrying that the results depend on that heavy smoothing with a Gaussian Kernel. Below are all the places that I was worried that things would change dramatically if you used a more modest 15ms kernel. Or a boxcar. Here are all the specific places that I was concerned about the effect of the choice of smoothing.

Line 204: There are some pretty modest p values in the results (line 204 and 208). How much does this depend on the smoothing kernel?

Line 146: As I described above. I'm skeptical of this method. 90% of the variance explained by 6 dimensions is a lot if you're thinking of actual neural dynamics. But it's not that much if you smooth responses massively. How much does this depend on the smoothing kernel? Does the number of dimensions found change dramatically for smaller or larger smoothing kernels? What does that mean for interpreting low-dimensional dynamics?

Same with line 710: How many trials go into these average firing rates typically? If I'm understanding the analysis right, it appears there are 308 ($2 \times 7 \times 11 \times 2$) unique conditions, so roughly 5 trials per condition given the average number of trials in a session. How much does the variance explained by the 1st 6 PCs depend on the smoothing window?

Line 695: is the Gaussian Kernel causal? If not, is it not problematic to look at pre-cue firing rates with such a wide acausal kernel?

Response to reviewers

Reviewer 1

Comments:

This paper extends previous work showing consistent relationships between reaction times and neural states in dorsal premotor cortex (PMd), prior to cue onset, during a perceptual discrimination task. The authors recorded in PMd of two monkeys performing a “reaction time” version of a 2-alternative forced-choice color discrimination task. While many results from this data set have already been published, here the authors specifically examined how a “neural space” approach may reveal features of the dynamical system that includes PMd. In particular, they examined whether the initial state of neural activity, prior to the discriminatory color checkerboard cue, predicts the choice and/or the RT of the animal, as well as the neural dynamics (speed, magnitude) of the state after the cue. They also examined how activity depends on the previous trial outcome, and whether that can explain post-error slowing, as previously reported in PMd during reaching (Thura et al. 2017) and LIP during oculomotor tasks (Purcell & Kiani 2016).

The authors report two main results as well as several auxiliary findings. First and foremost, they show that the neural state prior to the checkerboard cue robustly predicts reaction times in the task, in agreement with previous studies (Afshar et al 2011). Second, they show that the neural state does not predict choice, which is surprising given current models. In addition to these main results, they also show how post-cue dynamics depend on pre-cue state, and how the pre-cue state depends on previous trial outcome.

The paper is well written and the research is of very high-quality. The results are not very novel because the individual phenomena have previously been shown, but it is compelling to see the relationships among these phenomena analyzed together with congruent methods. I have a few major suggestions on how the manuscript could be further improved, as well as a number of smaller comments.

Response:

Thank you for your positive evaluation on 1) the quality of the work, and 2) the writing. As your comments highlight, perceptual decision-making is a remarkably well-studied field. Some of the insights in the paper are described previously by us and others or indirectly elsewhere resulting in a patchy understanding of how decisions emerge from neural activity.

Our goal when we embarked on this study was to use approaches from machine learning and electrical engineering to subsume various results and help build the next generation of experimental, analytical and modeling studies of perceptual decision-making. We are delighted that the results feel compelling and meaningful to you.

We are also very grateful for your close read of the manuscript and the associated comments. We now respond to them in considerable detail. Responding to these comments has considerably strengthened our manuscript. We particularly like your encouragement to be less trendy, more rigorous and straightforward in our writing and for pointing out where our analyses could be improved.

R1.1 —

I fully agree with the dynamical system approach, but its generality presents both advantages and challenges. Almost anything could be described as a dynamical system. In fact, equation 1 is so general that I am not sure what alternative hypothesis could not be expressed in this way just by using different settings of $F(X)$ and gains. What model are the authors really arguing against? The two hypotheses they present in Figure 1E,F are both dynamical systems models, and other types of models (e.g. drift-diffusion, etc.) could all also be expressed as a dynamical system. In the introduction, the authors state two key predictions: 1) that initial conditions predict dynamics and behavior; and 2) that they should combine

with inputs. But what model does not make those predictions? The predictions are spelled out a bit more precisely later (line 280) but again it is hard to see what model would not predict these phenomena. One way to respond to this comment would be to implement alternative hypotheses using synthetic neurons and then perform similar PCA analyses on the resulting activity patterns. But it would be best if those alternative hypotheses were not mere straw men.

Response:

We thank you for this comment. We see the issue with the manuscript as written. In some senses, arguing for a dynamical system is arguing for a *truism* (although there is still disagreement, see Reviewer 2's comments). Therefore, this line of reasoning could be considered as not particularly insightful or original. In our original manuscript, we had written how we are investigating if a dynamical systems perspective could help understand decisions. As you point out with enough reasoning it becomes apparent that it has to be this case. The more specific question is *what is the form of the dynamics*.

To address this issue, we have made two considerable changes to the manuscript.

- First, we now have edited both the abstract and the introduction to we say we are using the dynamical systems perspective as our overall approach. More pointedly we state that we are *arbitrating between specific models of decision-related dynamics*. Of course, the space of such models is quite large. Fortunately, we can constrain this space based upon results from the rest of the literature and the behavior of our monkeys to define specific dynamical models. We have now considered 4 different models of dynamics, which we enumerate below. We have also updated the relevant schematics in Fig. 1 to reflect this change and also edited the introduction to describe these dynamical models.
 1. Our first is a classical model of decision-making where there is no effect of initial conditions, and all the effects are due to differences in the strength of the sensory evidence (Fig. 1E). The testable prediction is that prestimulus firing rates should not correlate with any behavioral variables such as choice or RT. This dynamical model is not a straw man as we find that many researchers invoke such models to model behavior (usually mean RTs and accuracy). This model is similar to the classical drift diffusion model that is still being invoked as a model of decision-making data and neural activity (Steinemann et al., 2022).
 2. Our second dynamical model assumes that the onset of the sensory evidence depends on the coherence of the checkerboard (Fig. 1F). This model is based on studies that show that changing parameters of visual input like contrast leads to delay in responses in sensory areas (Cook and Maunsell, 2002; Oram, 2010). A closely related model assumes that the onset of the checkerboard is random and that leads to effects observed here (similar to random non-decision time assumed in some models of decision-making (Ratcliff et al., 2016)).
 3. Our third model assumes that the initial condition can be biased towards one or the other choice (Fig. 1G). In this case, prestimulus firing rates should correlate with choice but not RT. This is based for instance on data from LIP in fixed-duration decision-making tasks (Shadlen and Newsome, 2001).
 4. Finally, for our fourth model we set up a dynamical mechanism where initial conditions correlate with RT but not choice (Fig. 1H). Such a model is based on 1) studies of reach planning to single targets (Afshar et al., 2011), and 2) studies that argue that unspecific urgency-like signals are particularly important for decisions (Hanks et al., 2014; Thura et al., 2012). The additional prediction made is that fluctuations in initial conditions are dependent on trial outcome (Purcell and Kiani, 2016; Thura and Cisek, 2016).

In the revised introduction, we first present the 4 neural dynamics models for how the dynamics could relate to decision-making.

Here, we expanded on these findings from motor planning and timing studies and investigated which dynamical system best described decision-related neural population activity in dorsal premotor cortex (PMd). To derive hypotheses about dynamics, we leveraged three results from prior studies. First, rate at which choice-selective activity emerges depends on the strength of the sensory evidence (e.g., auditory pulses, random dot motion, static red-green checkerboards, etc.) (Chandrasekaran et al., 2017; Coallier et al., 2015; Hanks et al., 2015; Roitman and Shadlen, 2002). Second, in studies of speed-accuracy tradeoff, prestimulus neural activity is different for fast vs. slow blocks (Bogacz et al., 2010; Hanks et al., 2014; Murphy et al., 2016; Thura et al., 2022; Thura and Cisek, 2016) (Fig. 1C). Third, and finally, the prestimulus firing rates are altered by the outcome of the previous trial (Purcell and Kiani, 2016; Thura et al., 2017) (Fig. 1D). Based on these findings, we hypothesized four different dynamical mechanisms that could describe the data.

- The simplest dynamical system assumes initial conditions do not covary with RT or choice and that neural dynamics and behavior are driven largely by the sensory evidence (Fig. 1E).
- A second dynamical system assumes that the initial conditions do not vary, but that there are either systematic or random delays in sensory evidence processing (Oram, 2010), that alter choice-related dynamics and behavior (Fig. 1F).
- A third system assumes that initial conditions are biased towards one or another choice, correlate with RT, and that poststimulus dynamics are influenced by both initial conditions and sensory evidence (Fig. 1G).
- Finally, a fourth system assumes that initial conditions correlate with RT but not choice, and poststimulus dynamics depend on both sensory evidence and initial condition. Additionally, the changes in initial condition are in part due to the outcome of the previous trial (Fig. 1H).

- Second, we have taken on your suggestion to simulate synthetic populations with various properties. We did this in two ways.

1. First, we trained recurrent neural networks (RNNs) with various parameters and performed PCA on the firing rates of the recurrent units (Fig. S2). We show that a classical decision-making RNN (Fig. S2A, consistent with our first dynamical model, Fig. 1E) and input-delayed RNNs (Fig. S2B, consistent with our second dynamical model, Fig. 1F) do not show prestimulus covariation with RT. An RNN with an initial left-choice bias (Fig. S2C), consistent with our third dynamical model (Fig. 1G), demonstrated prestimulus firing rate covariation with RT and choice decodability prior to stimulus onset. Finally, to model our last dynamical model (Fig. 1H), we developed RNNs with either strong shared inputs before checkerboard onset (Fig. S2D) or with a multiplicative gain term on the nonlinearity (Fig. S2E). These RNNs showed prestimulus covariation with RT but only modest to little choice decodability prior to stimulus onset (Fig. S2D, E).
2. In a second complementary approach, we simulated synthetic neural populations (Fig. S3) based on Chandrasekaran et al. (2017) that characterized single neuron responses in PMd using the same data. In that study, we showed that PMd neurons either increased their firing rates, decreased their firing rates, or just responded at the time of movement onset. We built neural populations using this framework and either modulated the baseline responses to be unbiased (Fig. S3A), choice biased (Fig. S3B), or RT biased (Fig. S3C). We find distinct sets of PCA trajectories, and regression and decoder results for each of the synthetically derived populations that fit their particular biases (e.g., prestimulus variation with RT bins for the RT-biased population but no choice bias, Fig. S3C). We use the results from these synthetically-derived populations as predictions for our analyses (i.e., PCA trajectories, regressions, and decoders).

Altogether, the synthetic neuron populations and RNN models improve the quality of the manuscript and further our conclusions as they suggest specific mechanisms that could be driving the dynamics. Based on the simulations and the extensive analyses in our manuscript, neural activity in PMd is most consistent with a dynamical system where the initial condition covaries with RT but not choice (Fig. 1H & Fig. S3C) and this system could be driven by shared inputs that speed up the decision-making without altering choice-related dynamics before stimulus onset (Fig. S2D), or a multiplicative gain signal on the recurrent dynamics in PMd (Fig. S2E).

After presenting the dynamical models, we now allude to modeling results (see below) which we refer to as a means of adding further context to our results.

We used these different candidate dynamical mechanisms to build recurrent neural networks with various constraints (Fig. S2) and simulate synthetic neural populations (Fig. S3). We analyzed these simulations of neural activity using dimensionality reduction, decoding, and regression analyses. These different dynamical mechanisms make distinct predictions about the principal component trajectories and whether prestimulus activity covaries with choice and RT. We used the predictions to analyze the firing rates of neurons recorded in PMd of monkeys performing a red-green RT perceptual decision-making task (Chandrasekaran et al., 2017).

R1.2 —

The result that I found most surprising and interesting was the finding that the initial state of neural activity in PMd did not predict the animal's choice. Sure, for high coherence stimuli this is not surprising because any initial bias should be overcome by the strong evidence signal, but for the low, 4% coherence, one might expect some choice bias due to pre-cue activity. In almost all of the analyses, the authors combine across coherences, so such a phenomenon would be lost. One exception is the analysis on Figure 6D, where the choice selectivity signal is shown separately for different coherences, including the hard 4% trials at the bottom. I don't see any evidence of a pre-cue bias there, and I assume this is the main reason the authors have made this conclusion. However, I wonder if a more targeted examination could see something else. For example, if you limit the analysis to the fastest RT bin of the 4% coherence trials, in which performance is presumably close to chance (including of course both correct and error trials), does a bias appear? When I zoom into the plot I do see a trend for the blue lines to be above the orange, except for one, but I doubt this would pass significance. Nevertheless, a more careful and rigorous examination of this would be important, because it bears upon whether PMd is causally involved in determining the animal's choice in this task. For example, I would be curious to see a standard "population histogram" analysis of this (i.e. plot activity separately for when the monkey chose a cell's "preferred direction" versus when it chose the opposite target). It is possible that some pre-cue bias might be seen with such an approach (again, limited to the fastest RTs in the 4% trials, to exaggerate "selection bias"), but not with an analysis using PCA, which specifically focuses on components that explain the highest variance. My guess is that a population histogram will not show a significant pre-cue bias either, but it's worth doing just to be sure.

Response:

Thank you for this comment. We were also surprised by this finding. We had previously performed regression analyses to examine how neural activity correlates with choice and RT at the single neuron level. These findings were also in the back of our mind when we made the statement. We did not want to overload our previously submitted manuscript and chose not to include the single neuron regression data and instead focused more on the population data. Here we include two other analyses that should bolster these conclusions.

Decoding as a function of coherence

In our original manuscript, we included a supplementary figure (Fig. S16C), where we performed decoding analyses as a function of RT. The potential confound in this analysis is that we are including all of the stimulus difficulties and thus if there were subtle effects of choice modulation before stimulus onset for the hardest coherences, this effect would be missed.

We now replicated this decoding analysis using three RT bins and restricted trials to just the easiest and hardest coherences (Fig. S17A, B). Again, we find that prestimulus firing rates fail to predict choice in these decoding analyses.

Single neuron analyses

Our recordings were performed using linear multi-contact electrodes, so we could not really optimize to place neurons in the preferred vs. nonpreferred locations because the tuning can be heterogeneous and some neurons are tuned for left and the other for right. So we are not really sure how best to do a preferred vs. non-preferred analysis.

Nevertheless, we now performed a closely related single-neuron analysis to further address this open question. We performed a regression analysis where we examined if prestimulus firing rates were explained by choice for just the hardest coherences in the fastest and slowest RT bins (blue and red lines respectively, Fig. S17C). Both bins are non-overlapping (< 450 ms and > 450 ms). This regression analysis measures if there is a significant β_{Choice} for each time point. Significance was measured by examining if the 99% ($p=0.01$) confidence intervals for β_{Choice} overlapped with 0. By chance, we should expect $\sim 1\%$ of units to have a significant correlation with choice, and that is exactly what we see for both the fastest and slowest RTs for the hardest 4% coherence. None of these percentages would survive a reasonable multiple comparisons correction.

As a comparison, we also include the percent of units whose firing rates can be explained by RT for the hardest coherence (yellow line, Fig. S17C). In contrast to the choice analyses, at least 5% of neurons in our dataset show significant betas for RT before stimulus onset for the hardest coherence (Fig. S17C). Thus, the single neuron regression analysis strongly suggests that there is little to no covariation with choice before stimulus onset for just the hardest coherences and the fastest and slowest RTs. These results replicate even when pooled across all coherences as only $\sim 1\%$ of neurons demonstrate choice signals, but $\sim 20\%$ of neurons covary with RT in the pre-cue period (Fig. S17D).

The nature of null hypothesis significance testing means that we cannot assert the null that there is no choice bias in the data before stimulus onset. Nevertheless, it appears that the effect if it exists is far weaker than the the prestimulus relationship with RT. In our opinion, the conservative conclusion is that there is minimal prestimulus covariation with choice but strong covariation with RT. We include the following text in the relevant results section.

We further explored whether there was a prestimulus bias for faster RT bins (apparent larger prestimulus separation by choice for faster RT bins, Fig. 4B) or harder coherences as prestimulus activity has been found to be predictive of choice for harder coherences in previous experiments (e.g., Shadlen and Newsome, 2001). For one, prestimulus spiking activity was no better than chance at predicting eventual choice even when trials were grouped by RT bins (Fig. S16C). Next, we further refined this analysis by performing a decoding analysis where we restricted the analysis to just the hardest coherence and the fastest and slowest RT bins (Fig. S17B). Again we found no relationship between prestimulus neural activity and choice for any of the RT bins. Results were similar even when we restricted the trials to just the easiest coherence (Fig. S17A). Second, we also performed a simple regression analysis (50 ms causal bins stepped by 1 ms) where we examined if neural activity covaried with choice on a neuron-by-neuron basis for just the fastest and slowest RTs for the hardest coherences, and compared it to the percent of neurons that covaried with RT (Fig. S17C). We found that percent of neurons that covaried with choice before stimulus onset was largely at chance levels, whereas a modest (~5%) but significant portion of neurons correlated with RT even before stimulus onset. Including all coherences in this regression also did not change the results — again ~20% of neurons covary with RT before stimulus onset but only ~1% of units covary with choice (Fig. S17D). Thus, we found further evidence of the neural population covarying with RT before stimulus onset but did not observe any prestimulus choice bias.

Causal Role of PMd?

“Nevertheless, a more careful and rigorous examination of this would be important, because it bears upon whether PMd is causally involved in determining the animal’s choice in this task”.

This statement is thought provoking. It is an open question what PMd’s causal role is in “computing” the choice signals vs. amplifying choice inputs. In other experiments in the lab, we have been recording in the dorsolateral prefrontal cortex (DLPFC) in monkeys performing this task and compared them to the responses in PMd. PMd neurons, as reported in this paper, only covary with RT and choice. In contrast, in DLPFC, we see encoding of target configuration, color choice, and action choice during this task (Reviewer Fig. 3). Therefore, we believe, that the choice signals we observe in PMd are actually inherited from DLPFC (and other upstream areas) which combine with RT-related dynamics in PMd leading to the eventual choice. Of course, we need simultaneous recordings in PMd, DLPFC, and likely the supplementary motor area to fully address this question. These experimental data are also very consistent with our multi-area recurrent neural network model of the same task that shows that PMd likely inherits action choice signals from an upstream area and leads to the decision-making behavior (Kleinman et al., 2019). Note these results from **DLPFC should not be construed as PMd being less causally involved in these tasks**. In our RNN model (Kleinman et al., 2019), any perturbations of the analogue of PMd leads to alterations in network output and choice biases.

R1.3 —

The high degree of overlap between the RT bins that were analyzed (with adjacent bins sharing approximately 75% of the trials) is problematic because it violates the assumptions of most standard statistical tests, which rely on independent samples. This casts doubt on many of the statistics reported here (e.g. Figure 4C,G). Furthermore, it could mask some interesting phenomena such as differing slopes in choice selectivity as a function of RT (e.g. Figure 6D). I recommend doing the analyses with fewer bins that are not overlapping, and I’m certain you have enough trials for that. Incidentally, somewhere the number of trials per RT bin should be reported, especially if they’re very non-uniform.

Response:

Thank you for this comment. We used the overlapping bins because we felt it gave us a way to understand how the data changed smoothly across RTs. However, we understand the statistical issue and have performed the analysis with three non-overlapping bins. As expected none of the results materially change and they are

included as a supplementary figure (Fig. 5).

We have also included a supplementary figure where we report the number of trials used for each RT bin for both the overlapping and non-overlapping RT bin cases. They are weakly non-uniform (Fig. S6).

We have now included the following text in the manuscript in Section 2.4.

On average, we used 100 to 200 trials per RT bin for these analyses (Fig. S6).

As well as this text in Section 2.4.

These analyses are also robust to whether they are performed with multi-units and single units (Trautmann et al., 2019) (996 units, Fig. 4) as compared to solely well-isolated single neurons (801 single neurons, Fig. S10A), whether the RT bins overlap or not (Fig. 5A), and are not dependent on the smoothing used to produce the firing rates (Fig. S11A).

Note, we shared your caution about overlapping RT bins and differing trial counts possibly leading to spurious conclusions. This was further motivation for including decoding and regression analyses. These analyses operate at single-trial resolution and reassuringly provide the same conclusions (Fig. 6 and Fig. S16). Thus, our PCA results are fully corroborated by regression and decoding analyses.

R1.4 —

Most of the analyses combine data across trials with different coherences, raising the concern that all of these RT relationships are simply due to different strengths of evidence (especially given the high degree of overlap between the RT bins). The authors lay this concern to rest by subsequent analyses that document the effects within trials of a given coherence level, but why wait? I think the within-coherence analyses should come first, making the case more persuasively, and then combining across different coherences becomes largely superfluous. But this brings me to the next point.

Response:

Thank you for this comment. We were also cognizant that the firing rates in PMd are jointly dependent on both RT and stimulus difficulty which is why we addressed it early in the original manuscript (see slightly modified text below). To more thoroughly address the concern of mixed effects from RT bins and coherences upfront we reworked Fig. 4B, which now includes an inset which parenthetically shows within-coherence state space trajectories.

Such covariation between prestimulus neural state and RT was not a result of pooling across all the different stimulus difficulties and was even observed within a level of stimulus coherence (note similarities between state space trajectories in Fig. 4B & its *inset*). We discuss this further in section 2.8 where we analyze the joint effects of inputs and initial conditions.

R1.5 —

There are many different ways of doing the PCA, and it's not entirely clear to me what the authors did. Line 710 states that the trials were grouped by "coherence/RT/past outcome (... x 7/11/2)". What does that mean? Does it mean you grouped data into 154 categories? That seems unlikely as most neurons

would not have data in each category. So does it mean that for some analyses (e.g. Figure 4) you just had 11 RT categories each of which mixes coherences and outcomes, while for other analyses (e.g. Figure 7) you just had 2 categories for outcomes each mixing coherences and RTs? I assume the latter because the resulting PCs are clearly different (at least swapped). If that is correct, then what was done to produce the analysis in Figure 6C and D. Did you redo the entire analyses separately using RT categories but only one level of coherence? Or did you just use the same weight matrix produced by the aggregate PCA to project data into the same PCs but separately from trial of each coherence? In any case, what was done should be spelled out explicitly.

Response:

Thank you for highlighting this issue. We clearly were not explicit enough in these methodological details. Indeed, we did not perform an analysis with 154 conditions (7x11x2, seven coherences, 11 RT bins, and 2 choices), and, your latter understanding of the analysis is correct. For Fig. 4A, B we created a firing rate matrix with 11 RT bins and 2 choices (22 conditions), pooled across all coherences and outcomes. Similarly for Fig. 8A, B we created a firing rate matrix organized by 4 outcomes and 2 choices (8 conditions), pooled across all coherences and RTs. Then for Fig. 7A we we created a firing rate matrix organized by 7 coherences and 2 choices (14 conditions), pooled across all RTs and outcomes.

Finally, for Fig. 7C, again your latter suggestion is how we performed the analysis. We first separated trials by coherence and trial-averaged them by 11 RT bins and two choices producing 7 matrices, 1 for each coherence. We then projected each of these within-coherence firing rate matrices into the PC space of the aggregate, coherence-pooled, 11 RT bins x 2 choices to create the PC plots for the within coherence analysis (Fig. 7C).

We realize the language we used in the manuscript was imprecise. We have now modified the language, starting on line 870, clarifying what was done per PCA as well as how the within-coherence PCA was performed. We also corrected the number that was previously reported for past outcomes, from 2 to 4 types of trials. Please see changes below.

Typical matrix organization was windowed firing rate x units x reach x condition (C) ($\sim 1800 \times 996 \times 2 \times C$). Condition could be coherence (7, i.e., Fig. 7A), RT (11, i.e., Fig. 4B), or past outcome (4, i.e., Fig. 8B).

...

To perform the within-coherence RT PCAs (3 shown, Fig. 7C) we first generated seven (1 for each coherence) trial-averaged firing rate matrices organized by RT and choice (each 11 RT bins x 2 choices). Then we projected each of these 7 matrices into the PC space obtained by PCA on the RT and choice data (i.e., Fig. 4)

R1.6 — But I wonder if a different approach could be taken (optionally): to group trials into much more general categories before performing the PCA and then project more specific trial types into that space. For example, if you defined just three RT bins and three coherence levels then that's only 9 categories for which all cells should have data. I suspect the top 6 PCs will be quite similar to what you have, and then you can still project specific trial types into their space to do comparisons of finer RT bins, each coherence level separately, etc. Perhaps you may even see the effect of previous trials because that variance might be captured by same component as the variance in RTs. Just a suggestion...

Response:

Thank you for this suggestion. We performed the analysis as you suggested, with a 3 coherence bins by 3 RT bins, 9 category design. As you suspect, the top 6 PCs (4 shown below, Reviewer Fig. 1) and the trajectories for the first, 3rd and 4th PCs are very similar to what is demonstrated in Fig. 4.

Figure 1: **Combined coherence and RT PCA recapitulates previous findings**

(A) The first four PCs ($PC_{1,2,3,4}$) of trial averaged firing rates organized across 3 RT bins (violet - fastest bin to orange - slowest bin and 3 coherence bins from easiest to hardest (darkest opacity to lightest), both reach directions (right - dashed lines, left - solid lines), and aligned to checkerboard onset. Variance explained is indicated at the top of each plot. (B) State space trajectories of the 1st, 3rd and 4th PCs ($PC_{1,3,4}$) aligned to checkerboard onset (red dots). Prestimulus neural activity robustly separates as a function of RT bin. Diamonds and squares, color matched to their respective trajectories, indicate 250 ms post-checkerboard onset and 30 ms time steps respectively.

In fact, we considered this type of analysis leading up to the creation of Fig. 7. However we decided not to undertake this approach as it largely recapitulates what is already demonstrated in Fig. 4. Thus we preferred the analysis performed in Figure 7 as it captures the interplay between different levels of coherence and RT bins while also presenting original results. For example, the latency of choice selective signals is largely flat across all RT bins for the easiest coherence, but differences in latency emerge across RT bins for the hardest coherence (Fig. 7D, F).

As for projecting specific trial types into this space (i.e., finer RT bins, separate coherence levels and trial outcomes) we believe our visualizations and analyses generally address these concerns. For one, the constructed space above (Reviewer Fig. 1) already projects RT and coherence data together. Given that we found a 77% variance overlap between differently organized spaces (the 'outcome space' and 'RT space'; section 2.10, relevant text below), it's highly likely we would observe even more overlap if we projected finer RT bins or individual coherences into a space already organized by RT bins and coherences. Thus due to the high overlap in variance that would occur in this case, we do not think that projecting firing rate data from finer RT bins or individual levels of coherence into this space would be materially different from Reviewer Fig. 1 or Fig. 4, which already has many fine overlapping RT bins. As a case in point, when projecting firing rates into a highly overlapping space (projecting outcome organized firing rates into RT space and vice versa, Fig. S20B, C) the resulting figures look remarkably similar to their original counterparts (Fig. 4A & Fig. 8A).

To quantify the strength of this overlap we used a previously developed alignment index (Elsayed et al., 2016). Briefly, the index calculates the trace of the matrix that results from the projection of the RT space onto the first six principal components of the outcome subspace (i.e., sum of eigenvalues) and divides this by the sum of the eigenvalues from the PCA on firing rates organized by RT and choice (further detailed in the methods 4.22). Thus the index, as used here, quantifies the amount of variance in the RT space (Fig. 4A, B) that could be accounted for by the outcome subspace (Fig. 8A, B). This analysis revealed that ~77% of the total variance for the RT space was explained by the outcome subspace, suggesting that the previous trial's outcome has a large impact in explaining prestimulus firing rate covariation with RT.

R1.7 —

The analysis of the effects of previous trial outcomes is very interesting, but it needs to be done more carefully. If the animals have performance “streaks”, whereby sometimes errors are common and sometimes they're rare, then most of your post-error trials will come from the bad streaks as opposed to the good streaks. This means that the post-error effects (both RT and neural) could be caused not by previous trials, but by sampling unequally from times where the animal was performing differently. One way to avoid this problem is to limit the analysis to comparing ECC (a correct trial preceded by an error and followed by a correct trial) to CCE trials (a correct trial preceded by a correct trial and followed by an error). This will reduce the number of post-correct trials in your data but those are numerous anyway, and now you'll be sure that both groups come from a similar mixture.

Response:

Thank you for this comment (FYI we split it up across R1.7 and R1.8). We first examined if the animals actually had streaks in performance. These two monkeys were well trained and did not have dramatic dips in performance, thus errors were already relatively rare. ~11% of trials across both monkeys were errors and the errors were roughly equally distributed across a session (Fig. S18A). Thus, it is unlikely that error streaks are driving the effects we see in our analysis (added text as below).

We did not observe any error streaks (Fig. S18A).

Even when we performed our original analyses, we were sensitive to this issue of performance streaks biasing results from post-error slowing (Dutilh et al., 2012). This is why in our original analysis, the majority of our post-error and post-correct trials (~78%) came from the pattern “CCEC” to ensure that we are assessing local differences. Once we had these trials, we augmented the dataset by searching the data for any remaining EC sequences and generally paired them with the closest prior CC sequence (~22% of trials, lines 471-474 in the manuscript and copied below in italics). We included paired EC, CC sequences to improve the number of trials for the trial-averaged neural data.

We examined if post-outcome adjustment was present in the behavior of our monkeys. We identified all error, correct (EC) sequences and compared them to correct, correct (CC) sequences. The majority of the data are from sequences of the form “CCEC” (78%). We compared any remaining EC sequence to the nearest CC sequence, either before or after the EC sequence (22%).

As suggested we also repeated the neural analyses again with the suggested format of searching for CCE and ECC sequences. The behavioral (Fig. S19C) and neural analyses (Fig. S19) with ECC and CCE trials look nearly identical to the analyses with CCEC, and CC and EC trials because the trials used in these two analyses are highly overlapping (~90%).

We included the following text in the paper:

Results were near identical when we used ‘CCE’ and ‘ECC’ sequences (Fig. S19)

While repeating the analysis, we also realized a minor mistake in our original analysis of the behavior that was reported in the paper. Originally, the RT plot was created from all available trials in all sessions. However not all trials from a session were used in neural analyses. This happened for two reasons. First, trials were sometimes used for searching for neural activity especially with single electrode recordings. For the V-probe recordings, we often finalized the online spike sorting while the animals were performing behavior. We have now updated the post-outcome adjustment RT plot (Fig. S18B) and reported statistical analyses (see changes below) such that it only includes trials where neural activity was simultaneously recorded. There is no change to conclusions.

We found that correct trials following an error were significantly slower than correct trials following a correct trial (Mean \pm SD: 487 \pm 129 ms, 446 \pm 96 ms; Wilcoxon rank sum comparing median RTs, $p = 2.23 \times 10^{-308}$, Fig. S18A). Additionally, we found that correct trials following a correct trial were modestly faster than the correct trial that preceded it (M \pm SD: 446 \pm 96 ms, 451 \pm 105 ms; Wilcoxon rank sum comparing median RTs, $p = 1.81 \times 10^{-4}$, Fig. S18A).

R1.8 —

If I’ve understood correctly, the effect of previous trial outcome was shown in the space of PCs constructed using data that separated post-correct and post-error trials. Would such an effect be visible if you just plotted post-correct and post-error trials in the space of your original PCs (Figure 4)? It would be interesting to see if the state varied along PC3 or PC4. If not, then it is possible that there exist “error specific” cells in your data, which you can pull out by doing PCA on different previous-outcome groups but are lost if you combine across previous outcomes.

Response:

Thank you for this comment. You have correctly summarized our analysis. There are two analyses in the original manuscript that begin to address this question.

First, we calculated a modified alignment index developed in (Elsayed et al., 2016) that calculates whether the outcome dimensions and RT/choice related dimensions were orthogonal or aligned.

$$A = \frac{\text{tr}(D_{outcome}^T C_{RT} D_{outcome})}{\sum_{i=1}^{996} \sigma_{RT}(i)} \quad (1)$$

In this analysis, if the dimensions or alternatively eigenvectors ($D_{outcome}$) associated with outcome are orthogonal to the dimensions associated with RT/choice, then the numerator would be 0 and the alignment index would be 0 %. In contrast, if the dimensions for outcome/choice are aligned with those associated with RT/choice, then the index would be closer to 100%. The unit of this index is the percent variance explained by the first 6 dimensions of the outcome eigenvectors for the firing rates organized by choice and RT. We found that 77% of the RT and choice variance was explained by the first 6 dimensions of the outcome subspace. Thus the dimensions associated with outcome and choice, are closely aligned to the dimensions associated with RT and choice.

One issue with this analysis is that perhaps this alignment index is primarily driven by the choice-related dynamics.

We therefore performed an alternative analysis where we used dPCA to identify the axis that maximally separated RT and the axis that maximally separated firing rates by previous trial outcome in the prestimulus period. If these axes are aligned, then the angle between these axes should be less than 90° . Consistent with this prediction, the angle between these axes was 47.8° , and significantly less than 90° .

Additionally, we performed the suggested analysis and projected the firing rates organized by outcome and choice into the space organized by RTs and choice and vice versa (Fig. S20). As there was large overlap between these two spaces (77%), the projected outcome and choice data (Fig. S20B) look nearly identical to the original PCA (Fig. 8, Fig. S19A) and the states still vary along PC4.

We also plotted a scatterplot and a biplot of the loadings on PC 1 and PC 4 to examine if there were any relationships between the PCs that would suggest that a subpopulation of ‘outcome’ cells had the strongest contributions to PC 4 (Reviewer Fig. 2). Both the scatterplot and biplot suggest that outcome-related signals are distributed throughout the population. In other words, there is no clear correlation or pattern in either plot that suggests there is a specific subpopulation encoding outcome.

Figure 2: PC loadings do not reveal ‘outcome’-specific subpopulation of units

(A) Scatterplot of the loadings on the 4th PC and the 1st PC. (B) Biplot of loadings on 4th PC and 1st PC.

Minor Comments:

R1.m1 —

Line 7: This is not a correct definition of how “perceptual decision-making” is used in the field. Despite how it’s often tested in the lab, perceptual decision-making is really intended to be about perception, not action selection – e.g. about discriminating the direction of visual motion, not selecting a saccade for reporting that percept. Perhaps the tasks used didn’t ultimately address that question successfully, and instead revealed mechanisms of action selection, but it is not correct to say that the “process of choosing actions . . . is termed perceptual decision-making”.

Response:

Thank you for this comment. We can appreciate the importance of being more precise in our definition. In this task the monkeys must deliberate on the red-green checkerboard and choose red or green and then make the motor action to select the appropriate target. We have changed the definition on what is now line 6 as follows and are open to amending it further if the reviewer feels we are being particularly dense.

This process of discriminating sensory cues to arrive at a choice is termed perceptual decision-making (Brody and Hanks, 2016; Brunton et al., 2013; Cisek, 2012; Gold and Shadlen, 2007; Kiani et al., 2013)

R1.m2 —

Line 11: The neural basis for what kind of decision-making? Perceptual? Economic? Emotional?

Response:

We have added “perceptual” on what is now line 9 to clarify the type of decision-making.

It now reads as follows:

Research in invertebrates (Briggman et al., 2005; Kato et al., 2015), rodents (Guo et al., 2014; Hanks et al., 2015), monkeys (Churchland et al., 2008; Roitman and Shadlen, 2002), and humans (Kelly and O’Connell, 2013; Pereira et al., 2021) has attempted to understand the neural basis of perceptual decision-making.

R1.m3 —

Line 40, 431, 534: In several places the authors cite work on the neural correlates of speed-accuracy tradeoffs. In these places it would be appropriate to add citations to Hanks, Kiani & Shadlen (2014) who showed these effects in LIP, and Thura & Cisek (2016) who showed them in PMd.

Response:

Thank you for pointing us to these helpful citations. Citations were added to the text previously starting at lines 40 (now: 32), 431 (now: 595), and 534 (now: 692) (respectively):

Second, in studies of speed-accuracy tradeoff, prestimulus neural activity is different for fast vs. slow blocks (Bogacz et al., 2010; Hanks et al., 2014; Murphy et al., 2016; Thura et al., 2022; Thura and Cisek, 2016).

Finally, this dynamical system naturally bridges previously disparate findings from studies of speed-accuracy tradeoff (Bogacz et al., 2010; Hanks et al., 2014; Heitz and Schall, 2012; Murphy et al., 2016; Thura and Cisek, 2016) ...

In fact, differences in baseline neural activity between speed and accuracy conditions of speed-accuracy tradeoff tasks is found in frontal eye field (Heitz and Schall, 2012), pre-supplementary motor area (Bogacz et al., 2010), M1 & PMd (Thura et al., 2022; Thura and Cisek, 2016), and LIP (Hanks et al., 2014).

R1.m4 —

Line 107: Why include 195 “multi-units” in your data (including ones that are only “weakly separable from noise”, line 681) when you already have 801 well-isolated single neurons? I think a slightly smaller but cleaner data set is better, and I’m sure the results will be very similar and easier to be confident about.

Response:

We included multiunits in our data set for two reasons. First, Trautmann et al. (2019) has shown that spike-sorting is not strictly necessary for dimensionality reduction. More specifically, inclusion of multiunits in data analyses does not distort the recovery of low-dimensional trajectories. In fact, thresholded spiking activity alone is sufficient for recovering low-dimensional structure (Trautmann et al., 2019). Our multiunits were typically less separated from the noise, and more often than not were mixtures of two neurons and thus were good candidates for inclusion in our analyses.

Second, multi-units were included in regression and decoder analyses to improve the power of the analyses as these analyses tended to have lower numbers of units (i.e., 32 is the maximum number of units in any regression). Thus given that multi-units were included in the decoders and regressions anyways and that their inclusion in PCAs would not distort the analysis, we felt it was appropriate to include them as part of all our neural analyses.

Finally, we have repeated several of the major analyses in Fig. 4 with just the 801 single units (Fig. S10) and found that they recapitulate the main findings.

We have added the following text to the manuscript addressing the use of multi- and single units in our data analyses.

We included multiunits as well because they gave us additional power for our decoding analyses and prior work has shown that the inclusion of multiunits does not distort recovery of low-dimensional dynamics from neural activity (Trautmann et al., 2019).

These analyses are also robust to whether they are performed with multi-units and single units (996 units, Fig. 4, Trautmann et al., 2019) as compared to solely well-isolated single neurons (801 single units, Fig. S10A), whether the RT bins overlap or not (Fig. 5A), and are not dependent on the smoothing used to produce the firing rates (Fig. S11A).

Again, all of the KiNeT results were replicated even if we only 1) used single units for our analyses (Fig. S10B-E), 2) restricted the data to 3 nonoverlapping RT bins (Fig. 5B-E), or used different smoothing kernels (15 ms Gaussian or a 50 ms boxcar, Fig. S11B-E).

R1.m5 —

Line 123: I know it's trendy, but using the word "interrogate" in this context seems a bit silly. Is the neural activity seated under a bright lamp at a police station? ;-) How about just "test"?

Response:

Thank you for this comment. We wanted to avoid being repetitive in our writing and may have gotten carried away with our use of the thesaurus. After edits, we only find one instance of the word "interrogate". We have changed it to "understand" now on line 126.

In the next sections, we use dimensionality reduction, decoding, and regression analyses to understand how RT and choice are represented in the shared ...

R1.m6 —

Figure 3: Here and in all of the plots, there should be more tick-marks on the axes so that the reader has an easier time comparing the various magnitudes, even when the units are arbitrary. Sometimes readers might want to look at something you haven't discussed and giving them just the minimum and maximum values is not helpful.

Response:

We have added additional tick marks to all the figures for the interested reader.

R1.m7 —

Figure 4: The analysis on Figure 4E shows that "adjacent" trajectories were similar, but wouldn't that be expected given the high degree of overlap between adjacent RT bins? Again, I think using fewer bins that are non-overlapping would be more revealing of real trends and less subject to averaging artifacts.

Response:

The analysis in Fig. 4E holds similarly for less overlapping bins (Fig. 5E). In fact, with non-overlapping bins the results are stronger in the prestimulus period probably due to fewer pairwise comparisons of less noisy data. The differences between bins are now more apparent poststimulus as the fastest RT bin is likely moving into a movement initiation state prior to the slower RT bin hence the more than 90° angle between trajectories (Fig. 5E).

R1.m8 —

The excursions along the different PCs are quite different in magnitude, and this is not always clear. For example, Figure 4A shows the top four PCs using the same y-scale, but Figure 4B shows the trajectories with vastly different scales (I estimate about 20 for PC1 and 5 for PC4). This should be made clear in the plot. See my comment above about tick marks.

Response:

Thank you for this note. You are correct in that the axes are not equalized and thus fluctuations in PC4 look magnified. We have kept the figure as is to highlight the separation between RT bins but have added a note that the axes are not equalized in the figure caption (see below, Fig. 4B) and show an axes-equalized version in the supplementary material (Fig. S7).

Note axes are deliberately not equalized to better visualize prestimulus fluctuations (see Fig. S7 for an axes equalized version).

R1.m9 —

Line 210: I don't understand the rationale for comparing the speed of neural state change across different trajectories using the KiNeT approach. Why not just plot the magnitude of the derivative? What is the downside of this more straightforward measure?

Response: Thank you for this comment. In fact we do plot the average magnitude of the derivative of prestimulus firing rate activity (spikes/s/s) grouped by RT bin (Fig. 4G). We found a negative correlation where slower neural state changes in the prestimulus period are predictive of longer RTs. You are right that this more straightforward approach already suggests a relationship between prestimulus firing rates and RTs. We will refer to this measure as scalar speed from here on out.

So why not just use scalar speed? Our main reasoning is that neural states can start at different positions in state space *and* move at different speeds. So for example, we may measure the scalar speed of two neural trajectories, find they have the same scalar speed and conclude that they ran the same race to some movement initiation state. This neglects that neural trajectories may have had different starting positions at the time of an initiation cue. If that is the case, then one neural trajectory will reach the movement initiation state faster than the other as it had a head start, even though they moved at the same 'speed'. KiNeT solves this by measuring trajectories in relation to a common reference trajectory (what we call 'time to reference', e.g., Fig. 4F). By measuring 'speed' in comparison to a reference we can make stronger claims as to which trajectories were 'faster' than others. This for us is KiNeT's first major advantage over the measure of scalar speed.

Second, scalar speed by definition has no direction. For example, consider the case where neural activity in fast RT bins involve large changes along one dimension but slow RTs involve equally large changes along another orthogonal dimension. In both cases, the scalar speed would be the same. However, there is a significant difference between the two as the fast RT state and slow RT state are evolving along orthogonal dimensions. Again KiNeT measures the 'speed' relative to a reference trajectory in multiple dimensions and is therefore a more nuanced measure of the 'speed' at which trajectories evolve.

Finally, KiNeT allows us to measure the spatial ordering of the trajectories (e.g., Fig. 4C). Thus, we chose to use KiNeT in the manuscript as it is a more nuanced measure of how neural activity can evolve as compared to scalar speed.

R1.m10 —

Line 267: This sentence needs to be fixed grammatically.

Response:

Thank you for pointing this out, this sentence could be stated more clearly. Please see the revision below.

In contrast, a logistic regression using binned spiking activity failed to predict choice at greater than chance levels in the prestimulus period. The choice-decoding accuracy was not significantly greater than the 99th percentile of accuracy from a logistic regression using trial-shuffled spiking activity, until after stimulus presentation (Fig. 6D).

R1.m11 —

Line 304: It is said that the fast RTs have a steeper slope than slow RTs. I don't see that. To me the lines all look pretty parallel, particularly for 4% trials. The change in slope is not so compelling in the 90% and 31% coherence groups either, at least for the faster RT bins.

Response:

Thank you for catching this. We realize the error. We performed a standard analysis used in other papers (e.g., Chandrasekaran et al., 2017; Roitman and Shadlen, 2002) that just averaged the choice selectivity in a time window that in hindsight confounded latency changes with slope changes. Fortunately, the slope is already computed from our fits to the choice-selectivity signal and we now include that as a more nuanced and hopefully alternative to simple averaging (Fig. 7G).

For the easiest coherences, the latencies are largely flat (Fig. 7F) but the slopes do change with the initial condition (Fig. 7G). In contrast, for the harder coherences, the latencies are longer for initial conditions associated with slower RT bins (Fig. 7F) but the slopes are not that different from one another and largely parallel to one another (Fig. 7G). Overall, these results are still consistent with how both initial conditions and inputs decide the overall dynamics of choice selectivity in PMd.

Thank you once again for catching this analysis issue. We have now updated the relevant section and figure to incorporate a more nuanced analysis of this choice-selectivity signal. We first describe the average choice-selectivity signal in the 200 ms period and then ask if it comes from shifts in latency, slope or both.

We quantified these patterns by first measuring the *average choice-selectivity signal* in the 200 ms period from 125 to 325 ms after checkerboard onset as a function of the initial condition and for each of the 7 coherences. We obtained an estimate of the initial condition by using a PCA to project the average six-dimensional location in state space in the -300 ms to -100 ms period before checkerboard onset for each of these conditions on to a one-dimensional axis (see 4.15). As Fig. 7E shows, the average choice-selectivity signal is larger for easier coherences across the board but also weaker or stronger depending on the initial condition. Furthermore, when coherence is fixed, the average choice-selectivity signals depends on the initial condition. A partial correlation analysis found that the average choice selectivity in this time epoch depends on both the initial condition ($r_{10} = 0.85$, $p = 3 \times 10^{-22}$) and the sensory evidence ($r_6 = 0.34$, $p = 2.7 \times 10^{-3}$). These results are key evidence that choice-selective, decision-related dynamics are controlled both by the initial condition and the sensory evidence.

Do effects observed in Fig. 7E emerge from slope changes, latency changes or both? To address this question, we fit the choice-selectivity signal ($CS(t)$) using a piecewise function (eq. 4.14) with a latency and slope parameter. Fig. 7F plots the latency of the choice-selectivity signal (t_{Latency}) as a function of the sensory input and the initial condition. Latencies depend on both the initial condition and sensory evidence. Latencies are slowest when the initial condition is in the slow RT state and for weak inputs but faster for strong inputs or when the initial condition is in a fast RT state. Consistent with this joint dependence, a partial correlation analysis found that the latency of choice selectivity depends on both the initial condition ($r_{10} = -0.54$, $p = 3.95 \times 10^{-07}$) and stimulus coherence ($r_6 = -0.38$, $p = 8.34 \times 10^{-4}$).

Fig. 7G plots the slope of the choice-selectivity signal (m) as a function of the sensory input and the initial condition. In contrast to latency, slope of the choice-selectivity signal was strongly dependent on the initial condition but only weakly modulated by coherence. A partial correlation analysis confirmed these observations. Slope was strongly correlated with initial condition ($r_{10} = .77$, $p = 4.03 \times 10^{-16}$) but had modest to no relationship to sensory evidence ($r_6 = -0.16$, $p = 0.17$).

R1.m12 —

Figure 6: Are the PCs here the same as in Figure 4?

Response:

If you are referring to Figure 7C, then yes the trajectories are similar to the one presented in Figure 4B. The difference is that in Figure 7C the trajectories are organized within a single coherence whereas for Figure 4B the trajectories are organized across all coherences.

We believe this is related to your 4th major point and as a result of that point we have added an inset to Figure 4B. We hope by addressing that point and with the explanation now in the methods (see below) that we have made the similarities and distinctions between the trajectories in Figure 4B and Figure 7C clearer.

To perform the within-coherence RT PCAs (3 shown, Fig. 7C) we first generated seven (1 for each coherence) trial-averaged firing rate matrices organized by RT and choice (each 11 RT bins \times 2 choices). Then we projected each of these 7 matrices into the PC space obtained by PCA on the RT and choice data (i.e., Fig. 4)

R1.m13 —

Figure 7C: Why is there no response to the cue? I would expect to see one given PC3, above.

Response:

There is no response to the cue for the analysis in Figure 8C as this analysis was averaged across reaches. Meaning that there should be no separation by choice (as seen in PC 3). This is explained in section 4.11 of the methods and reproduced below.

All of the following calculations in this section were first performed within a particular choice and then averaged across choices.

Additionally, we added a statement when KiNeT analyses are first introduced, section 2.5, to clarify that 6 dimensions were used for the analyses and that they were first performed within a choice and then averaged across choices, see below.

For KiNeT analyses we used the first six PCs (>90% of variance) as these PCs were significantly different from noise principal components (Machens et al., 2010). KiNeT analyses are first performed within a choice and then averaged across choices.

R1.m14 — Line 376: I did not understand this paragraph.

Response:

Thank you for your candor. We have now described the findings from this analysis in a new section, 2.10, and rewrote the description to be clearer. In this section we perform the analyses that you describe in comment 1.8 (i.e., project firing rates organized by one condition (e.g., outcome) into the PC space organized by another condition (e.g., RT)) and quantify it using an alignment index developed in (Elsayed et al., 2016). Basically, when we project firing rates organized by RT and choice into the PC space defined by outcome and choice, the projections have the same structure as when the PCA is done on firing rates organized by RT and choice (or vice versa). To quantify, this degree of overlap, we used the alignment index.

First, we projected the firing rates organized by outcome and choice, onto the ‘RT subspace’ (defined using the first 6 principal components of the PCA in Fig. 4). If the space defined by trial outcome, and space defined by RT show a strong degree of overlap (“overlapping” in Fig. S20A), then the cross projection would reveal meaningful structure. In contrast, if the subspaces were independent, then cross projection would be largely unstructured (Elsayed et al., 2016, “independent” in Fig. S20A). Consistent with our hypothesis, when we projected the firing rates organized by outcome and choice onto the RT subspace, we found near identical structure to what we observed when performing PCA on the firing rates organized by trial outcome and RT (Fig. S20B). We also performed the converse of this analysis where we projected firing rates organized by RT and choice into the space defined by trial outcome and choice (Fig. S20C). Consistent with our hypothesis that changes in trial outcome leads to changes in RT, we found near identical structure to what we observed when performing PCA on the firing rates organized by choice and RT (Fig. 4). These cross projection analyses show that the subspaces identified by trial outcome and choice, and RT and choice are highly overlapping with one another.

To quantify the strength of this overlap we used a previously developed alignment index (Elsayed et al., 2016). Briefly, the index calculates the trace of the matrix that results from the projection of the RT space onto the first six principal components of the outcome subspace (i.e., sum of eigenvalues) and divides this by the sum of the eigenvalues from the PCA on firing rates organized by RT and choice (further detailed methods section 4.22). Thus the index, as used here, quantifies the amount of variance in the RT space (Fig. 4A, B) that could be accounted for by the outcome subspace (Fig. 8A, B). This analysis revealed that ~77% of the total variance for the RT space was explained by the outcome subspace, suggesting that the previous trial’s outcome has a large impact in explaining prestimulus firing rate covariation with RT.

R1.m15 —

Line 447: This paragraph is perhaps a bit too quick to dismiss the DDM. Indeed, a one-dimensional variable cannot explain the results shown here, but what about a 2-dimensional drift model in which the state can shift around on a plane? If the “thresholds” for initiation converge together near one end of that plane, then reaction times could be shortened if the system starts closer to that end.

Response:

Thank you for the comment. We agree and we were being cautious in saying that *one-dimensional* DDMs, often the dominant model for decision-making, fail to describe our neural data completely. But, we agree that a two-dimensional DDM would work for our results. We have added the following paragraph in the Discussion to address this.

Thus, these results suggest that additional dimensions might be needed for the DDM to faithfully replicate our neural data. For example, consider a two-dimensional DDM, where the x and y axes are the bounds. If the initial state is close to the origin along the 45° diagonal line, then RTs would be faster for both choices. Conversely, if the initial state is farther from the origin along the 45° diagonal line for both choices then the RTs would be longer. Such a model is consistent with our data and could potentially explain the behavior and neural responses described here.

R1.m16 —

Line 664: It is said that recordings were “lateral to the spur of the arcuate sulcus”. Is that correct? Most people would consider that to be PMv, not PMd.

Response:

Apologies for this error! You are correct. Very lateral to the spur of the arcuate sulcus would be in the ventral premotor cortex.

Our chamber center was at the stereotaxic coordinates +16, 15. Such a location is lateral to the precentral dimple and medial to the spur of the arcuate sulcus. Our recordings were therefore firmly in the caudal aspect of the dorsal premotor cortex.

Recordings were made anterior to the central sulcus, lateral to the precentral dimple and medial to the spur of the arcuate sulcus.

R1.m17 —

Equation 9: To what do the superscripts k and l refer? Are these just randomly chosen trials?

Response:

Yes. They are randomly chosen trials. The assumption is that these trials have the same signal component but different noise components and thus subtraction of the two yields an estimate of the noise process and thus can be used to compute the noise covariance matrix across neurons.

To generate representative noise traces for our firing rates, consider firing rates of two trials $r_i^k(t)$, and $r_i^l(t)$ for the i^{th} neuron (the superscripts k and l refer to any two different trials).

R1.m18 —

Line 837: Perhaps replace the “-” with “is the”, so it’s clearer.

Response:

There were several instances where the dashes were replaced with “is the”. The corresponding changes are shown below.

$\Omega_L(t)$ and $\Omega_R(t)$ are the six-dimensional location in state space for a left and right choice at time t .

β_0 is the intercept of the model, β_j is the model coefficient for the j th neuron in the current bin, X_j is the spiking activity of the j th neuron of the current bin.

Where J is the cost associated with coefficients, λ is the penalty term (1/number of in-fold observations), and β are the coefficients of the model.

R1.m19 —

Figure S3: It is stated that “Post-error choice selectivity may be larger than other trial outcomes”. What are you referring to? The slightly higher purple trace at the end? Is that significant?

Response:

This was an additional analysis that we had performed but was incorrectly omitted in the previous version of the paper. So thank you for catching this! Indeed we were referring to the slightly higher purple line.

Our single-trial analysis revealed that post-error trials take a more convoluted path and also end at a slightly different part of state space than post-correct trials just before movement onset (Fig. S18D). Thus we examined the choice-selectivity signal (again defined as the Euclidean distance between left and right choices in the first six dimensions) aligned to movement onset for correct, post-correct, error, and post-error trials (Fig. S18E). For all trial types, choice-selectivity signals peaked ~ 100 ms before movement onset, consistent with prior reports in the field (Roitman and Shadlen, 2002). In addition, we found that the choice-selectivity signal was higher for post-error trials as compared to all other trial-types considered. We now include the following text in the revised manuscript.

The single-trial dynamics organized by trial outcome also suggest that neural state for post-error trials is separated from the post-correct trials at the time of movement onset (Fig. S18D). We reasoned that such differences would lead to differences in choice selectivity between the different trial outcomes before movement onset. Consistent with this reasoning, the six-dimensional Euclidean distance between left and right choice trajectories was largely flat until ~ 250 ms before movement-onset at which point it increased for all trial types (Fig. S18E). Post-error trials demonstrated the strongest choice selectivity as compared to all other trial types at least 250 ms before movement onset, with the difference between trial types peaking ~ 90 ms before movement onset (Fig. S18E).

Reviewer 2

Comments:

R2.1 — In this manuscript, the authors use a dynamical systems framework to interpret neural responses measured from PMd during a perceptual decision-making task. They find that their framework provides a parsimonious way to interpret results that, by themselves, might appear to be arbitrary. Specifically, a dynamical systems approach interprets the firing rates before cue onset as the “initial state”, which, along with the stimulus input, sets how the dynamics unfold on each trial. The authors show that initial state covaries strongly with reaction time and not so much with choice. The paper is well written, the data and results are clear, and the concepts well illustrated.

Response:

Thank you for your kind comments on the quality of the manuscript and the precise summary of our findings. Our goal was to advance our understanding of decision-making through the analysis of neural data.

That said, I have three concerns: 1) I’m worried the results depend on the choice of smoothing. 2) I’m worried the analyses in this manuscript are overstating the variance explained. 3) I would like to see some demonstration of how good the dynamical systems framework is as a model of neural activity so that other approaches have something they can actually compare to.

Response: We thank you for these comments. We realize that the framing of our manuscript was a bit confusing and perhaps the variance explained should be understood in the context of the PCA analyses performed on the trial-averaged data. Much of the manuscript was also devoted to analysis of the PCA trajectories and only small amounts of cross-validated prediction of single-trial neural data and its link to behavior. To assuage your concerns, we have now done the following in the manuscript.

- **Smoothing:** First and foremost, principal component trajectories are largely similar even when using a smaller 15 ms Gaussian or a 50 ms causal boxcar smoothing kernel (Fig. S11A). Similarly, the results of the KiNeT analyses are largely similar for various levels of smoothing (Fig. S11B-E). Variance explained by the first 6 dimensions is 91%, 87%, and 86% for the 30 ms Gaussian, 15 ms Gaussian, and 50 ms boxcar smoothing kernels (Fig. S5). Finally, we also show that the smoothness of our data is not because we use overlapping RT bins. Even when we only use three nonoverlapping RT bins, we get the same results (Fig. 5).
- **Variance explained:** We recognize how our result that 90% of the variance is explained by just 6 dimensions can feel very surprising. But, we emphasize this is trial-averaged variance and not single-trial variance. Variance in neural activity can come from at least three sources: representation of task variables (usually termed signal), uncontrolled behavioral movements (uninstructed movements), and finally from noisiness of spiking activity from the spike generation process. All of these will contribute to single-trial variability.

For our experiments, the reviewer should first keep in mind that these are well trained monkeys whose body movements are gently restricted by our primate chair and also a sling that also does not allow movement of the other hand. We also rigorously control for arousal and eye movements — the task was stopped when the monkey was drifting off in behavior or the eyes start to droop and restarted after a pause. Second, trial-averaging means that we are reducing the variability of the responses by reducing the variability induced by the spiking activity. Finally, our animals are performing a behavioral task where the output is to reach left or right. Any other movements are not allowed in the task and that would fail

the trial. As described below, we are also recording in an area that is largely associated with computing directional choice, and not color choice or the target configuration signals.

Note, we are far from the first people to note that the trial-averaged variance of firing rates is often low-dimensional in well-controlled behavioral tasks where animals are often performing simple motor or discrimination tasks (e.g., Gao and Ganguli, 2015). As Table 2 shows, on average 85% of trial-averaged variance is explained by the first 10 dimensions in many studies. Some even go so far as to argue that behaviorally-relevant dynamics in decision making are largely one-dimensional (Ganguli et al., 2008; Steinemann et al., 2022) and that this dimension explains a large amount of the neural variance in that brain region.

Collectively, these studies suggest that the dimensionality of trial-averaged data strongly depends on the task, brain region and task variables included in the analysis and only weakly on the degree of smoothing in the data. For example, it only takes 3 dimensions in PMd but 8 dimensions in DLPFC to explain over 90% of the variance from a PCA on trial-averaged firing rates, aligned to target and checkerboard onset, organized by chosen target color and reach direction (Reviewer Fig. 3B, 30 ms Gaussian smoothing used for these analyses). This is a result of the fact that almost all of the variance in PMd is associated with action-choice related signals (Reviewer Fig. 3C). Note, we also have squashed variance related to RT and coherence in these analyses because of trial-averaging and thus marginalizing over these variables. Thus, the true dimensionality of a system is very different from the **dimensionality of trial-averaged data**. So in our hands, the 6 dimensions explaining > 85% of the variance is due to trial-averaging, simple representations in the brain area, and rigorous behavioral control.

- **Corroborating evidence:** We emphasize that our results are robust even if the reviewer disregards analyses that use dimensionality reduction, as our key results that prestimulus firing rate strongly correlates with RT (Fig. 6), and that it's modified by trial outcome (Fig. 8D) are supported by decoders and linear regressions. Furthermore a regression, now included as a supplementary figure, estimates the degree to which single neuron firing rates covary with choice or RT before and after stimulus onset. This new analysis demonstrates that ~20% of neurons covary with RT and ~1% (chance levels) of neurons covary with choice before stimulus onset (Fig. S17D).
- **Predictions of Neural Data:** Finally, how good are dynamical systems methods in fitting our data? This is an excellent and deep question. Before we describe our analysis to address this issue for this paper, we want to point the reviewer to a few excellent papers in the field that actually have tackled this issue in depth.
 - The first example is in the context of motor cortical responses during reaching tasks. Churchland et al. (2012) showed that trial-averaged firing rates of motor cortical neurons aligned to movement onset are consistent with a rotational dynamical system and Elsayed and Cunningham (2017) showed that such data are not a trivial result of tuning in this population.
 - Second, in Wei et al. (2019) the authors fit linear dynamical systems to activity of neurons in the mouse premotor cortex (ALM) and showed that such an approach could predict neural activity and also perform much better at decoding various behavioral metrics.
 - Finally, Pandarinath et al. (2018) show that the LFADS approach that is also used in the paper and fits a dynamical system can provide excellent descriptions of the neural activity in a given brain area.

As recommended by you, we now use Tensor Component Analysis (Fig. S12), reduced-rank regression (Fig. S15), a simple linear dynamical system (Fig. S13) and LFADS (Fig. S14) to predict held out data from single-trial firing rates of small populations of neurons. We use these results to 1) show that dynamical systems are a good model of neural activity for PMd in this decision-making task, and 2) that the dynamical system captures considerable variance in RT associated with this task during the prestimulus period. Together these single trial analyses still support our results that prestimulus neural activity predicts RT but not choice.

Figure 3: DLPFC dynamics are higher-dimensional and have more context-related signals than PMd dynamics in the checkerboard task (A) PCA trajectories for DLPFC (top, $PC_{1,2,4}$) and PMd (bottom, $PC_{1,2,3}$) from trial-averaged firing rates aligned to target (black dots) and checkerboard onset (red dots) and organized by chosen target color (red & green) and reach direction (right - dashed, left - solid). Prestimulus neural activity separates as a function of target context (i.e., the arrangement of the green and red targets) in DLPFC but not in PMd. Both areas show strong choice separation after stimulus onset. (B) Cumulative variance explained by the first 20 PCs for PCAs organized by target color choice and reach direction for PMd (first 3 PCs explain >90% of the variance) and DLPFC (first 8 PCs explain >90% of the variance). (C) Demixed-PCA of trial-averaged firing rates aligned to target and checkerboard onset organized by chosen target color and reach direction in DLPFC and PMd. Only DLPFC dynamics have signals related to context and dominant checkerboard color while these modulations in PMd are minimal.

R2.2 —

Full disclaimer: I don't use the dynamical systems framework to interpret neural activity, but I was left with a general feeling reading this: dimensionality reduction is a useful visualization tool, but is dynamical systems a good model of neural activity?

Response:

This is an excellent point. As you point out, we did not explicitly perform many cross-validated single-trial analyses in the first version of our manuscript. We relied heavily on dimensionality reduction of trial-averaged data to visualize population neural activity and analyses of the geometry of these trajectories to argue for the dynamical system (Remington et al., 2018a,b). The dynamical systems perspective/approach assumes that in neural circuits in brain areas associated with behavioral tasks there is a latent low-dimensional computation that evolves according to a dynamical system (Vyas et al., 2020). The assumption is that individual neurons in these brain areas are a readout of this underlying hidden dynamical system. One way to visualize these low-dimensional computations is dimensionality reduction. Thus dimensionality reduction is a useful visualization tool in the context of understanding and recovering the dynamics in these brain areas. It is not the only technique but it is the starting point for our analyses.

As to your point of whether dynamical systems is a good model of neural activity, we are not the first people to suggest this. First, in classical single neuron modelling with Hodgkin-Huxley and integrate-and-fire neurons (e.g., Izhikevich, 2003), the voltage is described by a differential equation that is a dynamical equation. Second, models of decision-making (Niyogi and Wong-Lin, 2013; Wang, 2002) and recurrent neural networks are obvious examples of using dynamical systems based approaches to model neural activity. Furthermore the field of decision-making implicitly thinks in dynamical systems terms. For instance, the classical drift diffusion model of decision-making assumes that evidence is integrated over time (Ratcliff et al., 2016). Apart from the "everyone else is doing it" argument, in our hands it provides a useful framework for forcing us to think in mathematical terms about latent neural processes.

We apologize if in our previous writing of the manuscript the dynamical systems perspective was portrayed as the be-all, end-all or as an analysis approach rather than as an arbitration between different models. It's hardly the only way to think about the brain nor is it antagonistic towards other approaches. We mean to use the dynamical systems perspective as a means to **derive various dynamical hypotheses** of how neural activity may evolve in PMd in this task. Your comments have made us realize this and we have reworked the manuscript such that we explicitly arbitrate between dynamical models (Fig. 1D-G, Fig. S2, Fig. S3, and see below) and have added analyses to further corroborate our findings (Fig. S17D).

We use a large suite of methods that arbitrate between these dynamical models such as dimensionality reduction, reduced-rank regression (based on your recommendation), decoding and single-trial analysis approaches. All of these approaches are useful for rejecting various dynamical hypotheses. We now write in our revised introduction:

Based on these findings, we hypothesized four different dynamical mechanisms that could describe the data.

- The simplest dynamical system assumes initial conditions do not covary with RT or choice and that neural dynamics and behavior are driven largely by the sensory evidence (Fig. 1E).
- A second dynamical system assumes that the initial conditions do not vary, but that there are either systematic or random delays in sensory evidence processing (Oram, 2010), that alter choice-related dynamics and behavior (Fig. 1F).
- A third system assumes that initial conditions are biased towards one or another choice (Shadlen and Newsome, 2001), correlate with RT, and that poststimulus dynamics are influenced by both initial conditions and sensory evidence (Fig. 1G).
- Finally, a fourth system assumes that initial conditions correlate with RT but not choice, and poststimulus dynamics depend on both sensory evidence and initial condition. Additionally, the changes in initial condition are in part due to the outcome of the previous trial (Fig. 1H).

R2.3 —

If the fundamental statement here is that the activity of neural populations depends both on new inputs and the previous state of the system, what is the alternative?

In other words, how could the authors build more trust in these extremely smooth trajectories they plot?

Response:

We recognize the issue with our paper as previously written. In essence, we were arguing for a truism (i.e., the brain is a dynamical system) and now based on the advice from both reviewers, we now recognize that what we really wish to do is arbitrate between models of dynamics.

First, we have now edited the manuscript to say dynamical systems are a general framework and in our mind the important question is what specific dynamical mechanism is most consistent with our data. For instance, our data are strongly inconsistent with classical models of decision-making that posit that activity during decision-making is entirely based on the input sensory evidence (e.g., Ratcliff et al., 2016). Such a dynamical mechanism predicts that choice and RT are wholly dependent on the sensory evidence, neither of which are dependent upon prestimulus neural activity. Given our results we hope that it is clear that for this brain region and task that this alternative is not an acceptable model. Additional models were built after the feedback from reviewers (Fig. 1E-H, Fig. S2, Fig. S3) and we hope that it can be seen that most of these models don't capture rich covariation between RT and prestimulus neural activity.

Second, to build more trust in the extremely smooth trajectories that we present we took your suggestion to use different smoothing kernels (i.e., 15 ms Gaussian and 50 ms Boxcar; PCA and KiNeT: Fig. S11). The change in smoothing kernels does not change any of the findings of this study, as addressed below. As an aside, and as explained in Table 1 and 2 below, our use of a 30 ms kernel generally was close to many other publications and was even smaller than some others.

R2.4 —

How much variance in the single trial responses is explained by this approach?

Or, is this just a convenient way to visualize and interpret heterogeneous PSTHs?

If it is the latter, then could we not test underlying models of neural activity more directly with regression? How should people using other approaches compare to your approach?

Response:

Thank you for this comment. We have now explicitly performed two sets of analyses to determine how well our data is described by a dynamical system. We fit both a simple linear dynamical system and LFADS, a sequential variational autoencoder that fits a nonlinear dynamical system, to single-trial firing rates from small populations of neurons. Both approaches show that a dynamical system can predict substantial amounts of variance in held-out data (Fig. S13, Fig. S14). Even just a 10-dimensional linear dynamical system explains substantial variance in single-trial firing rates ($\sim 40\%$) and is almost as good as the full dataset in capturing RT variance before and after stimulus onset (Fig. S13B, C). Similarly, LFADS, an 8-dimensional nonlinear dynamical system, powerfully describes held out neural activity in our task (Fig. S14A, B).

We can absolutely test the underlying models of neural activity with regression! We believe that we quite prominently featured linear and logistic regression in the original manuscript to verify and to further quantify our observations from dimensionality reduction (Fig. 6E, F). In addition we have added additional regressions based upon reviewer feedback (e.g., Fig. S15B, C, Fig. S17D) which further corroborate our main findings. The use of the dynamical systems perspective is simply another perspective from which to understand brain activity and as we emphasize throughout this review process, we are trying to recast this as an arbitration between dynamical models most consistent with our data. Furthermore for that arbitration process, one should use every single tool in the book including: dimensionality reduction, regression, decoding, reduced-rank regression, modeling, simulations, explicit fits of dynamical models, analysis of residuals and so on. Thus dependent upon the specific analyses being performed there are likely analogues that others use that can compare to any of the methods used here. Indeed throughout this manuscript we pair 'newer' methods (e.g., Pandarinath et al., 2018) with more time-tested methods as well, as in for example Fig. 6.

As an aside, thank you for the advice to use reduced-rank regression! We now use it to explicitly show that prestimulus neural activity is strongly predictive of post-stimulus neural activity (Fig. S15B-D).

R2.5 —

Essentially, I would like to see some of the dynamics analyses explicitly cross-validated and I would like to see the results run with smaller amounts of smoothing (and only causal smoothing). For example, in the PCA analysis, 6 PCs explain 90% of the variance of the PSTHs... after smoothing with 30ms gaussians. That is a lot of smoothing! As a quick check, I took white noise of the similar dimensions to what was reported here (1800 x 996) and smoothed it columnwise with a gaussian (with standard deviation = 30) and did PCA. With that level of smoothing, 90% of the variance of my simulated 996 white noise neurons were explained with 20 dimensions. 6 is a lot smaller than 996, but it's not a lot smaller than 20. So, I'm left wondering how good a model are these smooth trajectories?

Response:

Thank you for this comment, the example you provide concerned us as well. However the size of the white noise matrix in your example is smaller than any matrix that we used in any of our PCAs. All of our PCAs are organized by time in trial, number of units, levels of one of the conditions (i.e., coherence, RT, or previous outcome), and choice. So trial-averaged firing rates of units organized by RT and choice is a $1800 \times 996 \times 11 \times 2$ dimension matrix, organized by coherence and choice is a $1800 \times 996 \times 7 \times 2$ dimension matrix, and organized by outcome and choice is a $1800 \times 996 \times 4 \times 2$ dimension matrix. However, when the PCA is actually performed the time dimension is actually truncated based upon level of condition (e.g., median RT for post-error trials versus median RT for post-correct trials). So on average each level of a condition is ~ 700 ms to ~ 800 ms in length. Given this when the matrices are preprocessed for PCA they actually end up becoming matrices of size: 17428×996 (RT), $11,000 \times 996$ (coherence), and $6,000 \times 996$ (trial outcome).

Therefore we followed your example and recreated the analysis you describe (column-wise smoothing of a white noise matrix with a 30 ms Gaussian). First, as a sanity check we recapitulated your exact analysis on a 1800×996 dimension matrix and also found that 20 dimensions explains over 90% of the variance. However, when these white noise matrices actually match the size of our RT matrix, it takes 193 dimensions to explain over

90% of the variance, for the coherence matrix it takes 127 dimensions to explain over 90% of the variance, and finally 74 dimensions to explain over 90% of the variance for the trial outcome sized-matrix. Our data require 1-2 orders of magnitude fewer dimensions to explain over 90% of the variance than size-matched noise matrices.

As expected when we try this using smaller kernels, the amount of variance explained scales linearly. In other words it requires more dimensions to explain more of the variance for the data and for the noise matrices. But regardless of the kernel used for smoothing, the data matrices require significantly fewer dimensions than the noise matrices in order to explain over 90% of the variance.

Two other analyses reassure us, and hopefully the reviewer, that it is not the smoothing that leads to the smooth trajectories and only needing 6 dimensions to explain 90% of the variance. First, in the originally submitted manuscript we compare signal+noise components to pure noise components (obtained by subtracting two random trials from one another and performing PCA on the residuals) to determine when signal+noise components explain less variance than the pure noise components (Fig. S5A, Fig. S18C). We felt this analysis was a principled method (Machens et al., 2010) for determining a cut off for components to include in following analyses (e.g., KiNeT). Therefore, as the noise dimensions have the same level of smoothing as the signal+noise dimensions, we believe this is yet further demonstration that that our data is not overly smoothed.

If size of smoothing kernel is not causing these extremely smooth trajectories, then what is? We believe the source of the 'smoothness' of these trajectories has to do with trial averaging. Consider the simulations where we generated a hypothetical neural population that resembled PMd (Fig. S3). In this simulation, there is no animal, arousal effects, unintended movements etc. The variance comes from the generative latents in the data (choice selectivity, correlation with RT, condition-independent signal, etc.) that are combined to lead to the firing rates and the Poisson spiking noise. If we smooth the spike trains with a 30 ms Gaussian and perform PCA for single trial firing rates, we need ~210 dimensions to describe 90% of variance in the data. In contrast, if we first average trials in different bins and then perform PCA on this trial-averaged data, we only need 7 dimensions to describe 90% of the variance in this data (Reviewer Fig. 4A). The reason trial-averaging is so powerful is that it reduces the variance from the single trials and provides an estimate of how neural activity evolves on average for these different conditions. If we reduce the smoothing kernel, the data essentially becomes noisier and the number of dimensions needed for both trial-averaged and single-trial data increase (Reviewer Fig. 4B). However, this drop in variance explained by the first few dimensions caused by a smaller smoothing kernel is much smaller than the change in variance between trial-averaged vs. non-trial averaged firing rates. Thus, these simulations suggest that the effect of trial-averaging is way more powerful than the effect of smoothing.

In summary, your comment asks if smoothing is what caused the extremely smooth trajectories. Smoothing kernels have a very minor effect. However, it appears that trial-averaging robustly suppresses overall noise (e.g. spiking noise) as well as latent variability, as intended, and thus the trial-averaged data can be summarized by only a few dimensions.

R2.6 —

The regression analysis in figure 5 is cross-validated, which is the only place in the manuscript that phrase is mentioned, and one of the appealing reasons to use regression to analyze and neural data. Why do I care? Well, the regression analyses in figure 5 tells us the main data result: firing rates before cue onset are predictive of RT, but not choice. Which isn't even true of all perceptual decision-making datasets (as the authors point out in the discussion). Something like reduced-rank regression can provide low-dimensional descriptions of neural activity which can be modeled as dependent on both inputs and previous states of the neural activity. With reduced-rank regression, it would be easy to evaluate the variance explained on withheld data. There are a lot of strengths a regression framework would have for interpreting these results that are missing from the dynamical systems framework.

So, I think, as an outsider to this approach, any good model of neural activity has to first show that it can predict neural activity. As it stands, that is missing, and this manuscript would benefit from stating that.

Figure 4: Variance explained by the first few components when PCA is performed on either the trial-averaged data from the simulated neural population in Fig. S3 by RT and choice, or on the single-trial firing rates. Spike trains are either smoothed by a 30 ms Gaussian (A) or by a 20 ms Gaussian (B). Note, trial-averaging suppresses spiking noise as well as single-trial variability leading to PCA needing far fewer dimensions to explain the data. In contrast, many more dimensions are needed to explain single-trial data because of variability induced by the latents as well as the spiking variability.

At a minimum, it would reveal whether this framework is just an (admittedly nice) visualization tool or whether it is a useful model of neural activity.

Response:

Thank you, you are absolutely correct. Any good model of neural activity should demonstrate that it can predict future neural activity and our previous manuscript was missing this key point.

We have now included a reduced-rank regression model as you suggest (Fig. S15A). The reduced-rank regression model furthers our previous results as it demonstrates that neural activity 550 ms before checkerboard onset can predict future neural activity better than a shuffle control up to ~ 200 ms after checkerboard onset (about the point that choice signals begin to emerge) in a single session (Fig. S15B) and on average across 40 sessions (Fig. S15C). The reason that the trial-shuffled model performs so well after checkerboard onset is that the reduced-rank regression models include terms for choice and stimulus coherence.

Furthermore this reduced-rank regression corroborates and furthers the insights from the subspace alignment analysis performed using our trial-averaged KiNeT approach (Fig. 4D, E). Our KiNeT analysis with trial-averaged data suggested that the subspace was largely stable during the prestimulus period and started to change after stimulus onset when choice-related signals emerge. We repeated this analysis for the β estimated from reduced-rank regression at each time point (Fig. S15D). We calculated the angle between the β for the first time point to the β for all other time points and found that the β was largely small and stable throughout the prestimulus period and into the poststimulus period (Fig. S15D). In contrast for the shuffle control, this angle was much higher suggesting that the β at each time point was not related to the β at other time points. These results suggest a temporally (prestimulus to poststimulus) stable dynamical system that reliably predicts future neural activity until about the point that choice signals emerge (~ 200 ms). At this point there's appears to be a robust change to the underlying system, likely related to movement signals.

Thus, reduced-rank regression supports a dynamical model of decision-making where the initial conditions predict future neural activity, correlate with RT. We hope that this further helps to illustrate that the dynamical systems framework is more than just a visualization tool and that it can be a useful model for neural activity.

R2.7 —

So, why not withhold single trials from the PCA analyses and ask how much of their variance is explained? Or, even better, use the pre-stimulus firing rates plus some measure of the dynamics during the stimulus to predict responses to withheld data. Ideally, you would do matrix factorization on pseudo populations at the single trial level with something like the speckled holdout approach in Williams et al., 2018 (which has some author overlap, so I expect you could come up with something). Although, in that paper, the hold out should have blocked the speckling to the level of trials (i.e., withhold neurons for entire trials not timepoints). There is simply too much autocorrelation to hold out individual time points, which leads to bleed-through from the training set to the test set and an overestimation of the variance explained. Basically, what I am saying is that this seems like a potential weakness of the dynamical systems framework and since this paper is mainly an argument that dynamical systems is useful as a framework here, I think it's necessary to address.

Response:

Thank you for this comment. As we understand it you would like us to perform tensor component analysis (TCA), using the Williams et al. (2018) approach but replace the speckled holdout approach by holding out neurons for entire trials. We are familiar with the work in Williams et al. (2018) however the only authorship overlap is the late Dr. Shenoy whose lab provided some of the data for Dr. Williams's analysis. Nevertheless, we took the suggestion and performed TCA on our data. We also performed two other cross-validated analyses that operate at the level of single trials to further corroborate our findings. We describe them below and also include them in the manuscript as supplementary figures.

Tensor Component Analysis

First, we used TCA to model the data from single sessions as a combination of neuron factors, temporal factors, and trial factors. As the reviewer is probably well aware, TCA is a generalization of PCA to tensors and thus superior to PCA which operates on matrices (see Williams et al., 2018). We found that TCA could explain considerable single-trial firing rate variance (Fig. S12C, D). As dimensionality of the TCA increases it explains more of the data for both the training and the test sets (Fig. S12D). This effect was robust to speckled hold out or, as recommended, to withholding neurons from trials. We also found that the denoised single-trial trajectories obtained from TCA reliably covaried with RT even before checkerboard onset (Fig. S12C).

Linear Dynamical Systems

Second, we analyzed single-trial firing rates from small populations of neurons on single sessions and asked how well their data is described by a simple dynamical system. We first binned spike counts (50 ms bins) to obtain single-trial firing rates ($X \in R^{T \times N \times M}$), where T is the number of time points, N is the number of neurons, and M is the number of trials) and then performed PCA on these binned spike counts to get a reduced representation ($\underline{X} \in R^{T \times \underline{N} \times M}$). We then asked how well \underline{X} were described by the following dynamical system

$$\dot{\underline{X}} = J\underline{X} \quad (2)$$

We specifically estimated how well we could predict \underline{X}_{t+100} from \underline{X}_t . Although, our data was smoothed with a 50 ms bin, we felt that a stronger test of a dynamical system is to predict data further into the future (Fig. S13A).

Importantly, we held out a trial and estimated J from all other trials and obtained a leave-one out cross validation coefficient of determination. We also varied \underline{N} from 2 to 10 and found that higher dimensionality improved the description of these firing rates especially as compared to shuffle controls, where we randomized the neurons for each trial, preserving temporal smoothness (Fig. S13B). The smoothed representation of this data obtained from the above equation logically predicted RT and that the dynamical system fit to the data was capturing behaviorally relevant variance in the neural data (Fig. S13C). Finally, the 10D dynamical system was almost as good as the full data in capturing the RT variance in these sessions (Fig. S13C).

LFADS

Our third approach was to use a LFADS (Pandarinath et al., 2018) analysis to predict the variance of held-out single-trial, single neural firing rates (Fig. S14A). LFADS assumes that

$$\dot{\mathbf{x}}(t) = \mathbf{F}(\mathbf{x}(t), \mathbf{u}(t)) \quad (3)$$

The state of the dynamical system $x(t)$ at each time step is updated by the non-linear, vector-valued function F and accepts an optional input $u(t)$. F is seeded by an initial condition, $x(0)$. LFADS models $F, x(0)$, and optionally $u(t)$. For simplicity, we just considered a dynamical system without explicit inputs.

$$\dot{\mathbf{x}}(t) = \mathbf{F}(\mathbf{x}(t)) \quad (4)$$

x is then used to generate final spike counts through an exponential nonlinearity and Poisson counts. We elide some of the details here, but LFADS uses a variational autoencoder which takes spike counts as inputs to generate initial states that are then fed into a generator recurrent neural network which fits these dynamical systems to finally obtain denoised single-trial firing rates.

We again fit binned spike counts from single sessions and estimated the variance on held out trials. We found that we could predict single trial firing rates better than the 99th percentile of shuffled data for each of 14

sessions (Fig. S14B). Additionally, factors estimated from the generator RNN could describe the RT of the animal reliably (Fig. S14C).

Thus, these three approaches 1) strongly suggest that a dynamical system can help describe held out data, 2) capture behaviorally relevant variance, and 3) show that before stimulus onset these firing rates are strongly predictive of RT. When combined with other results the broad stroke suggestion is that a dynamical system where prestimulus firing rates predict RT but not choice is a good model of PMd data.

Minor Comments:

R2.m1 —

Throughout the manuscript, I kept finding myself worrying that the results depend on that heavy smoothing with a Gaussian Kernel. Below are all the places that I was worried that things would change dramatically if you used a more modest 15ms kernel. Or a boxcar. Here are all the specific places that I was concerned about the effect of the choice of smoothing.

Response:

Thank you for this comment.

Kernels used in other studies: We recognize that smoothing is at the center of your concerns. We first note that our choice of a 30 ms Gaussian kernel is well within the range of Gaussian kernels published in the literature. In Table 1 we summarize Gaussian kernels, organized in ascending order, used in more classical studies and in Table 2 summarizes the Gaussian kernels used in studies that use PCA (organized by number of dimensions in ascending order needed to capture some percentage of variance). These papers are all from systems neuroscience labs across the world studying similar questions. So smoothing with a 30 ms Gaussian kernel is a typical choice in many labs around the world.

Citation	Kernel	Region
Churchland et al. (2012)	20 ms & 24 ms Gaussian	PMd/M1
Purcell and Kiani (2016)	25 ms Gaussian, 100 ms boxcar	LIP
Churchland and Shenoy (2007)	25 ms Gaussian	PMd/M1
Kaufman et al. (2016)	28 ms Gaussian	PMd/M1
Churchland et al. (2006)	30 ms Gaussian	PMd/M1
Mohan et al. (2021)	30 ms Gaussian	LIP
Mante et al. (2013)	40 ms Gaussian	PFC
Jun et al. (2010)	50 ms Gaussian	PFC
Kobak et al. (2016)	50 ms Gaussian	PFC
Fitzgerald et al. (2013)	50 ms Gaussian	LIP

Table 1: The mean Gaussian kernel from this selection of publications is 33.82 ms. Abbreviations: PMd - dorsal premotor cortex, M1 - primary motor cortex, LIP - lateral interparietal cortex, PFC- prefrontal cortex.

Analysis with other kernels: Nevertheless, we understand your concerns of over smoothing and have set out to assuage them by using more modest kernels. We have added supplementary figures (Fig. S4 & Fig. S11) which compares individual units (Fig. 3, 30 ms Gaussian kernel) and our main analysis (Fig. 4, 30 ms Gaussian kernel) with the same analyses done with a 15 ms Gaussian and 50 ms boxcar, as you suggest. Regardless of smoothing kernel, the major conclusions that we draw are the same.

R2.m2 —

Citation	Kernel	PCs	Var. (%)
Saxena et al. (2022)	20 ms Gaussian	2	>75
Sohn et al. (2019)	40 ms Gaussian	6	>85
Machens et al. (2010)	50 ms Gaussian	6	~95
Thura et al. (2022)	25 ms Gaussian	7	91
Wang et al. (2018)	40 ms Gaussian	9	~80
Elsayed et al. (2016)	20 ms Gaussian	10	~90
Cowley et al. (2020)	10 ms Gaussian	12	59-72
Russo et al. (2020)	25 ms Gaussian	12	87-89
Churchland and Shenoy (2007)	25 ms Gaussian	18	80
Ames et al. (2014)	25 ms Gaussian	15	~90
Zimnik and Churchland (2021)	25 ms Gaussian	20	68-78
Okazawa et al. (2021)	100 ms boxcar	10	76-89
Egger et al. (2019)	150 ms boxcar	10	>93
Remington et al. (2018b)	150 ms smoothing	10	89

Table 2: The mean Gaussian kernel from this selection of publications is 27.73 ms and the mean boxcar kernel is 133.33 ms. Abbreviations: PCs - principal components, Var. (%) - percent variance explained.

Line 204: There are some pretty modest p values in the results (line 204 and 208). How much does this depend on the smoothing kernel?

Response:

The p-values that we provide here for the KiNeT analysis depend on the number of bootstraps that we computed.

Assume the test statistic for the i^{th} bootstrap is S_i and S_{test} is what you are comparing to (e.g., 90° for the alignment angles). Then if you perform n bootstraps, then the unbiased p-value for a one-tailed test is $(1 + \sum_{i=1}^n (S_i > S_{test})) / (1 + n_{bootstraps})$. We typically bootstrapped firing rates 50 times to obtain our p-values. So the minimum p-value possible is $1/51$ for a one-tailed t-test, which is exactly what we had for our results.

There is little to no dependence on the smoothing kernel for these p-values (Fig. S11D, E).

R2.m3 —

Line 146: As I described above. I'm skeptical of this method. 90% of the variance explained by 6 dimensions is a lot if you're thinking of actual neural dynamics. But it's not that much if you smooth responses massively. How much does this depend on the smoothing kernel? Does the number of dimensions found change dramatically for smaller or larger smoothing kernels? What does that mean for interpreting low-dimensional dynamics?

Response:

As we addressed above, variance explained for trial-averaged data is not dramatically altered by different smoothing kernels (Fig. S5). 6 PCs after smoothing with a 30 ms Gaussian kernel explain over 90% of the variance, after a 15 ms Gaussian kernel 6 PCs explain 87.59% of the variance and after a 50 ms boxcar 6 PCs explain 85.67% of the variance. To reach 90% of the variance with the 15 ms Gaussian it takes 10 PCs and for the 50 ms boxcar it takes 17 of the components. Again, however we can see that 6 PCs still explain 85-90% of the variance even with smaller smoothing kernels. Thus number of dimensions needed to capture 85-90% of the variance does not change dramatically based on smaller or larger smoothing kernels. Additionally our main results are consistent across smaller or larger smoothing kernels (Fig. S11). Ultimately, this means that, as far as smoothing kernels are concerned, we can reliably interpret low-dimensional dynamics.

However we should also address that the 'smoothness' of the data can largely be attributed to trial-averaging.

Perhaps we did not adequately emphasize in our previous manuscript that we focused on the variance explained for *trial-averaged data* and in no way should this be construed as the variance explained for single-trial data. These are vastly different things. Trial-averaged data will massively reduce the amount of variance that is not immediately associated with the task such as uninstructed movements and spiking noise, and thus increase the variance associated with the task at the expense of non-task relevant dimensions.

Single-trial data is going to be much higher dimensionality, especially if animals are freely moving, interacting with a large range of stimuli and undergoing state fluctuations from drowsiness, to awake, to engaged in a task. Consistent with this assertion, work from Stringer and collaborators (Stringer et al., 2019) has shown that the single-trial dimensionality in such cases especially with extremely high-density recordings can be > 100 for more than 10,000 neurons recorded simultaneously. They also showed that when only 32 stimuli are used much of the signal associated variance is largely in this 32-dimensional space.

Additionally, as described above in response to one of the major comments, our synthetic population of hypothetical neurons are a great illustration of this issue (Fig. S3). Our original simulated matrix is 350 neurons \times 300 trials \times 2 choices \times time. We can perform PCA on either the trial-averaged data organized by RT and choice or on the single-trial data and compute the number of dimensions needed for explaining 90% of the variance in both cases. It is immediately clear that the first 7 components explain $\sim 90\%$ of the variance in the trial-averaged data (Reviewer Fig. 4A). In contrast, for the single trial analysis, the first 210 dimensions are needed to explain 90% of the variance in the data (Reviewer Fig. 4A).

So perhaps a fuller answer is that our 'smooth' low-dimensional dynamics should be appreciated in the full context that this is trial-averaged data.

R2.m4 — Same with line 710: How many trials go into these average firing rates typically? If I'm understanding the analysis right, it appears there are 308 ($2 \times 7 \times 11 \times 2$) unique conditions, so roughly 5 trials per condition given the average number of trials in a session. How much does the variance explained by the 1st 6 PCs depend on the smoothing window?

Response: Thank you for this comment. We realize that the text you are referencing from 4.8 in the Methods section was not written clearly. We have rewritten it as below, which hopefully clears up the confusion.

The general procedure for performing a PCA involved creating a 4D matrix containing all 996 units (or 801 single units as in Fig. S10) and their condition-averaged firing rate activity (i.e., peri-stimulus time histograms; section 4.7) windowed about checkerboard onset (~ -600 ms: ~ 1200 ms) and organized by level of condition (e.g., coherence, RT, or past outcome) within a reach direction. Typical matrix organization was windowed firing rate \times units \times reach \times condition (C) ($\sim 1800 \times 996 \times 2 \times C$). Condition could be coherence (7, i.e., Fig. 7A), RT (11, i.e., Fig. 4B), or past outcome (4, i.e., Fig. 8B).

So for the PCA including RT bins and choice there are 22 unique conditions, coherence and choice there are 14 unique conditions and outcome and choice there are 8 unique conditions. We have also included a figure which details the number of trials per condition for the 11 overlapping RT bins (Fig. S6B). We believe the advantage of such an analysis is that it allows us to smoothly tile the RT space. Typically, we are using on average ~ 100 trials or more per RT bin and choice (Fig. S6B) allowing us to reduce the spiking noise and focus on the task-related dynamics. This provides us with the smooth trajectories that we observed in our PCA analyses.

Now, this perhaps is problematic when it comes to performing statistics because many methods are not built for overlapping time bins. But our results do not necessitate overlapping RT bins, as the results replicate equally as well for three non-overlapping RT bins (300 - 425 ms, 425 - 500 ms, 500 - 1000 ms, Fig. S6A, Fig. 5). The

resulting dimensionality is again 6 dimensions to explain 90% of the variance.

R2.m5 — Line 695: is the Gaussian Kernel causal? If not, is it not problematic to look at pre-cue firing rates with such a wide acausal kernel?

Response: Gaussian filters are not causal, they smooth both forward and backward in time and the concern is that perhaps spikes from the poststimulus period spuriously leads to prestimulus effects that we observe. We used a Gaussian filter as it robustly smooths firing rates and is standard practice in the field as shown in Tables 1 & 2 above.

As we show in our responses to reviewers, our results are not a trivial artifact of using such a kernel. Our main results (Fig. 4) are reliably observed when we used a 50 ms causal box car kernel (Fig. S11C) or a smaller 15 ms Gaussian kernel (Fig. S11B). Even if one ignores every piece of evidence from the PCAs, our decoding and regression analyses (Fig. 6, 20 ms nonoverlapping bins) reliably demonstrate that prestimulus firing rates correlate with RT but not choice.

Finally, we believe that the Gaussian kernel would have to be much larger than 30 ms in order for this to truly be a concern. For one prestimulus effects that we observe are observed at least 400 ms before stimulus onset, well before there would be any overlap with poststimulus spiking activity. Also, if poststimulus spiking activity was of concern then we should see more of a prestimulus effect of choice, which we clearly do not observe. Thus, we can confidently assert that the use of a Gaussian kernel cannot trivially explain our results.

References

- Afshar, A., Santhanam, G., Yu, B. M., Ryu, S. I., Sahani, M., and Shenoy, K. V. (2011). Single-trial neural correlates of arm movement preparation. *Neuron*, 71:555–564.
- Ames, K. C., Ryu, S. I., and Shenoy, K. V. (2014). Neural dynamics of reaching following incorrect or absent motor preparation. *Neuron*, 81:438–451.
- Bogacz, R., Wagenmakers, E.-J., Forstmann, B. U., and Nieuwenhuis, S. (2010). The neural basis of the speed–accuracy tradeoff. *Trends in Neurosciences*, 33:10–16.
- Briggman, K. L., Abarbanel, H. D. I., and Kristan, W. B. (2005). Optical imaging of neuronal populations during decision-making. *Science*, 307:896–901.
- Brody, C. D. and Hanks, T. D. (2016). Neural underpinnings of the evidence accumulator. *Current Opinion in Neurobiology*, 37:149–157.
- Brunton, B. W., Botvinick, M. M., and Brody, C. D. (2013). Rats and humans can optimally accumulate evidence for decision-making. *Science*, 340:95–98.
- Chandrasekaran, C., Peixoto, D., Newsome, W. T., and Shenoy, K. V. (2017). Laminar differences in decision-related neural activity in dorsal premotor cortex. *Nature Communications*, 8:614.
- Churchland, A. K., Kiani, R., and Shadlen, M. N. (2008). Decision-making with multiple alternatives. *Nature Neuroscience*, 11:693–702.
- Churchland, M. M., Afshar, A., and Shenoy, K. V. (2006). A central source of movement variability. *Neuron*, 52:1085–1096.
- Churchland, M. M., Cunningham, J. P., Kaufman, M. T., Foster, J. D., Nuyujukian, P., Ryu, S. I., Shenoy, K. V., and Shenoy, K. V. (2012). Neural population dynamics during reaching. *Nature*, 487:51–56.
- Churchland, M. M. and Shenoy, K. V. (2007). Temporal complexity and heterogeneity of single-neuron activity in premotor and motor cortex. *Journal of Neurophysiology*, 97:4235–4257.
- Cisek, P. (2012). Making decisions through a distributed consensus. *Current Opinion in Neurobiology*, 22:927–936.
- Coallier, E., Michelet, T., and Kalaska, J. F. (2015). Dorsal premotor cortex: neural correlates of reach target decisions based on a color-location matching rule and conflicting sensory evidence. *Journal of Neurophysiology*, 113:3543–3573.

- Cook, E. P. and Maunsell, J. H. (2002). Dynamics of neuronal responses in macaque mt and vip during motion detection. *Nature Neuroscience*, 5:985–994.
- Cowley, B. R., Snyder, A. C., Acar, K., Williamson, R. C., Yu, B. M., and Smith, M. A. (2020). Slow drift of neural activity as a signature of impulsivity in macaque visual and prefrontal cortex. *Neuron*, 108:551–567.e8.
- Dutilh, G., Ravenzwaaij, D. V., Nieuwenhuis, S., der Maas, H. L. V., Forstmann, B. U., and Wagenmakers, E. J. (2012). How to measure post-error slowing: A confound and a simple solution. *Journal of Mathematical Psychology*, 56:208–216.
- Egger, S. W., Remington, E. D., Chang, C. J., and Jazayeri, M. (2019). Internal models of sensorimotor integration regulate cortical dynamics. *Nature Neuroscience*, 22:1871–1882.
- Elsayed, G. F. and Cunningham, J. P. (2017). Structure in neural population recordings: An expected byproduct of simpler phenomena? *Nature Neuroscience*, 20:1310–1318.
- Elsayed, G. F., Lara, A. H., Kaufman, M. T., Churchland, M. M., and Cunningham, J. P. (2016). Reorganization between preparatory and movement population responses in motor cortex. *Nature Communications*, 7.
- Fitzgerald, J. K., Freedman, D. J., Fanini, A., Bennur, S., Gold, J. I., and Assad, J. A. (2013). Biased associative representations in parietal cortex. *Neuron*, 77:180–191.
- Ganguli, S., Bisley, J. W., Roitman, J. D., Shadlen, M. N., Goldberg, M. E., and Miller, K. D. (2008). One-dimensional dynamics of attention and decision making in lip. *Neuron*, 58(1):15–25.
- Gao, P. and Ganguli, S. (2015). On simplicity and complexity in the brave new world of large-scale neuroscience. *Current Opinion in Neurobiology*, 32:148–155.
- Gold, J. I. and Shadlen, M. N. (2007). The neural basis of decision making. *Annual Review of Neuroscience*, 30:535–574.
- Guo, Z. V., Li, N., Huber, D., Ophir, E., Gutnisky, D., Ting, J. T., Feng, G., and Svoboda, K. (2014). Flow of cortical activity underlying a tactile decision in mice. *Neuron*, 81:179–194.
- Hanks, T., Kiani, R., and Shadlen, M. N. (2014). A neural mechanism of speed-accuracy tradeoff in macaque area LIP. *eLife*, 3.
- Hanks, T. D., Kopec, C. D., Brunton, B. W., Duan, C. A., Erlich, J. C., and Brody, C. D. (2015). Distinct relationships of parietal and prefrontal cortices to evidence accumulation. *Nature*, 520:220–223.
- Heitz, R. P. and Schall, J. D. (2012). Neural mechanisms of speed-accuracy tradeoff. *Neuron*, 76:616–628.
- Izhikevich, E. M. (2003). Simple model of spiking neurons. *IEEE Transactions on Neural Networks*, 14(6):1569–1572.
- Jun, J. K., Miller, P., Hernández, A., Zainos, A., Lemus, L., Brody, C. D., and Romo, R. (2010). Heterogeneous population coding of a short-term memory and decision task. *Journal of Neuroscience*, 30:916–929.
- Kato, S., Kaplan, H. S., Schrödel, T., Skora, S., Lindsay, T. H., Yemini, E., Lockery, S., and Zimmer, M. (2015). Global brain dynamics embed the motor command sequence of *caenorhabditis elegans*. *Cell*, 163:656–669.
- Kaufman, M. T., Seely, J. S., Sussillo, D., Ryu, S. I., Shenoy, K. V., and Churchland, M. M. (2016). The largest response component in the motor cortex reflects movement timing but not movement type. *eneuro*, 3:ENEURO.0085–16.2016.
- Kelly, S. P. and O’Connell, R. G. (2013). Internal and external influences on the rate of sensory evidence accumulation in the human brain. *Journal of Neuroscience*, 33:19434–19441.
- Kiani, R., Churchland, A. K., and Shadlen, M. N. (2013). Integration of direction cues is invariant to the temporal gap between them. *Journal of Neuroscience*, 33:16483–16489.
- Kleinman, M., Chandrasekaran, C., and Kao, J. C. (2019). Recurrent neural network models of multi-area computation underlying decision-making. *bioRxiv*.
- Kobak, D., Brendel, W., Constantinidis, C., Feierstein, C. E., Kepecs, A., Mainen, Z. F., Qi, X.-L., Romo, R., Uchida, N., and Machens, C. K. (2016). Demixed principal component analysis of neural population data. *eLife*, 5.

- Machens, C. K., Romo, R., and Brody, C. D. (2010). Functional, but not anatomical, separation of "what" and "when" in prefrontal cortex. *Journal of Neuroscience*, 30:350–360.
- Mante, V., Sussillo, D., Shenoy, K. V., and Newsome, W. T. (2013). Context-dependent computation by recurrent dynamics in prefrontal cortex. *Nature*, 503:78–84.
- Mohan, K., Zhu, O., and Freedman, D. J. (2021). Interaction between neuronal encoding and population dynamics during categorization task switching in parietal cortex. *Neuron*, 109:700–712.e4.
- Murphy, P. R., Boonstra, E., and Nieuwenhuis, S. (2016). Global gain modulation generates time-dependent urgency during perceptual choice in humans. *Nature Communications*, 7.
- Niyogi, R. K. and Wong-Lin, K. (2013). Dynamic excitatory and inhibitory gain modulation can produce flexible, robust and optimal decision-making. *PLoS computational biology*, 9(6):e1003099.
- Okazawa, G., Hatch, C. E., Mancoo, A., Machens, C. K., and Kiani, R. (2021). Representational geometry of perceptual decisions in the monkey parietal cortex. *Cell*, 184:3748–3761.e18.
- Oram, M. W. (2010). Contrast induced changes in response latency depend on stimulus specificity. *Journal of Physiology-Paris*, 104(3):167–175.
- Pandarínath, C., O’Shea, D. J., Collins, J., Jozefowicz, R., Stavisky, S. D., Kao, J. C., Trautmann, E. M., Kaufman, M. T., Ryu, S. I., Hochberg, L. R., Henderson, J. M., Shenoy, K. V., Abbott, L. F., and Sussillo, D. (2018). Inferring single-trial neural population dynamics using sequential auto-encoders. *Nature Methods*, 15:805–815.
- Pereira, M., Megevand, P., Tan, M. X., Chang, W., Wang, S., Rezai, A., Seeck, M., Corniola, M., Momjian, S., Bernasconi, F., Blanke, O., and Faivre, N. (2021). Evidence accumulation relates to perceptual consciousness and monitoring. *Nature Communications*, 12:3261.
- Purcell, B. A. and Kiani, R. (2016). Neural mechanisms of post-error adjustments of decision policy in parietal cortex. *Neuron*, 89:658–671.
- Ratcliff, R., Smith, P. L., Brown, S. D., and McKoon, G. (2016). Diffusion decision model: Current issues and history. *Trends in Cognitive Sciences*, 20:260–281.
- Remington, E. D., Egger, S. W., Narain, D., Wang, J., and Jazayeri, M. (2018a). A dynamical systems perspective on flexible motor timing. *Trends in cognitive sciences*, 22:938–952.
- Remington, E. D., Narain, D., Hosseini, E. A., and Jazayeri, M. (2018b). Flexible sensorimotor computations through rapid reconfiguration of cortical dynamics. *Neuron*, 98:1005–1019.e5.
- Roitman, J. D. and Shadlen, M. N. (2002). Response of neurons in the lateral intraparietal area during a combined visual discrimination reaction time task. *The Journal of Neuroscience*, 22:9475–9489.
- Russo, A. A., Khajeh, R., Bittner, S. R., Perkins, S. M., Cunningham, J. P., Abbott, L., and Churchland, M. M. (2020). Neural trajectories in the supplementary motor area and motor cortex exhibit distinct geometries, compatible with different classes of computation. *Neuron*, 107:745–758.e6.
- Saxena, S., Russo, A. A., Cunningham, J., and Churchland, M. M. (2022). Motor cortex activity across movement speeds is predicted by network-level strategies for generating muscle activity. *eLife*, 11:11.
- Shadlen, M. N. and Newsome, W. T. (2001). Neural basis of a perceptual decision in the parietal cortex (area LIP) of the rhesus monkey. *Journal of Neurophysiology*, 86:1916–1936.
- Sohn, H., Narain, D., Meirhaeghe, N., and Jazayeri, M. (2019). Bayesian computation through cortical latent dynamics. *Neuron*, 103:934–947.e5.
- Steinemann, N. A., Stine, G. M., Trautmann, E. M., Zylberberg, A., Wolpert, D. M., and Shadlen, M. N. (2022). Direct observation of the neural computations underlying a single decision. *bioRxiv*.
- Stringer, C., Pachitariu, M., Steinmetz, N., Carandini, M., and Harris, K. D. (2019). High-dimensional geometry of population responses in visual cortex. *Nature*, 571(7765):361–365.
- Thura, D., Beauregard-Racine, J., Fradet, C.-W., and Cisek, P. (2012). Decision making by urgency gating: theory and experimental support. *Usher and McClelland*, 108:2912–2930.
- Thura, D., Cabana, J.-F., Feghaly, A., and Cisek, P. (2022). Integrated neural dynamics of sensorimotor decisions and actions. *PLOS Biology*, 20:e3001861.
- Thura, D. and Cisek, P. (2016). Modulation of premotor and primary motor cortical activity during volitional adjustments of speed-accuracy trade-offs. *Journal of Neuroscience*, 36:938–956.

- Thura, D., Guberman, G., and Cisek, P. (2017). Trial-to-trial adjustments of speed-accuracy trade-offs in premotor and primary motor cortex. *Journal of Neurophysiology*, 117:665–683.
- Trautmann, E. M., Stavisky, S. D., Lahiri, S., Ames, K. C., Kaufman, M. T., O’Shea, D. J., Vyas, S., Sun, X., Ryu, S. I., Ganguli, S., and Shenoy, K. V. (2019). Accurate estimation of neural population dynamics without spike sorting. *Neuron*, 103:292–308.e4.
- Vyas, S., Golub, M. D., Sussillo, D., and Shenoy, K. V. (2020). Computation through neural population dynamics. *Annual Review of Neuroscience*, 43:249–275.
- Wang, J., Narain, D., Hosseini, E. A., and Jazayeri, M. (2018). Flexible timing by temporal scaling of cortical responses. *Nature Neuroscience*, 21:102–112.
- Wang, X.-J. (2002). Probabilistic decision making by slow reverberation in cortical circuits. *Neuron*, 36(5):955–968.
- Wei, Z., Inagaki, H., Li, N., Svoboda, K., and Druckmann, S. (2019). An orderly single-trial organization of population dynamics in premotor cortex predicts behavioral variability. *Nature communications*, 10(1):216.
- Williams, A. H., Kim, T. H., Wang, F., Vyas, S., Ryu, S. I., Shenoy, K. V., Schnitzer, M., Kolda, T. G., and Ganguli, S. (2018). Unsupervised discovery of demixed, low-dimensional neural dynamics across multiple timescales through tensor component analysis. *Neuron*, 98:1099–1115.e8.
- Zimnik, A. J. and Churchland, M. M. (2021). Independent generation of sequence elements by motor cortex. *Nature Neuroscience*, 24:412–424.

Reviewer Comments

Reviewer #1

The authors have revised their paper and provided a very long and thorough response to all of the comments made in the first round. In particular, they better focused their description of what they mean by a dynamical system and what hypotheses are being compared. This is an important improvement. They have answered all of my questions and I'm generally satisfied by their responses, with just two exceptions. I will describe these remaining issues below.

First, however, I would repeat my main critique – which is that these results are not really very novel. Almost all of the main results have been reported before, and the only observation I found surprising is that there was no effect of pre-stimulus activity on the choice made in the monkeys. In the revised manuscript, the authors looked at this more carefully and convinced me that it is the case. However, this is just a null result. It helps one to distinguish among different candidate models, but all of those models are really just relatively minor variations of each other. So in short, it's an interesting paper, but I wonder how much impact it would have on the broad audience of this journal.

Returning to the specific issues, that in my opinion, still remain unresolved:

In the first round, I raised the issue of the overlapping RT bins and how that could distort the results and violate assumptions of statistical tests. The authors repeat some of their analyses with non-overlapping bins, showing similar results for those (though not all). However, this has been relegated to a supplemental figure, while the main results still use highly overlapping data sets. That means the main results still distort the data in ways that potentially mislead readers. For example, Figure 6D shows how distance grows over time in different RT bins, and the authors discuss how these appear to have different slopes (line 427-436). But all of those differences are subject to averaging artifacts because some trials drop out of parts of these averages at different times. Do you get significantly different slopes when bins are not overlapping? So I apologize for not being clearer: I meant that the analyses in the main text should not use overlapping RT bins and the conclusions should be based on those.

I also commented about the use of "multi-units", which I think is both unwise and unnecessary. I think it is unwise because it amounts to allowing noise and sloppiness into one's data, and I'll come back to that. But it is also really unnecessary in this case. There are already 801 well-isolated neurons, so adding another 195 (<20%) is not really going to change the results. That is indeed the case, as shown in Fig S9. But again, why relegate the cleaner data to the supplemental figures while leaving potentially distorted results in the main text?

On the issue of whether "multi-units" are useful, the authors respond by citing the Trautmann et al. 2019 paper, which shows that the inclusion of multi-unit data does not change neural space trajectories. The reason is simply because such analyses are already constructing a weighted sum of firing rates, so poor isolation just creates a "pre-weighted" sum of two or more neurons that will just be treated together in the total population. I understand that some people welcome this argument, but I strongly disagree with it and the attitude it represents. As far as I'm concerned, what the Trautmann et al. paper really shows is not that it's ok to use multi-unit data in neural space analyses – what it really shows is that neural space analyses are inherently limited. If a method gives similar results whether we are more or less precise in our data collection, then I would say that is a cause for worry. In particular, we already know that precision is important when looking at the activity patterns of individual cells. In fact previous results from these same authors showed that there are important and meaningful differences in decision timing between superficial versus deep layers of cortex

(Chandrasekaran et al. 2017 Nat Comm). Those would obviously not get mixed, but other studies have also shown that there are timing differences between narrow vs. broad spiking neurons (e.g. Song & McPeck 2010 J Neurophys), or response properties between cortico-cortical versus cortico-collicular projecting neurons (e.g. Ferraina, Pare & Wurtz, 2002 J Neurophys), findings that would be lost if one was not careful with isolation. And of course there are countless studies in rodents and zebrafish in which clear differences in properties are found when different cell types are analyzed separately (differently projecting neurons, different kinds of interneurons). That is the precision toward which we should strive. So no, I do not think it is ok to merge cells together, and analyses that are insensitive to such merging are an impoverished window into the data. Finally, even neural space analyses will suffer if one includes multi units: If enough cells are merged together then important variance between them will be lost and the remaining components are likely to be missing relevant dynamics (e.g. if interneurons are mixed with pyramidal cells). Also consider the analysis of PC loadings shown in Figure 2 in the rebuttal document: if cells are not well isolated then this analysis will fail to pick up any structure or clusters in the population. Again, I just don't understand why one would prefer to have a larger data set that raises potential concerns instead of one that is only 20% smaller but much cleaner.

Reviewer #2

Thank you for the very detailed and thoughtful response to my and the other reviewer's concerns.

I have no further comments.

Response to reviewers

Reviewer 1

Comments:

The authors have revised their paper and provided a very long and thorough response to all of the comments made in the first round. In particular, they better focused their description of what they mean by a dynamical system and what hypotheses are being compared. This is an important improvement. They have answered all of my questions and I'm generally satisfied by their responses, with just two exceptions. I will describe these remaining issues below.

Response:

Thank you for your generally positive comments about the quality of our manuscript. We agree that the focused description of what we mean by a dynamical system and the hypotheses compared is an important improvement of the manuscript. We are grateful to you and the other reviewer's guidance for making our manuscript more impactful.

Please find our responses to your two other comments below. We felt the comment about multi-units made multiple points and as such broke it up across two points.

R1.1 —

First, however, I would repeat my main critique – which is that these results are not really very novel. Almost all of the main results have been reported before, and the only observation I found surprising is that there was no effect of pre-stimulus activity on the choice made in the monkeys. In the revised manuscript, the authors looked at this more carefully and convinced me that it is the case. However, this is just a null result. It helps one to distinguish among different candidate models, but all of those models are really just relatively minor variations of each other. So in short, it's an interesting paper, but I wonder how much impact it would have on the broad audience of this journal.

Response: Thank you for this comment. Naturally, as authors we would claim that this work is very novel and believe one would be hard pressed to find an author who would claim the opposite outside of reviews and replications. Unfortunately a metric for measuring novelty does not exist and the discussion of how to measure novelty is highly philosophical in nature and thus difficult to resolve. That being said, here are five reasons why we think this paper is exciting and of interest to a broad audience:

First, to the point, here are three findings that we find novel, are excited about, or provide an advance over prior work. The effect of the initial condition is far stronger than the effect of stimulus condition. This effect is quite surprising given the research of groups led by those such as Ratcliff or Shadlen which deem decision-making to be a 1-dimensional process reliant solely on stimulus evidence (e.g., Ratcliff et al., 2016; Steinemann et al., 2022). Next the initial condition hypothesis has been demonstrated in motor tasks (i.e., delayed reach to sample) but not in a decision-making context. How the initial condition combines with input was an outstanding question that we have now begun to address. Finally, our findings bridge results from speed-accuracy tradeoff, post-outcome adjustment, timing tasks, and motor planning demonstrating a powerful framework for understanding decision-making. Which actually leads into the following point.

Second both you and R2 found these results compelling and meaningful. For instance, as you pointed out in the first round of comments : 'The results are not very novel as the individual phenomena have previously been shown, but it is *compelling* to see the relationships among these phenomena analyzed together with congruent methods.' In fact, this comment indicates that these results may be compelling for a broad audience as they tie together phenomena from across the field. Thus, our effort here will make it easier to follow the field in one

coherent story rather than having to piece together the state of the field across multiple papers.

Third, if these models are just minor variants of each other that is because these are the models used in the literature. Collectively, the decision-making community (perhaps including yourself) has put in an enormous amount of work developing these various decision-making models. Thus we feel arbitrating between them with data and modelling to be a rigorous approach that is novel and compelling. Our results show that the simple models cannot capture our data and that a gain term is necessary for explaining these results. So this manuscript should help us converge on a better model for decision-making.

Fourth, this research has received positive interest and questions from the community at two competitive conferences. An abstract of this manuscript, incorporating feedback from you and R2, was accepted as a poster for COSYNE 2023, a conference with roughly 48% acceptance rate. It was also selected for a talk at Advances in Motor Learning and Control (MLMC) 2022, a SFN satellite conference. Consequently we are eager to present our work as transparently as possible and believe the community is eager to see this work published so they can judge for themselves.

Fifth, our opinion is that a paper is novel if it generates new hypotheses that could be tested by other researchers in the field (perhaps including you from our read of the comments). For example, two new directions for the lab have been inspired by our results. We are actively interested in pursuing simultaneous recordings with higher density probes (i.e., Neuropixels) in PMd, supplementary motor area, and the anterior cingulate cortex during decision-making to identify the sources of prestimulus covariation with RT and trial outcome. The use of Neuropixels will also allow us to perform further single-trial analyses as we are inspired by what further LFADs analyses can reveal.

We are not only happy that our research can advance our understanding in the field but that it also replicates previous findings. It means our research lays on a firm bedrock of previous findings from different labs and contributes to those foundations. We believe this is fundamental for advances in science. Finally, after the rigorous peer review from you and R2 the revised paper is going to be even more exciting for potential readers.

R1.2 —

Returning to the specific issues, that in my opinion, still remain unresolved:

In the first round, I raised the issue of the overlapping RT bins and how that could distort the results and violate assumptions of statistical tests. The authors repeat some of their analyses with non-overlapping bins, showing similar results for those (though not all). However, this has been relegated to a supplemental figure, while the main results still use highly overlapping data sets. That means the main results still distort the data in ways that potentially mislead readers. For example, Figure 6D shows how distance grows over time in different RT bins, and the authors discuss how these appear to have different slopes (line 427-436). But all of those differences are subject to averaging artifacts because some trials drop out of parts of these averages at different times. Do you get significantly different slopes when bins are not overlapping? So I apologize for not being clearer: I meant that the analyses in the main text should not use overlapping RT bins and the conclusions should be based on those.

Response:

Thank you for this comment. In our opinion, the visualization and analysis with the overlapping bins is very useful because it helps provide a parallel to our single-trial analysis by helping us assess how firing rates covary with RT. But we appreciate the statistical issue and your advice. We have now taken your advice and moved the PCA and KiNeT analysis figure with nonoverlapping bins (Fig. 5) into the main text right after Fig. 4. The figures are paired together and referred to next to each other in the text. This will ensure that readers are not confused or misled by any claims in the paper.

We also investigated whether the covariation between stimulus coherence and averaged firing rate exists for Fig. 7d even if we performed the analysis with non-overlapping bins. Fig. 7h-k shows the results of this analysis.

Consistent with the analysis for nonoverlapping bins, we found that the choice-selectivity signal appears to have a steeper rate of rise for faster compared to slower RT bins for the easiest coherence but less so for the harder coherences. Significance tests for partial correlation analyses are underpowered with such few data points (3 initial conditions \times 7 coherences). Instead, we report bootstrap confidence intervals for these correlation values. Again, the conclusions of the slope analysis is the same regardless of whether we use overlapping or nonoverlapping bins.

This result again suggests that the dynamics of choice signals depend on both the stimulus condition and the initial condition. However, we are very clear in the manuscript that the effect of the initial condition is far stronger than the effect of stimulus condition, again a considerable advance over prior work.

Again, these effects were reaffirmed when using non-overlapping bins for the analyses. Fig. 7h shows the choice-selectivity signals for the three non-overlapping bins considered in Fig. 5a. For instance, for the easiest coherence, activity increases faster for faster RTs compared to slower RTs but does not appear to do so for the harder coherences, consistent with the patterns observed in Fig. 7d-g. We computed the average choice-selectivity for each of these three RT bins in the 125 to 375 ms period and again found both initial condition and coherence had an impact on the average choice-selectivity in the 125 to 375 ms period (mean and 99% confidence intervals, coherence: $r = 0.225$ (0.204, 0.241), initial conditions: $r = 0.77$ (0.76, 0.78), $p = 0.0196$ for both cases, 50 bootstraps, Fig. 7i). Subsequent analysis of the latency and slope of these choice-selectivity signals (Fig. 7j-k) were also consistent with the conclusions from non-overlapping bins. Latency was strongly impacted by initial condition ($r_{18} = -.63$, (-0.72, -0.53)) and modestly by coherence ($r_{18} = -0.38$ (-0.43, -0.33)). Slope was again strongly influenced by the initial condition ($r_{18} = 0.91$ (0.88, 0.93), $p = 0.0196$) but had almost no relationship to coherence ($r_{18} = -0.007$, (-0.099, 0.085), $p = 0.46$).

R1.3 —

I also commented about the use of “multi-units”, which I think is both unwise and unnecessary. I think it is unwise because it amounts to allowing noise and sloppiness into one’s data, and I’ll come back to that. But it is also really unnecessary in this case. There are already 801 well-isolated neurons, so adding another 195 (<20%) is not really going to change the results. That is indeed the case, as shown in Fig S9. But again, why relegate the cleaner data to the supplemental figures while leaving potentially distorted results in the main text?

Response:

We thank you for this comment. The prestimulus covariation with RT and trial outcome is stronger in some of the lower PCs which by definition have lower variance. Moreover, as our regression analysis shows only \sim 20% of units had a prestimulus correlation with RT, and within a coherence, because you have fewer trials, the percentages drop even further. So losing 20% of these units will reduce the overall power of our analyses by reducing meaningful neural variance. In no way are we trying to be sloppy or add noise to our dataset. As is common practice, all analyses should be performed with as much power as one can manage and for us the inclusion of multi-units provides us with more power for the analyses. Unlike many studies, we even provide a quality assessment as to the number of ISI violations in these multi-units. Our opinion is that removing the multi-units actually reduces the power in the analyses and thus makes it harder to reliably interpret effects. We also feel it is potentially distorting to readers to have just single neurons in the PCA analysis while at the same time using multi-units for decoding, and single-trial analyses. So we have decided to keep our current approach of showing the main results with multi-units and including the supplementary figure with only single units.

R1.4 —

On the issue of whether “multi-units” are useful, the authors respond by citing the Trautmann et al. 2019 paper, which shows that the inclusion of multi-unit data does not change neural space trajectories. The reason is simply because such analyses are already constructing a weighted sum of firing rates, so poor isolation just creates a “pre-weighted” sum of two or more neurons that will just be treated together in the total population. I understand that some people welcome this argument, but I strongly disagree with it and the attitude it represents. As far as I’m concerned, what the Trautmann et al. paper really shows is not that it’s ok to use multi-unit data in neural space analyses – what it really shows is that neural space analyses are inherently limited. If a method gives similar results whether we are more or less precise in our data collection, then I would say that is a cause for worry. In particular, we already know that precision is important when looking at the activity patterns of individual cells. In fact previous results from these same authors showed that there are important and meaningful differences in decision timing between superficial versus deep layers of cortex (Chandrasekaran et al. 2017 Nat Comm). Those would obviously not get mixed, but other studies have also shown that there are timing differences between narrow vs. broad spiking neurons (e.g. Song & McPeck 2010 J Neurophys), or response properties between cortico-cortical versus cortico-collicular projecting neurons (e.g. Ferraina, Pare & Wurtz, 2002 J Neurophys), findings that would be lost if one was not careful with isolation. And of course there are countless studies in rodents and zebrafish in which clear differences in properties are found when different cell types are analyzed separately (differently projecting neurons, different kinds of interneurons). That is the precision toward which we should strive. So no, I do not think it is ok to merge cells together, and analyses that are insensitive to such merging are an impoverished window into the data.

Finally, even neural space analyses will suffer if one includes multi units: If enough cells are merged together then important variance between them will be lost and the remaining components are likely to be missing relevant dynamics (e.g. if interneurons are mixed with pyramidal cells). Also consider the analysis of PC loadings shown in Figure 2 in the rebuttal document: if cells are not well isolated then this analysis will fail to pick up any structure or clusters in the population. Again, I just don’t understand why one would prefer to have a larger data set that raises potential concerns instead of one that is only 20% smaller but much cleaner.

Response:

We completely agree that only well-isolated single units should be used for analyses of how individual cell types are involved in cognitive or motor function. Use of multi-units would not be appropriate in such analyses. In manuscripts from our lab that are focused on cell types, we only use high-quality single units (Lee et al., 2021). In the afore cited publication, we developed a novel approach, WaveMAP, that allows one to infer candidate cell type from waveform properties. In this case high-quality single units are hugely important as the scientific question concerns the existence of cell types. Similarly, in the lab there is another manuscript under preparation on cell types in V1. For this manuscript, we began with a database of 2500 units, identified by KiloSort (Pachitariu et al., 2016), from a recent manuscript examining Neuropixel recordings in V1 (Trepka et al., 2022). After stringent curation with various quality metrics we ended up with ~800 well-isolated single units.

That being said, the point of this study is not to identify cell types or their specific contributions, it is to understand neural population dynamics as a whole. Principal component analysis and allied methods, operate to reveal patterns of activity at the level of the neural population and in our opinion are more robust with the inclusion of multi-units. One should curate the data for an analysis based upon what would be most beneficial to that analysis. In this case single neurons and multi-units are both useful to our analyses as they both reveal population-level patterns. In no way are we suggesting that one should wholesale use multi-units for all analyses. As we describe above, all analyses should be done with as much power as appropriate for the questions at hand and the inclusion of multi-units provides us with additional power for all analyses and thus makes us more confident in our statements.

We do not agree with the argument that it is necessarily a flaw in an analysis approach if you get similar results from an analysis based upon different data curation practices. For instance, a grand average means of single

neurons and multi-unit firing rates will look quite similar, does that imply we should reject the mean as an analysis? All the Trautmann et al. paper implies is that for a lot of these types of analyses where we are trying to find the low-dimensional subspace involved in various cognitive and motor tasks, the inclusion of multi-units does not distort the recovery of these spaces and in our opinion might actually help since the multi-units often have slightly higher firing rates and help delineate these low-dimensional subspaces better.

Concerning your insightful critique about the PC loading figure, we apologize for missing this detail previously. As such we have updated the PC loading figure to only use single units (Fig. S22).

Reviewer 2

Comments:

R2.1 — Thank you for the very detailed and thoughtful response to my and the other reviewer’s concerns.

I have no further comments.

Response:

Thank you for your time and effort that you gave in generating thoughtful critiques and helpful suggestions during this process. The quality of the manuscript is much improved and we have a richer appreciation for our data as a result.

References

- Lee, E. K., Balasubramanian, H., Tsolias, A., Anakwe, S. U., Medalla, M., Shenoy, K. V., and Chandrasekaran, C. (2021). Non-linear dimensionality reduction on extracellular waveforms reveals cell type diversity in premotor cortex. *eLife*, 10.
- Pachitariu, M., Steinmetz, N., Kadir, S., Carandini, M., and D., H. K. (2016). Kilosort: realtime spike-sorting for extracellular electrophysiology with hundreds of channels. *bioRxiv*.
- Ratcliff, R., Smith, P. L., Brown, S. D., and McKoon, G. (2016). Diffusion decision model: Current issues and history. *Trends in Cognitive Sciences*, 20:260–281.
- Steinemann, N. A., Stine, G. M., Trautmann, E. M., Zylberberg, A., Wolpert, D. M., and Shadlen, M. N. (2022). Direct observation of the neural computations underlying a single decision. *bioRxiv*.
- Trepka, E. B., Zhu, S., Xia, R., Chen, X., and Moore, T. (2022). Functional interactions among neurons within single columns of macaque v1. *eLife*, 11:e79322.